



# Retrieval of aerosol properties using relative radiance measurements from an all-sky camera

Roberto Román[1], Juan C. Antuña-Sánchez[1], Victoria E. Cachorro[1], Carlos Toledano[1], Benjamín Torres[2], David Mateos[1], David Fuertes[3], César López[4], Ramiro González[1], Tatyana Lapionok[2], Oleg Dubovik[2], and Ángel M. de Frutos[1]

[1]Group of Atmospheric Optics (GOA-UVa), University of Valladolid, 47011, Valladolid, Spain
[2]Laboratoire d'Optique Atmosphérique, CNRS, Lille 1 University, France
[3]GRASP-SAS, Remote Sensing Developments, Villeneuve D'Ascq, France
[4]Sieltec Canarias S.L., La Laguna, Spain

**Correspondence:** Roberto Román (robertor@goa.uva.es)

**Abstract.** This paper explores the potential of all-sky cameras to retrieve aerosol properties with GRASP code (Generalized Retrieval of Atmosphere and Surface Properties). To this end, normalized sky radiances (NSR) extracted from an all-sky camera at three effective wavelengths (467, 536 and 605 nm) are used in this study. NSR observations are a set of relative (uncalibrated) sky radiances in arbitrary units. NSR observations have been simulated for different aerosol loads and types with the forward radiative transfer module of GRASP, indicating that NSR observations contain information about the aerosol type as well as about the aerosol optical depth (AOD), at least for low and moderate aerosol loads. An additional sensitivity study with synthetic data has been carried out to quantify the theoretical accuracy and precision on the aerosol properties (AOD, size distribution parameters, etc.) retrieved by GRASP using NSR observations as input. As result, the theoretical accuracy on AOD is within $\pm$ 0.02 for AOD values lower or equal than 0.4; while the theoretical precision goes from 0.01 to 0.05 when AOD at 467 nm varies from 0.1 to 0.5. NSR measurements recorded at Valladolid (Spain) with an all-sky camera for more than two years have been inverted with GRASP. The retrieved aerosol properties are compared with independent values provided by co-located AERONET (AErosol RObotic NETwork) measurements. AOD from both data sets correlate with determination coefficient ($r^2$) values about 0.87. Finally, the novel multi-pixel approach of GRASP is applied to daily camera radiances together, by constraining the temporal variation in certain aerosol properties. This temporal linkage (multi-pixel approach) provides promising results, reducing the highly temporal variation in some aerosol properties retrieved with the standard (one by one or single-pixel) approach. This work implies an advance in the use of all-sky cameras for the retrieval of aerosol properties.

## 1 Introduction

Atmospheric aerosol particles (hereinafter, 'aerosols'), which are the solid and liquid particles floating in the atmosphere (Willeke et al., 1993), impact on the Earth radiative balance mainly through aerosol-radiation and aerosol-cloud interactions (Boucher et al., 2013). These interactions are related to the direct absorption and scattering of incoming solar radiation, as





well as the modification of cloud properties, like cloud lifetime and albedo, since aerosols act as water droplet or ice crystal nuclei. These interactions depend significantly on the aerosol load and properties like the aerosol size distribution, chemical composition or refractive index. These properties vary with the aerosol type. The high spatial and temporal variability of

aerosols on global scale makes that the effect of aerosols on the energy budget are still largely uncertain (Boucher et al., 2013; IPCC, 2014). Thus, aerosol property monitoring on a global scale is a crucial task.

Aerosol measurements from satellite instruments like MODIS (MODerate resolution Imaging Spectroradiometer; Remer et al. 2005; Levy et al. 2013), are frequently used for global monitoring of aerosol properties like the aerosol optical depth (AOD). Unfortunately, the temporal resolution of these satellite measurements is not high and, in addition, these measurements

need to be subjected to calibration/validation procedures using ground-based measurements. Some global networks focused on aerosol measurements, like AERONET (AErosol RObotic NETwork[1]; Holben et al. 1998), are used to this end. AERONET is a federation of ground-based remote sensing aerosol networks with measurement stations distributed worldwide, using all the same standard instrument model. This standard instrument is a Sun-sky (recently also Moon) photometer, which is capable to measure direct Sun irradiance and sky radiances at several wavelengths. Sun measurements are used to derive spectral

AOD (Giles et al., 2019); while sky radiances, which contain important aerosol information (Nakajima et al., 1996), are used together with AOD in an inversion algorithm to retrieve other aerosol properties like volume size distribution or refractive index (Dubovik and King, 2000; Dubovik et al., 2000, 2006; Sinyuk et al., 2020).

A new inversion algorithm that allows a similar retrieval of aerosol properties, is the GRASP code (Generalized Retrieval of Atmosphere and Surface Properties[2]; Dubovik et al. 2014). This algorithm is capable of retrieving aerosol properties by

inverting Sun-sky photometer measurements, but its versatility also allows inversion of other kinds of measurements like: lidar/ceilometer signal plus AOD and sky radiances (Lopatin et al., 2013; Benavent-Oltra et al., 2017; Román et al., 2018; Titos et al., 2019; Herreras et al., 2019; Molero et al., 2020; Tsekeri et al., 2017); satellite measurements (Chen et al., 2020; Wei et al., 2021); nephelometer data (Espinosa et al., 2017); and even only AOD measurements (Torres et al., 2017).

As mentioned, sky radiance measurements are useful to retrieve aerosol properties, these measurements being usually taken

with high accuracy and precision by sky photometers. However, sky radiance measurements from photometers over different sky positions (almucantar geometry, for example) are not collected simultaneously, since the instrument needs to scan the various sky positions in a sequence that results in time lag among the measurements. An instrument which is sensitive to sky radiance at every point of sky in a short time interval is the all-sky camera. All-sky cameras are mainly used to detect clouds (see Tapakis and Charalambides (2013) and references therein), but some works have demonstrated that they are capable of

other purposes like: obtaining sky radiance and luminance measurements (Rossini and Krenzinger, 2007; Román et al., 2012; Tohsing et al., 2014), characterizing aerosol properties (Cazorla et al., 2008), or performing advanced aerosol characterization, by combining in GRASP the sky radiance from an all-sky camera and other instrument data like lunar photometer and lidar (Román et al., 2017; Benavent-Oltra et al., 2019). In this sense, Antuña Sánchez et al. (2021) recently published a new method to extract normalized sky radiances (NSR), which are relative (normalized) sky radiances in arbitrary units. The norm in this

---

[1]https://aeronet.gsfc.nasa.gov

[2]www.grasp-open.com



case is calculated as the sum of of all observed multi-angular radiances for each wavelength. NSR are calculated at three effective wavelengths from an all-sky camera equiped with narrower than usual spectral filters. These NSR measurements potentially contain enough information to retrieve some aerosol properties with an inversion algorithm such as GRASP.

In this framework, the main objective of this work is to propose a new methodology to retrieve aerosol properties using normalized sky radiance measurements from an all-sky camera as input on GRASP code. Another goal is to quantify the

accuracy and precision on these retrieved properties, as well as to compare them with independent real measurements of the Aerosol Robotic Network. The ultimate goal is to achieve an affordable alternative to obtain aerosol information in places where sky radiance measurements from photometer are not available. The use of accessible and widely spread instruments like an all-sky camera can contribute to fill this lack of aerosol data.

This paper is structured as follows: Section 2 introduces the main instrumentation used in this work and the characteristics

of the place where is located, as well as the description of the inversion methodology used in this study; the sensitivity of NSR measurements to aerosol variations and the theoretical accuracy and precision of the retrieved aerosol properties with NSR measurements are discussed in Section 3. Section 4 presents a comparison of the aerosol properties retrieved using real NSR measurements as input against the aerosol properties obtained independently from AERONET. Finally, the main conclusions are summarized in Section 5.

## 2    Data and Method

### 2.1    Site and Instrumentation

#### 2.1.1    Valladolid station

The data used in this study were collected at a platform installed on the rooftop of the Science Faculty, University of Valladolid (41.6636ºN; 4.7058ºW; 705 m a.s.l.), located at Valladolid (Spain). Valladolid is an medium-sized city with a population

around 400,000 inhabitants including the metropolitan area. The climate at this place is Mediterranean, with cold winters and hot summers (Csb Köppen-Geiger climate classification). The predominant aerosol type is "clean continental", but the presence of Saharan desert dust particles is also frequent, especially in summer when the highest AOD monthly mean values are reached (Bennouna et al., 2013; Román et al., 2014; Cachorro et al., 2016).

#### 2.1.2    CE318 photometers and AERONET products

The "Grupo de Óptica Atmosférica" (Group of Atmospheric Optics) of the University of Valladolid (*GOA-UVa*) manages the mentioned instrumentation platform. This group is in charge of the calibration of part of the European AERONET photometers since 2006. The facility is now also part of the European infrastructure ACTRIS (Aerosol, Clouds and Trace Gases Research Infrastructure[3]), with active contribution to sun and moon photometry at European level (Barreto et al., 2019; González et al., 2020). A couple of calibrated AERONET reference photometers is always installed and routinely operating at Valladolid for

---

[3]www.actris.eu





side-by-side calibration of field instruments. All these photometers are CE318 (*Cimel Electronique SAS*), the standard instrument of AERONET, being the most recent model the CE318-T Sun-sky-Moon photometer (Barreto et al., 2016), also available at Valladolid station since 2016. CE318 photometers measure direct Sun irradiance (CE318-T allows also direct Moon irradiance) at several narrow spectral bands selected with interference filters mounted in a rotating wheel. These measurements are used by AERONET to calculate AOD at different wavelengths (Giles et al., 2019) with an uncertainty of $\pm 0.01$ for wave-

lengths longer than 440 nm (Holben et al., 1998). Moreover, sky radiance scans are also measured by these photometers at various wavelengths and for two geometries: almucantar (azimuth varying while zenith angle is set equal to solar zenith angle) and principal plane (zenith angle varying while azimuth is set equal to solar azimuth angle). CE318-T allows the sky radiance measurements in an additional geometry: hybrid scenario (Sinyuk et al., 2020), in which sky positions are set to fix scattering angles. AERONET uses the AOD and sky radiances (at almucantar or at hybrid scenario), at 440, 675, 870 and 1020 nm, to re-

trieve advanced aerosol properties like aerosol volume size distribution at 22 log-spaced radius bins and the complex refractive index at various wavelengths (Dubovik and King, 2000; Dubovik et al., 2006).

In this work, we use AERONET version 3 level 1.5 cloud-screened products (almucantar and hybrid) directly downloaded from AERONET webpage. These products are: AOD at 440, 500 and 675 nm (Giles et al., 2019); the three log-normal parameters of the volume size distribution (volume median radius, $R$; standard deviation, $\sigma$; and aerosol volume concentration,

$VC$) for both coarse and fine modes; real part of refractive index at 440 and 675 nm; and the fraction of spherical particles (sphericity factor) (Sinyuk et al., 2020). The AERONET retrieved products with a sky error above 10% are rejected.

### 2.1.3   All-sky camera and relative radiances

The main instrument of this work is a SONA202-NF (*Sieltec Canarias SL*) all-sky camera installed at the Valladolid *GOA-UVa* platform since July 2018. This camera consists of a CMOS sensor with a fisheye lens inside a weatherproof case and a dome

on the top. The sensor has an RGB Bayer filter plus another triband filter reducing the bandwidth of the three RGB channels. The effective wavelengths of these channels are 467, 536 and 605 nm (Antuña Sánchez et al., 2021). This camera performs every 5 minutes a fast sequence of sky images captured at different nominal exposure times: 0.3 ms, 0.4 ms, 0.6 ms, 1.2 ms, 2.4 ms, 4.8 ms and 9.6 ms. This multi-exposure sequence allows the calculation of a linear high dynamic range (HDR) image and, hence, the calculation of relative sky radiance at the three effective wavelengths for any sky direction. The process to

derive the relative sky radiances and associated uncertainty from the multi-exposure camera images, is explained in detail by Antuña Sánchez et al. (2021). This paper refers to relative radiances as uncalibrated radiances in arbitrary units but linearly proportional to the real sky radiances in absolute physical units (e.g. $Wm^{-2}sr^{-1}$).

As example for one camera sequence, Figure 1 shows in the top-left panel a tonemap of an HDR image captured on August $17^{th}$, 2019, 11:35 UTC at Valladolid. The solar zenith angle (SZA) was equal to 30º. The sky conditions were cloudless,

however, due to some reflections on the lens and the dome, the sky relative radiance cannot be extracted in the scattering angles below 10º (solar aureole) neither zenith angles between 48º to 65º (Antuña Sánchez et al., 2021); these banned areas are marked in yellow in the top-right panel of Figure 1. The chosen geometry to extract relative radiances is the AERONET hybrid geometry –rejecting the angles over the banned areas–, since this geometry allows long scattering angles even for low SZA





values and it presents a symmetry with respect to the Sun position which is useful for cloud-screening. The red points in the
top-right panel of Figure 1 represent the chosen sky points of the hybrid geometry in this case; the lowest scattering angle is
10º and there are no points in the yellow banned areas. Following the method of Antuña Sánchez et al. (2021), the relative sky
radiance is calculated at the chosen hybrid points as observed in the bottom panel of Figure 1. Once the relative radiances are
extracted, both left and right symmetrical sky points are averaged for each wavelength and the points with right-left differences
above 5% are rejected, as well as points with uncertainty above 5%, in order to warranty cloud-free conditions and high-quality
data. Afterwards, the remaining relative radiances for each wavelength are normalized, dividing each value by the total sum of
all of observations, obtaining a normalized sky radiance (*NSR*). This process is described in detail by Antuña Sánchez et al.
(2021), who characterized the uncertainty of this normalized radiances, once the explained quality and cloud-screening criteria
are applied, as: (-0.4±3.3)%, (-0.5±4.3)%, and (-0.4±5.3)% for 467, 536 and 605 nm, respectively.

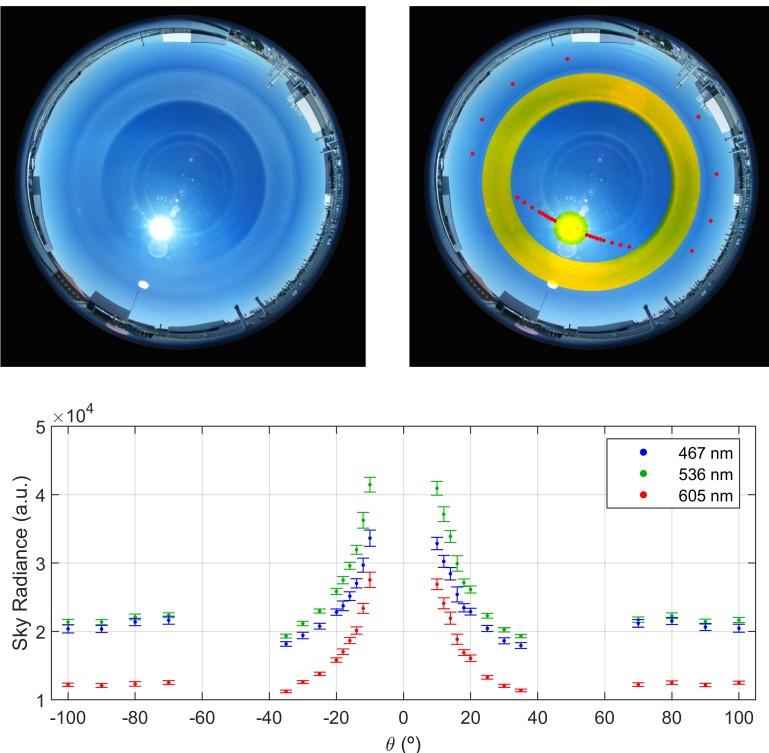

**Figure 1.** Tonemap of the high dynamic range (HDR) sky image on August $17^{th}$, 2019, 11:35 UTC at Valladolid (top panels). Top-right panel also shows in yellow the sky areas banned to extract sky radiance, and in red the points of the hybrid scan in this case. Bottom panel presents the relative sky radiance, in arbitrary units, extracted from the camera channels at 3 wavelengths as a function of the scattering angle ($\theta$), corresponding to the sky points of the hybrid scan shown in the top-right panel.





This process has been applied to all multi-exposure sequences taken by the sky camera from July 2018 to September 2020 at
Valladolid. NSR measurements in the AERONET hybrid geometry at 467, 536 and 605 nm every 5 minutes along these period,
are available for analysis.

## 2.2 Inversion strategy

The normalized sky radiances obtained at the three effective wavelengths from the sky camera, have been used as input in
the GRASP inversion code to retrieve aerosol properties. Additional information about the surface reflectance is introduced in
GRASP through Bidirectional Reflectance Distribution Function (BRDF) data. These BRDF values are obtained from a 8-days
climatology table created for Valladolid station using satellite data (MCD43C1 product from MODIS V005 Collection; Schaaf
et al. 2011) from 2000-2014 period; more details about these climatology values are provided in Román et al. (2018).

The inversion of camera radiances with GRASP has been carried out considering some assumptions: 1) the aerosol size
distribution is bimodal, with one fine and one coarse mode, and each mode equal to a log-normal distribution; 2) there is no
dependence of the real and imaginary parts of refractive index on the wavelength, since we assume the differences between
the three camera wavelengths are short to produce high variation in the complex refractive index; and 3) the imaginary part of
refractive index (*IRI*) cannot be retrieved (NSR measurements are not sensitive to aerosol light absorption) and it is assumed
equal to 0.005 for all retrievals. The IRI value of 0.005 is chosen in this work because it is the most frequent for the aerosol at
Valladolid (from AWRONET derived climatology); however, if the inversions are carried out at other location, the use of the
most representative IRI value of that location is recommended.

Eight aerosol parameters are retrieved with this strategy: 1 for the real part of refractive index (*RRI*); 1 for the fraction of
spherical particles (*SPH*); and six quantities (variables) linked to size distribution, modal radius, width and concentration (*RF*,
*RC*, $\sigma F$, $\sigma C$, *VCF* and *VCC*; the last letter corresponds to the fine, *F*, or coarse, *C*, mode). Other aerosol properties like AOD
are derived from the retrieved aerosol products. This configuration of GRASP and the products derived by it, are labelled in
this paper as "GRASP-CAM" in order to make reference to the use of camera measurements in GRASP.

The GRASP-CAM retrieval is run for each camera measurement (5 minute sampling interval) only if at least 6 cloud-free
*NSR* measurement points are available for each wavelength, in order to ensure enough information about the aerosol properties
in the measurements (at least double number of input measurements, 18, than the number of retrieved parameters, 8). In
addition, every GRASP retrieval provides, for each wavelength, the residual differences between the *NSR* measured (input) and
generated by the retrieval (modeled). This information is useful to reject non convergent retrievals; in this configuration we
classify a retrieval as non convergent if the residual in *NSR* is higher than the uncertainty of the measured *NSR* for any of the
three wavelengths.

The GRASP-CAM inversion is run in single-pixel approach. It means that each retrieval is stand-alone; the inversion of one
set of *NSR* measurements from a specific camera sequence in a given time is independent of the measurements in other times.
However, considering that the temporal variation of aerosol properties should not be abrupt, all the *NSR* measurements in a
day can be inverted together, with additional constraints about the smoothness of the temporal variation in the different aerosol
properties. This concept is called multi-pixel approach, and GRASP allows this kind of temporal (and spatial) constraints



(Dubovik et al., 2011, 2014; Lopatin et al., 2021). Hence, in order to explore the performance of temporal multi-pixel approach, the *NSR* camera measurements, that satisfy at least 6 measurements per wavelength during a full day, have been inverted

together constraining the smoothness of the time variation in the size distribution parameters, in the real part of refractive index and in the fraction of spherical particles. This configuration has been called "GRASPmp-CAM", denoting the GRASP-CAM configuration in multi-pixel (mp) approach.

## 3  Sensitivity Analysis

A detailed sensitive analysis is developed in this section using synthetic data, in order to study the capability to retrieve aerosol

properties using normalized sky radiance measurements like the ones obtained from the SONA202-NF all-sky camera. We also intend to quantify the uncertainty on the retrieved properties. To this end, and following the methodology of Torres et al. (2017), seven different aerosol models, obtained from the climatology reported by Dubovik et al. (2002), are chosen. These models correspond to the next sites: Goddard Space Flight Center (GSFC; Maryland, USA), Mexico city (MEXI, Mexico), African Savanna (ZAMB; Zambia), Solar Village (SOLV; Saudi Arabia), Bahrain (BAHR; Persian Gulf), Lanai (LANA; Hawaii, USA),

INDOEX (MALD; Maldives). GSFC and MEXI are classified as Urban/Industrial aerosol, BAHR y SOLV as desert dust, ZAMB as Biomass burning, LANA as oceanic aerosol and MALD as mixed aerosol. These seven models cover a range of aerosol types with different absorption and size distribution properties (GSFC, MEXI and ZAMB with predominance of fine mode while BAHR, SOLV and LANA with predominance of coarse mode). For each aerosol model, and using the Valladolid coordinates as reference, nine aerosol scenarios with different aerosol loads (AOD at 467 nm values ranging from 0.1 to 0.9 in

0.1 steps) have been defined.

### 3.1  Radiance simulations

Once the aerosol scenarios are defined, the radiative transfer forward module of GRASP is run for each one, to simulate the normalize radiance observations that the camera would register (hybrid scan removing banned area affected by dome reflection). These simulations have been performed for three different Sun positions (SZA equal to 30º, 50º and 70º) but this

study is focused on the SZA=70º case (the results for the other SZA angles are provided in the supplementary material), since any sky point of the hybrid scan with scattering angle higher or equal 10º, falls out of the banned area for this Sun position.

Figure 2 presents the simulated NSR values as function of the scattering angle for different $AOD_{467}$ (AOD at 467 nm) at the three camera wavelengths, and for the GSFC, ZAMB and SOLV aerosol models (Figure A1 shows the rest of the models). The changes on NSR values with AOD are appreciable when the $AOD_{467}$ values are below 0.4-0.5 for all models and wavelengths,

however, the NSR values tend to show no significant dependence on AOD as it increases above 0.5. This result is similar for all the models and for other SZA values (see Figures A1, A2 and A3) and even including lower scattering angles (Figure A4). The explanation behind this results is, that sky radiance is mainly controlled by Rayleigh scattering when the aerosol load is low, but the aerosol scattering starts to dominate the sky radiance as the AOD increases. Therefore, the sky radiance behaviour is a mixture between Rayleigh and aerosol scatterings depending on AOD, at least from very low AOD values until AOD values





about 0.5, where the sky radiance is dominated by aerosol scattering and the NSR does not present further significant changes. As example of this, the NSR differences between the cases of $AOD_{467}$ of 0.1 and 0.2 are higher for 467 nm than 605 nm since Rayleigh scattering at 605 nm is lower for longer wavelengths and hence NSR at 605 nm is more dominated by aerosol scattering than at 467 nm. Multiple scattering and surface albedo also affect NSR but their impact on NSR is small, at least for the analyzed aerosol loads.

NSR measurements are sensitive to AOD but this sensitivity decreases as the AOD increases, hence, NSR measurements could be useful to derive AOD values but until a threshold value (when Rayleigh scattering is much lower than aerosol scattering). The sky radiances are more sensitive to AOD variationswhen absolute values are measured, instead of normalized ones (see Figure A5).

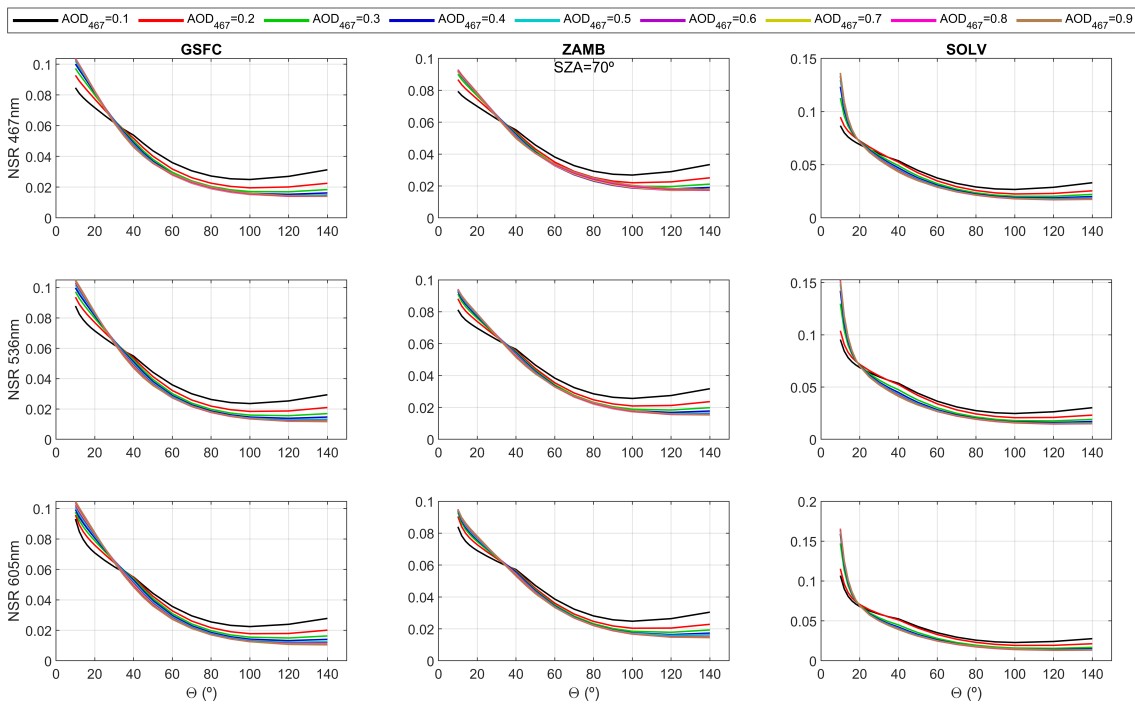

**Figure 2.** Normalized sky radiance (NSR) for solar zenith angle (SZA) of 70º at: 467 nm (top row), 536 nm (middle row) and 605 nm (bottom row), as a function of scattering angle ($\theta$) for different AOD (at 467 nm) values. Left, middle and right columns correspond to GSFC, ZAMB and SOLV aerosol models, respectively.

To observe the sensitivity of NSR observations to the aerosol type, Figure 3 shows the simulated NSR observations as
function of scattering angle for the different aerosol models in each panel, and for three different aerosol loads (the rest of aerosol loads are shown in Figure A6). In this case, the NSR values differ from one aerosol type to another, especially at low scattering angles, and this difference is appreciable for the different AOD values. Analogous result is found for other SZA values (Figures A7 and A8) and if lower scattering angles are included (Figure A9). These results point out that NSR





observations contain information about the aerosol type, independently on AOD and not only for low and moderate AOD
values, as it is found in the case of the NSR sensitivity to AOD. In the case of absolute sky radiances, the dependence on
aerosol type is even more evident for all AOD loads (see Figure A10).

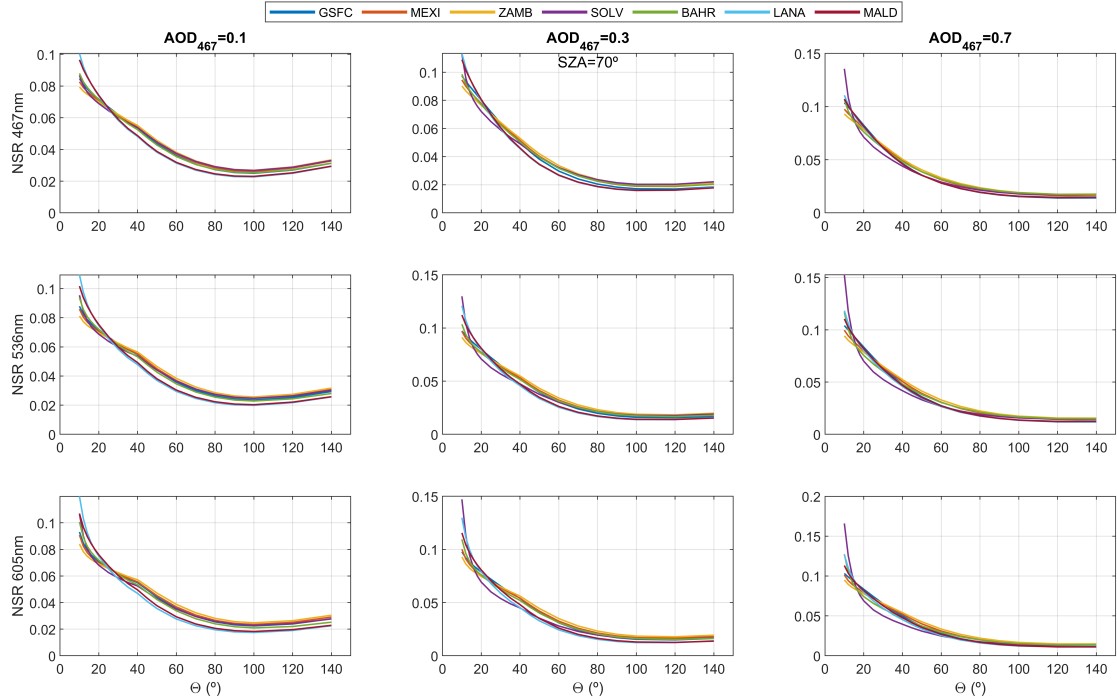

**Figure 3.** Normalized sky radiance (NSR) under solar zenith angle (SZA) of 70° at: 467 nm (top row), 536 nm (middle row) and 605 nm (bottom row), as a function of scattering angle ($\theta$) for different aerosol models. Left, middle and right columns correspond to AOD at 467 nm ($AOD_{467}$) values of 0.1, 0.3 and 0.7, respectively.

## 3.2 Aerosol properties

The results of Section 3.1 reveal that normalized sky radiances contain information about the aerosol properties; this new
section aims to know what aerosol information can be extracted from this kind of measurements and what is the uncertainty of
this information. To this end, the synthetic NSR values simulated for the different aerosol scenarios (see Section 3.1) have been
inverted following the GRASP-CAM method. These retrievals are not purely realistic since the used normalized sky radiances
are ideal (not perturbed). To obtain more realistic results, random noise has been added to each simulated sky radiance in
accordance with the NSR uncertainty of the camera product (see Section 2.1.3). Up to 1000 different sets of normalized
radiances with random noise have been created from the original radiances, and then inverted for each aerosol scenario. The



median (Md) and standard deviation (STD) of each retrieved aerosol property have been calculated, rejecting the retrievals with no convergence.

### 3.2.1   Aerosol Optical Depth

Figure 4 presents the AOD of the original (black line) chosen aerosol scenario (the one used to simulate the NSR observations), the AOD retrieved from the simulated NSR observations without noise (blue line), and the median of the retrievals with noise

(red line), with its ±standard deviation (red shadowed-area). These AOD values are plotted for the seven aerosol models and for $AOD_{467}$ values of 0.1, 0.2, 0.3 and 0.4 (same plots for higher AOD values are shown in Figure A11). The retrieved AOD values, both with and without noise, fit well with the original values at the three camera wavelengths for all aerosol scenarios and AOD (467 nm) values below 0.5-0.6 (see Figure A11). However, the AOD accuracy of the retrievals is worse for the higher aerosol loads, with low precision in the noise-perturbed retrievals. This worse fit for high AOD values must be caused by

the observed low sensitivity of NSR to AOD for high AOD values (see Section 3.1). The accuracy of AOD is not significantly higher if scattering angles from 3º instead from 10º are added to the retrieval (see Figures A12 and A13). The results are similar for SZA of 30º (see Figure A14), but the uncertainty on the retrieved AOD is much higher for SZA equal to 50º (see Figure A15), which is likely due to the lack of low scattering angles in this scenario (the Sun image appears inside the banned area of the camera).

A more quantitative analysis about the GRASP-CAM performance has been done for each retrieved aerosol parameter. To this end, and for each aerosol property, type and load, the median and standard deviation of the difference between the retrieved aerosol property (only convergent retrievals) and the original value have been calculated. Figure 5 shows, for SZA=70º and for each aerosol type and load, the number of retrievals showing convergence, and the median and standard deviation of the retrieved-original differences for AOD values at 467, 536 and 605 nm. Figure A16 shows the same plots but with x-

axis ($AOD_{467}$) limited to 0.5 for a better observation of the low-medium AOD values. The number of convergent retrievals decreases with increasing AOD. The accuracy on the retrieved AOD, represented by the median difference Md, is within ±0.02 for all wavelengths and aerosol types if $AOD_{467}$ is lower or equal than 0.4. This bias presents high values, generally negative except for GSFC and MALD, for $AOD_{467}$ above 0.6. Regarding the precision on AOD, given by STD, it decreases (STD increases) with $AOD_{467}$, the highest STD values being for SOLV. The AOD absolute precision is slightly higher for longer

wavelengths, but in general it ranges from 0.01 to 0.05 when $AOD_{467}$ varies from 0.1 to 0.5. The accuracy and precision on retrieved AOD are worse when there is a lack of information like in the mentioned cases of SZA=30º (Figure A17) and SZA=50º (Figure A18). On the other hand, if scattering angles lower than 10º (until 3º) are added to the inversion of SZA=70º (Figure A19), the precision on AOD is improved (STD is reduced), but not the accuracy.

### 3.2.2   Other aerosol properties

The retrieved aerosol volume size distributions with and without noise are shown for SZA=70º in Figure 6, in a similar way than Figure 4. Both retrievals, with and without noise, fit similarly with the original size distribution. In general, the accuracy of the retrieved size distributions decreases for high AOD values (see Figure A20), which is likely related to the mentioned lack



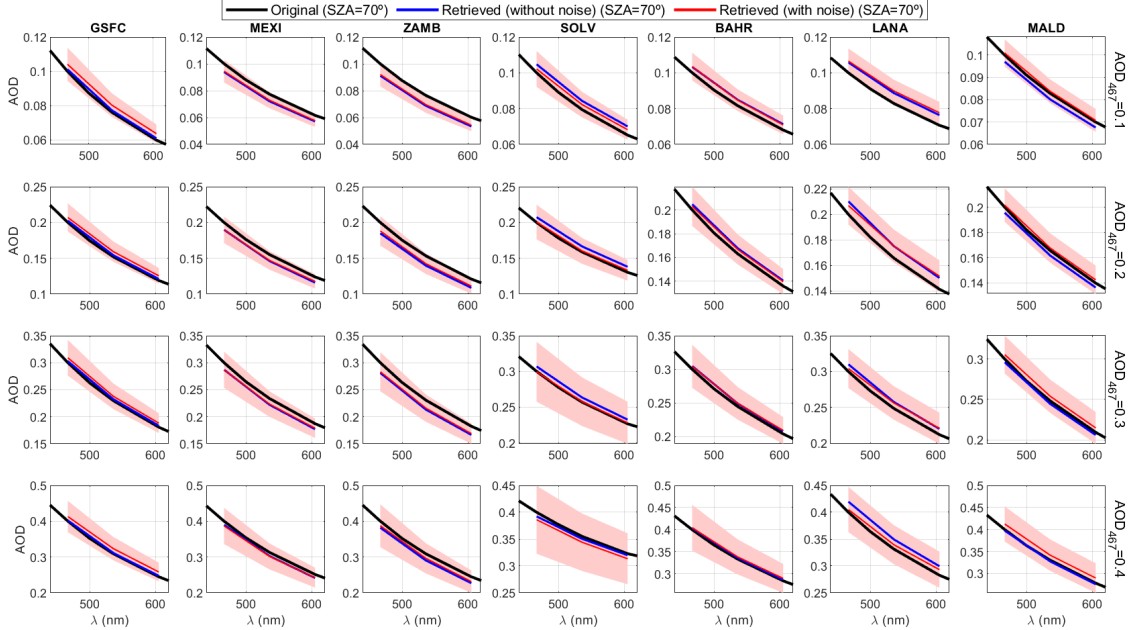

**Figure 4.** Reference (black line), retrieved without noise (blue line), and median of all retrieved with noise (red line) aerosol optical depth (AOD) under solar zenith angle (SZA) equal to 70º. These AOD values are represented for different aerosol types (one type per column) and for AOD at 467nm ($AOD_{467}$) values of 0.1 (first row), 0.2 (second row), 0.3 (third row) and 0.4 (last row). Red shadowed-area corresponds to ± the standard deviation of all averaged size distributions retrieved with noise-perturbed radiances.

of sensitivity on NSR observations to AOD, for high aerosol loads. The retrieved size distributions show higher accuracy for fine mode than for coarse mode (especially for LANA, SOLV and BAHR); higher standard deviation is found for coarse mode,

i.e. less precision. The lower accuracy for the coarse mode could be caused by the lack of low scattering angles, which contain information about the coarse mode (Tonna et al., 1995; Román et al., 2017; Torres and Fuertes, 2020); this is corroborated by Figures A21 and A22, where the accuracy on the coarse mode is better for the same retrievals if we include scattering angles from 3º instead of 10º. All these results are similar for SZA=30º (see Figure A23), but the uncertainty in the coarse mode is much higher for the case of SZA=50º (see Figure A24); it could be caused by the lack of low scattering angles under SZA=50º

due to the banned area of the camera.

Figure 7 presents a similar analysis than Figure 5 but for the other aerosol properties retrieved by GRASP-CAM. Regarding size distribution parameters, Figure 7 shows that the accuracy and precision on fine mode radius are within ±0.01 $\mu$m and below 0.02 $\mu$m, respectively, even for high aerosol loads; while, for y¡the coarse mode, the accuracy and precision of the radius is within ±0.6 $\mu$m and below 0.2 $\mu$m, respectively, for most of the cases. The accuracy on $\sigma$ is worse for the coarse mode (within

±0.1 $\mu m^3/\mu m^2$) than for the fine one (within ±0.02$\mu m^3/\mu m^2$), but the precision is similar and below 0.08 $\mu m^3/\mu m^2$ in most cases. Regarding the volume concentration, the accuracy of the fine and coarse mode is, for $AOD_{467}$ below 0.6, within ±0.01





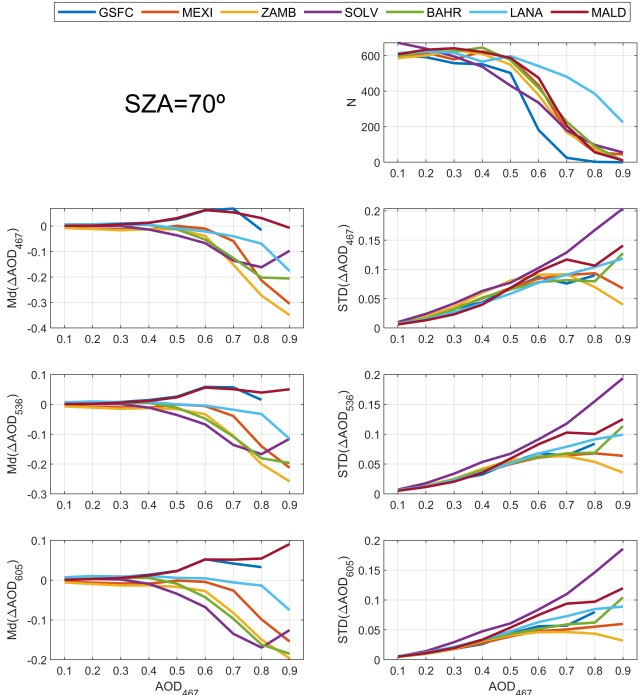

**Figure 5.** Median (Md) and standard deviation (STD) of the $\Delta$ differences between the available retrieved aerosol properties with noise-perturbed radiances, and the original (reference) properties. The amount of available retrievals (N) is also shown. Only the retrievals with solar zenith angle (SZA) equal to 70º are used. The aerosol properties provided are: aerosol optical depth (AOD) at 467 nm ($AOD_{467}$), 536 nm ($AOD_{536}$) and 605 nm ($AOD_{605}$). The Md and STD are represented as a function of ($AOD_{467}$) for different aerosol types.

$\mu m^3/\mu m^2$ and $\pm 0.04 \ \mu m^3/\mu m^2$, respectively; while the precision is below $0.04 \mu m^3/\mu m^2$ and $0.1 \mu m^3/\mu m^2$ for both fine and coarse modes, respectively. These precision and accuracy get worse as AOD increases. The results for SZA=30º (Figure A25) are similar than for SZA=70º regarding size distribution parameters, but the accuracy on these parameters is generally

worse for the SZA=50º case (Figure A26). In general, the retrieved coarse mode parameters show higher dependence on the aerosol type than fine mode ones. This dependence is lower when low scattering angles (from 3º instead of 10º) are used in the inversion (Figure A27); the addition of low scattering angles also improves the accuracy of the retrieved coarse parameters. However, RC and $\sigma$C still present a significant dependence on aerosol type, which could indicate that the retrieved values of these parameters do not vary significantly from the initial guess value and, hence, the proposed GRASP-CAM methodology

has not enough sensitivity to retrieve both parameters.

Figure 7 also presents the Md and STD values for the differences on RRI, showing an accuracy on RRI between -0.02 and 0.04 for AOD values below 0.5, with a precision about 0.05; the STD values are in general higher for MEXI and ZAMB than for the other aerosol types. In addition, as supplementary material, Figure A28 presents the same results than Figures 6 and





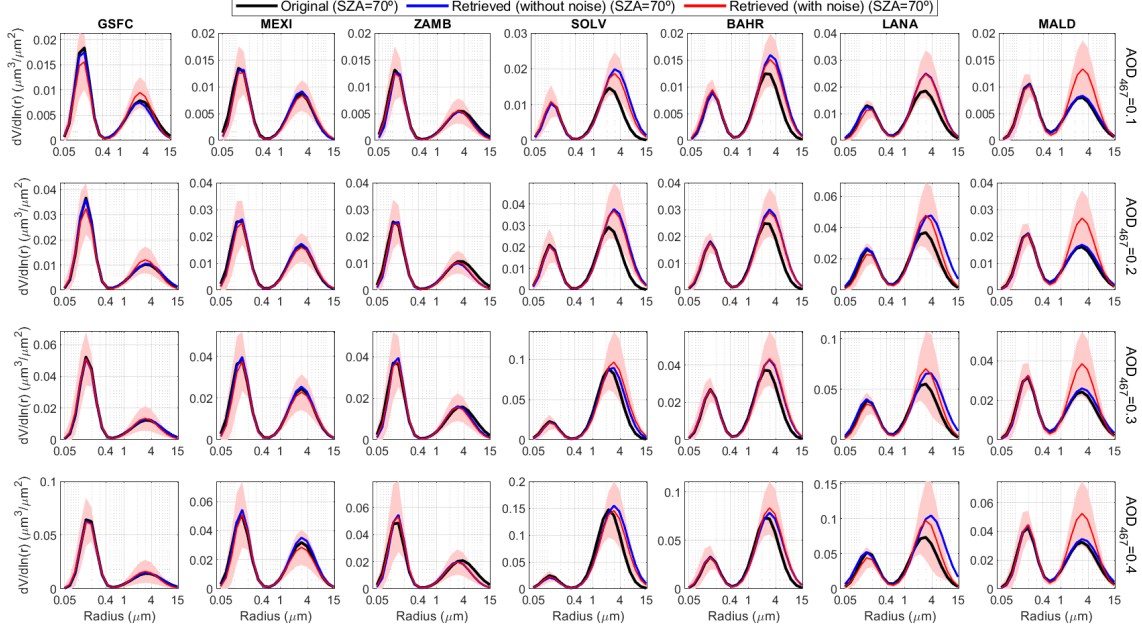

**Figure 6.** Original (black line), retrieved without noise (blue line), and median of all retrieved with noise (red line) aerosol volume size distributions under solar zenith angle (SZA) equal to 70°. These size distributions are represented for different aerosol types (one type per column) and for AOD (aerosol optical depth) at 467nm ($AOD_{467}$) values of 0.1 (first row), 0.2 (second row), 0.3 (third row) and 0.4 (last row). Red shadowed-area corresponds to $\pm$ the standard deviation of all averaged size distributions retrieved with noise-perturbed synthetic radiances.

4 but for RRI. The retrieved values correlate with the original ones, especially for the retrievals without noise. These results

280  indicate that NSR measurements contain information about the real part of refractive index.

The Md values of $\Delta$SPH are also shown in Figure 7, being close to zero for the entire AOD range, except for desert dust aerosols. The STD of $\Delta$SPH is similar for all aerosol types, varying from 30% to about 20% when AOD rises from 0.1 to 0.9. In general, the accuracy and precision of all parameters observed in Figure 7 are worse when the scattering angles are reduced, like in the cases of SZA=30° (Figure A25) and especially SZA=50° (Figure A26). If scattering angles from 3° instead of 10°

285  are used, the precision on the parameters of Figure 7 is slightly better in most cases (Figure A27).

A similar analysis than the one presented in this section, was done but assuming that IRI can be also retrieved from the NSR measurements; we assumed an initial guess IRI value of 0.01. The retrieved IRI values are shown in Figure A29. There is not correlation between the original and retrieved values (even without noise perturbation). These results point out that, as expected, there is no sensitivity of NSR measurements to the imaginary part of refractive index, which motivated the exclusion

290  of IRI from the set of parameters to be retrieved by the GRASP-CAM strategy (see Section 2.2).





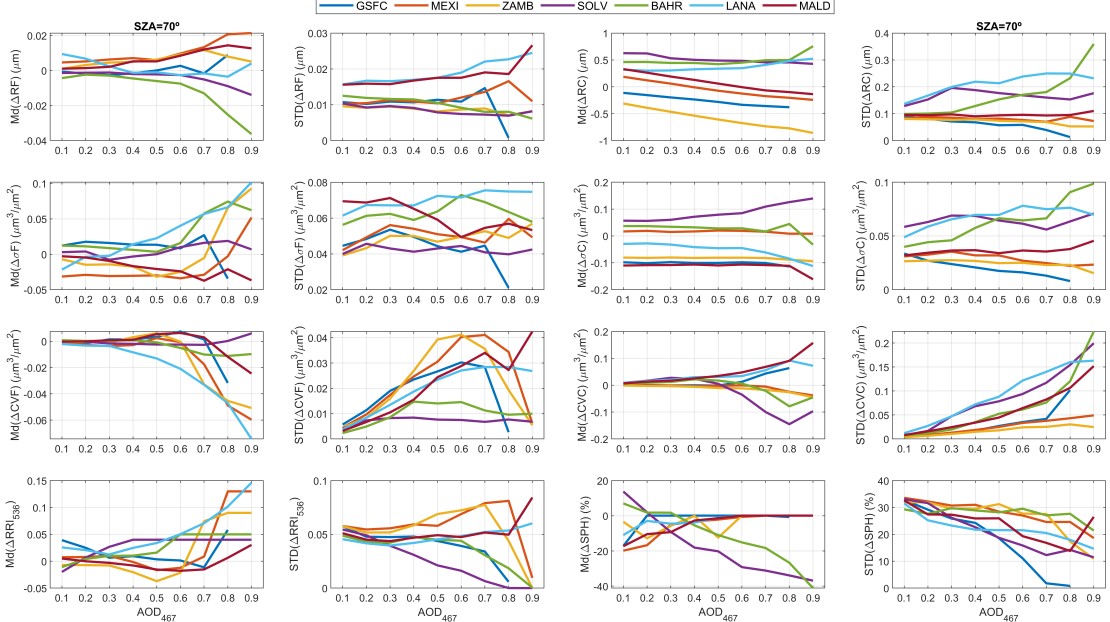

**Figure 7.** Median (Md) and standard deviation (STD) of the Δ differences between the available retrieved aerosol properties with noise and the original ones for solar zenith angle (SZA) of 70º. The aerosol properties are: volume median radius of fine (RF) and coarse (RC) mode; standard deviation of log-normal distribution for fine ($\sigma$F) and coarse ($\sigma$C) mode; aerosol volume concentration for fine (VCF) and coarse (VCC) mode; real part of refractive index at 536 nm ($RRI_{536}$); and the fraction of spherical particles (SPH). These Md and STD values are represented as a function of ($AOD_{467}$) for different aerosol types.

## 4 Results

### 4.1 Single-pixel approach

In order to study the performance of the GRASP-CAM products using real measurements, the GRASP-CAM method has been applied to NSR measurements obtained with the all sky camera at Valladolid (see Section 2.1). The data span from $11^{st}$ July 2018 to $15^{th}$ September 2020. This has provided a total of 42105 retrievals (satisfying at least 6 cloud-free NSR data per wavelength), but only 34536 pass the convergence criteria.

An additional quality-control criterion has been added: the need of at least one NSR measurement, in any wavelength, with a low scattering angle ($\leq$14º) and with a high one (>80º). This new criterion is based on the results of Section 3, which revealed the need of low scattering angles for an accurate retrieval, but also based on the AERONET criteria (Holben et al., 2006), which demand a minimum number of sky radiance measurements in various scattering angle ranges, including high angles, for quality-assured retrievals. This new criterion reduces the available retrievals to 23368, removing all retrievals with SZA values between 47.2º and 64.2º and with SZA values higher than 82.5º.





These criteria are required for quality-assurance of the retrieval. However, some data contaminated by clouds could pass them. Hence, additional cloud-screening criteria, based on the AERONET cloud-screening version 3 (Giles et al., 2019), have been applied. For this purpose, the time series of the GRASP-CAM retrieved AOD has been used. A GRASP-CAM retrieval has been assumed as cloud contaminated, and hence removed, if: 1) the time variation on the remaining cloud-free AOD at 536 nm is higher than $0.01min^{-1}$ (temporal smoothness criterion); 2) there is no remaining cloud-free AOD within ±1 hour (stand-alone criterion); 3) the remaining cloud-free AOD at 536 nm is without the range defined by the daily mean of this variable ±3sigma, being sigma the daily standard deviation of this AOD (AOD 3sigma criterion); 4) the remaining cloud-free Angstrom Exponent (AE) is without the range defined by the daily mean of this variable ±3sigma, being sigma the daily standard deviation of this AE (AE 3sigma criterion); 5) the number of remaining cloud-free data in one day is lower than 3 or than the 10% of the potential retrievals on this day (potential measurements criterion). These criteria are described in detail in Giles et al. (2019) and González et al. (2020). Finally, a total of 22501 GRASP-CAM retrievals have been classified as cloud-free and quality-assured after applying these cloud-screening criteria. These remaining data are the measurements used in this section.

### 4.1.1 Aerosol Optical Depth

To study the goodness of the AOD retrieved by GRASP-CAM, it needs to be compared with alternative and independent measurements; in our case with the AOD measured by a collocated AERONET Sun-sky photometer. To this end, the AOD from AERONET has been interpolated to the effective camera wavelengths following the Ångström law (Angström, 1930, 1961). Figure 8 shows both GRASP-CAM and AERONET AOD time series for a 12-day period in summer 2020 at Valladolid. AOD from GRASP-CAM looks a bit noisier than AERONET; however, both data series are well correlated for the three wavelengths, showing similar AOD values. For example, a decrease from moderate to low AOD load can be observed in both series at $1^{st}$ August. Sometimes AOD from GRASP-CAM is available when AERONET one is not, which points out that GRASP-CAM could be useful to complement AOD data series. The full time series of retrieved AOD can be observed in Figure A30.

A more quantitative analysis has been done by a match-up of GRASP-CAM and AERONET AOD data. Each available GRASP-CAM AOD data has been paired up with the closest AOD AERONET data within ±2.5 min. The ±2.5 min interval has been used because GRASP-CAM data is available each 5 minutes and this interval avoids overlapping one AERONET value for two or more GRASP-CAM retrievals. After this match-up, a total of 16935 AOD data pairs (GRASP-CAM vs AERONET) are available for each wavelength. Upper panels of Figure 9 show these data pairs through density scatter-plots of AOD from GRASP-CAM as function of AERONET AOD for each wavelength. The least squares linear fit, its equation and the determination coefficient ($r^2$), of these data pairs are also included. AOD from GRASP-CAM correlates well with AERONET measurements, with $r^2$ about 0.87 at the three wavelengths. The differences between both AOD sources increase for high AOD values, but the availability of data pairs under these conditions is scarce. The scatter-plots and the linear fit equations show an slightly overestimation of GRASP-CAM to AERONET for low AOD values, while GRASP-CAM tends to underestimate the highest AOD values.





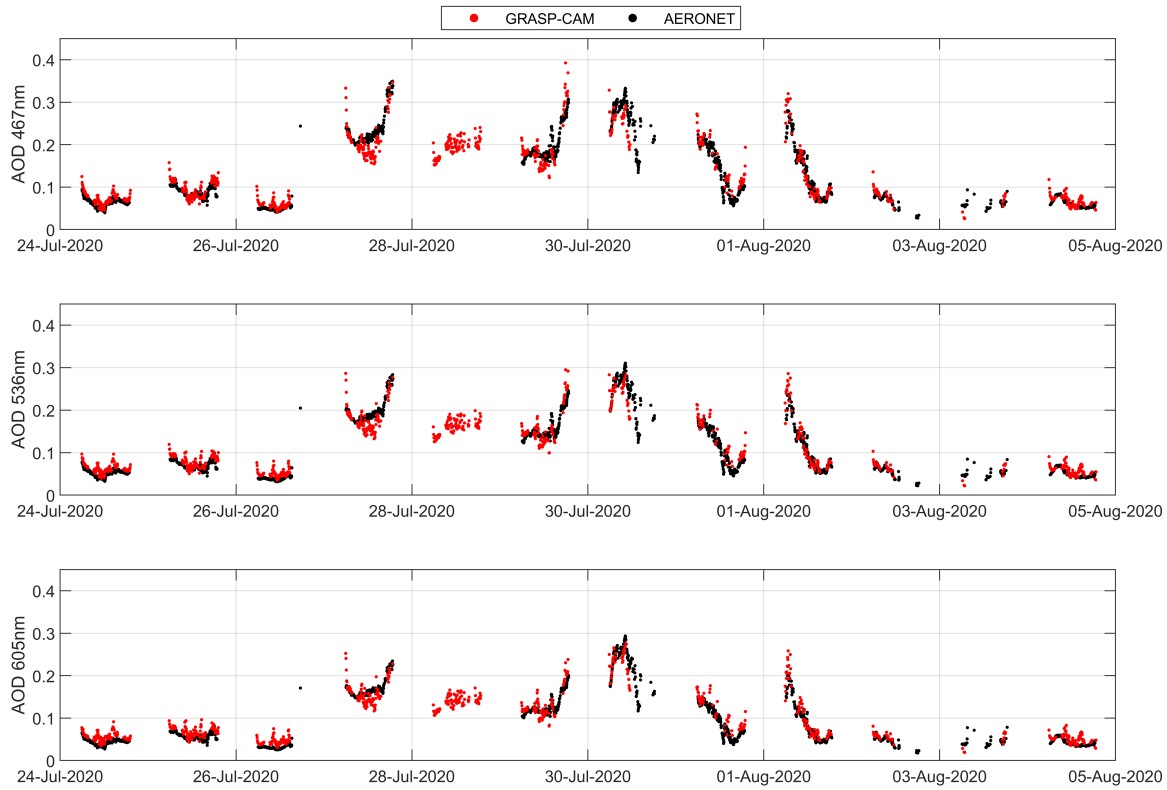

**Figure 8.** Aerosol optical depth (AOD) at 467 nm (upper panel), 536 nm (middle panel) and 605 nm (bottom panel) retrieved by GRASP under single-pixel approach (GRASP-CAM) and by AERONET at Valladolid from $24^{th}$ July 2020 to $4^{th}$ August 2020. AERONET data have been interpolated to the all-sky camera wavelengths.

The bottom panels of Figure 9 present the frequency distribution of the differences between the AOD from GRASP-CAM and AERONET, and the mean (M), median, and standard deviation of these differences. The peak of the three distributions are slightly biased to positive values, which could be caused by the mentioned overestimation of GRASP-CAM to AERONET for low AOD values (the most frequent in Valladolid). On the other hand, a longer tail appears on the distributions for negative

values, likely indicating the mentioned underestimation of GRASP-CAM to AERONET for the highest AOD values, which are less frequent. Considering all data, the median of the differences, which can be assumed as the accuracy considering AERONET as reference, ranges between 0.006 at 467 nm and 0.010 at 605 nm. The mean values are slightly lower. The uncertainty of AERONET AOD is ±0.01 for their nominal wavelengths.Hence, the bias on AOD between GRASP-CAM and AERONET is within the AERONET uncertainty. Using the AERONET as reference, the precision on AOD, associated with





standard deviation of the distributions of Figure 9, goes from 0.030 at 467 nm to 0.024 at 605 nm, these values being larger

than the AERONET AOD uncertainty.

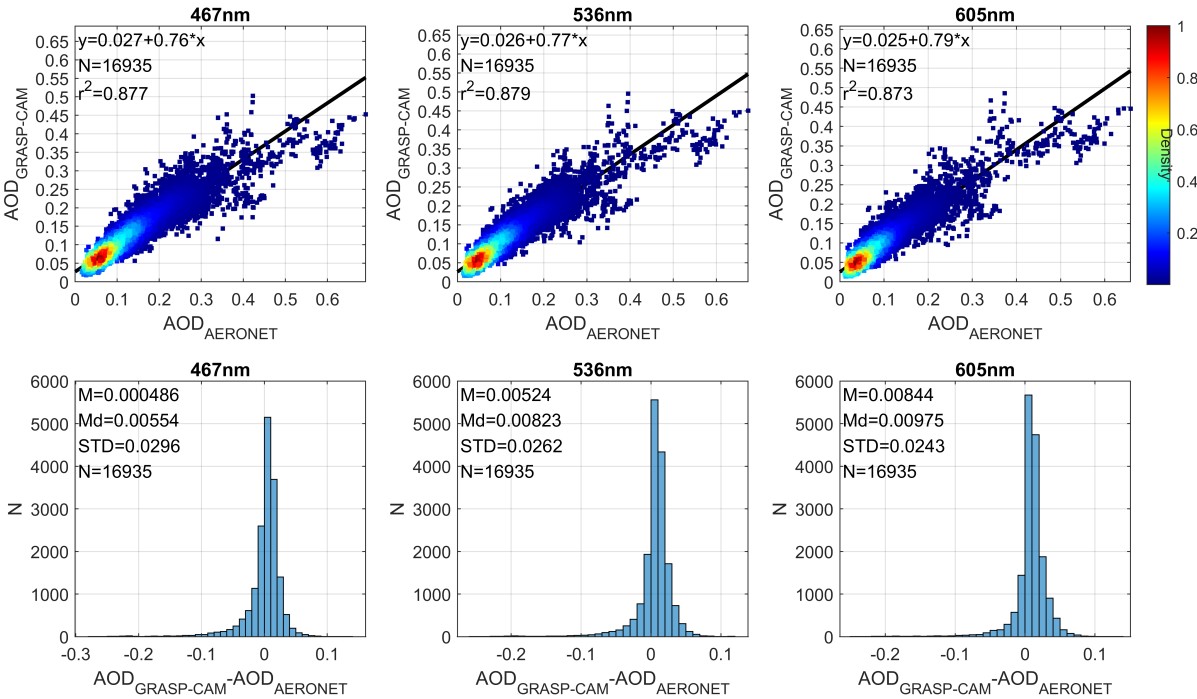

**Figure 9.** Upper panels: density scatter-plots of the aerosol optical depth (AOD) retrieved by GRASP in single-pixel approach (GRASP-CAM) versus the AOD from AERONET at 467 nm (left panel), 536 nm (middle panel) and 605 nm (right panel); linear fit (black line), its equation and the determination coefficient ($r^2$) are also shown. Bottom panels: Frequency histograms of the differences in AOD between GRASP-CAM and AERONET at 467 nm (left panel), 536 nm (middle panel) and 605 nm (right panel); The mean (M), median (Md) and standard deviation (STD) of these differences are also shown.

The last values of the experimental accuracy and precision on GRASP-CAM AOD have been globally obtained using all available data, but the results could vary as a function of AOD, as suggested in Section 3.2. In order to obtain a more detailed description of the accuracy and precision, the differences between AOD from GRASP-CAM and AERONET (ΔAOD) has been

calculated for different AOD bins (±0.025 AOD bins). The number of available data, the median and the standard deviation of all these distributions are represented on Figure 10 as function of AOD. The amount of data available per AOD bin is much higher for AOD values below 0.25 (N>1000), as it can also be observed in Figure 9; hence, the results will be more representative for the bins of lower AOD. The median of the ΔAOD differences decreases with AOD and it is within ±0.015 and with similar values for all wavelengths for AOD values below 0.25. The STD values are below 0.02 for AOD lower than





0.15 and below 0.04 for AOD lower than 0.25. These Md and STD results are similar to the observed with synthetic data in Section 3.2 (see Figures 5, A17 and A18).

AOD from GRASP-CAM underestimates the AERONET product for AOD values above 0.25, decreasing the Md values from -0.05 (AOD ≈0.3) until to -0.2 (AOD ≈0.6). This underestimation and lack of accuracy of the GRASP-CAM AOD for high AOD values, have been also observed in the synthetic data analysis (Section 3.2), especially for the case of SZA=30º and

for desert dust coarse particles, which are usually the predominant particles at Valladolid during high AOD episodes. The STD for AOD values above 0.25 still increases up to 0.06-0.07 when AOD is about 0.375, but then it decreases to 0.01-0.02 as the AOD goes up to 0.7. These results are approximately between the ones obtained theoretically with SZA=70º and SZA=30º. However, the obtained accuracy and precision on AOD from GRASP-CAM using AERONET as reference must be carefully considered for high AOD values, since the amount of available data under these conditions may be not representative.

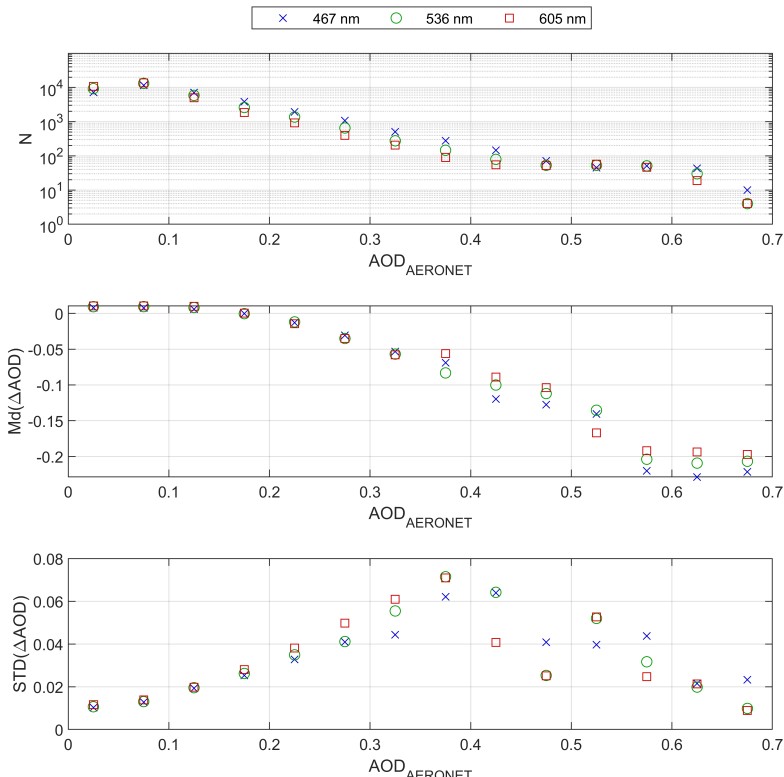

**Figure 10.** Median (Md; middle panel) and standard deviation (STD; bottom panel) of the Δ differences on the aerosol optical depth (AOD) retrieved by GRASP under single-pixel approach (GRASP-CAM) and the obtained from AERONET at 467, 536 and 605 nm for different AOD bins. The available number of ΔAOD data (N) per AOD bin is also shown in upper panel for the three wavelengths.





The accuracy and precision on GRASP-CAM AOD depends on the aerosol load, but they could also depend on the SZA and the availability of input data as pointed out in Section 3. Figure 11 shows the AOD differences between GRASP-CAM and AERONET for the three wavelengths as a function of SZA. The results are similar for the three wavelengths. Most of differences are around zero for SZA values below 47.2º; however, for higher SZA values the centre of the differences increases from about 0 to about 0.01 (overestimation). A similar overestimation is observed in the middle of the 30º-40º SZA range.

Regarding data outliers, they are most frequent for low SZA values, reaching negative values (underestimation). The lack of data between 47.2º and 64.2º is caused by the requirement of at least one NSR measurement with scattering angle lower or equal than 14º in the input. Figure A31 shows the same differences of Figure 11 but including retrievals without scattering angles ≤14º. It must be taken into account that the addition of these retrievals modified the number of retrievals passing the cloud-screening criteria and, hence, some data appearing in Figure 11 do not appear in Figure A31 and vice-versa. The

most important result is that the AOD from GRASP-CAM clearly overestimates AERONET in the 47.2º- 64.2º SZA range (where there is a lack of scattering angles ≤14º); it confirms the need of rejecting the retrievals without, at least, one NSR measurements with scattering angle ≤14º.

### 4.1.2   Other aerosol properties

The previous subsection is focused on the AOD performance from the GRASP-CAM retrievals, but there are more aerosol

properties of interest in these retrievals. Figure 12 shows the time series of GRASP-CAM and AERONET retrieved size distribution parameters of the fine and coarse mode, for the same period shown in Figure 8. It is important to remark that these other aerosol properties retrieved by AERONET are obtained inverting AOD plus sky radiance measurements, and these sky measurements are less frequent (and need more time to be measured) than the AOD ones. Thus, the amount of data retrieved by GRASP-CAM is higher than AERONET, GRASP-CAM values being available usually each 5 minutes. In general, the

size distribution parameters retrieved by GRASP-CAM present a noisier behaviour than AERONET values, especially for the coarse mode. GRASP-CAM and AERONET values look well correlated and with similar values for VCF, VCC and RF parameters, but not for the rest.

The mentioned correlation between GRASP-CAM and AERONET size distribution parameters can be better observed in Figure 12, which also presents the total aerosol volume concentration (VCT). This figure shows the density scatter-plots of the

GRASP-CAM values as a function of the AERONET ones. The GRASP-CAM – AERONET data pairs have been matched up by pairing the AERONET retrievals with the closest GRASP-CAM data within ±2.5 min. In addition, AERONET retrievals with AOD at 440 nm below 0.05 have been discarded in order to have a minimum of aerosol load for the retrieval. A total of 1853 data pairs have been obtained. The highest correlations between GRASP-CAM and AERONET appear for VCT and VCC, both with $r^2$ above 0.5, and for VCF and RF, with $r^2$ of 0.38 and 0.35, respectively. The largest lack of correlation is

on the $\sigma$C parameter, where GRASP-CAM frequently provides the value of 0.9 $\mu m^3/\mu m^2$ while AERONET provides more variation of values, with 0.7 $\mu m^3/\mu m^2$ being the most frequent. Likely, this worse performance of the coarse mode retrieval could be partially related with the lack of scattering angles below 10º in the GRASP-CAM input, since it has been observed





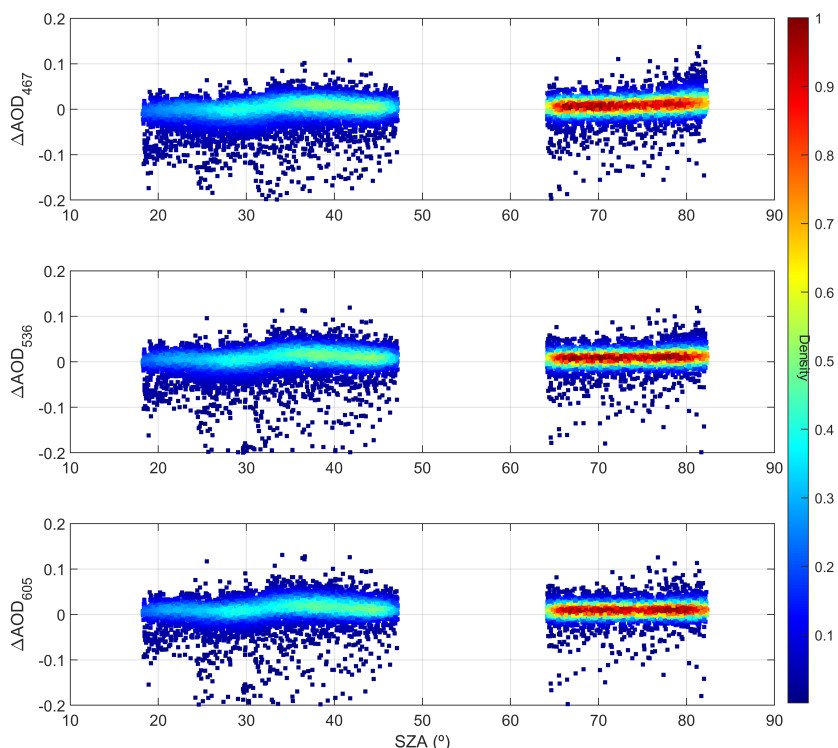

**Figure 11.** $\Delta$ differences on the aerosol optical depth (AOD) retrieved by GRASP under single-pixel approach (GRASP-CAM) and the obtained from AERONET at 467 nm (upper panel), 536 nm (middle panel) and 605 nm (bottom panel) as function of the solar zenith angle (SZA). Colour legend represents the density of data points.

that these lowest angles contain more information about the coarse mode microphysics. A low sensitivity of GRASP-CAM to retrieve RC and $\sigma$C, the parameters with the lowest correlation in Figure 13, was also observed in Section 3.2.2.

The lack of correlation does not mean that GRASP-CAM products are not accurate. To estimate the accuracy and precision on these GRASP-CAM parameters, the frequency distributions of the differences between GRASP-CAM and AERONET on the aerosol parameters of Figure 13, have been calculated and shown in Figure 14, in addition to their statistical estimators. Regarding volume median radius, an intensive aerosol property, both fine and coarse modes present a symmetric distribution with a Gaussian behaviour, being the median $\pm$STD equal to -0.02$\pm$0.04 $\mu$m and -0.3$\pm$0.7 $\mu$m for the fine and coarse modes,

respectively. GRASP-CAM slightly underestimates the aerosol radius for both fine and coarse mode, the precision being better for the fine mode. Regarding $\sigma$F and $\sigma$C, both GRASP-CAM intensive properties present similar results, with an overestimation about 0.07-0.10 $\mu m^3/\mu m^2$ and a precision given by the STD of 0.12-0.13 $\mu m^3/\mu m^2$. The size distribution of VCF and VCC from GRASP-CAM are similar, both centred around zero, but with a long negative and positive tail for fine and coarse mode,

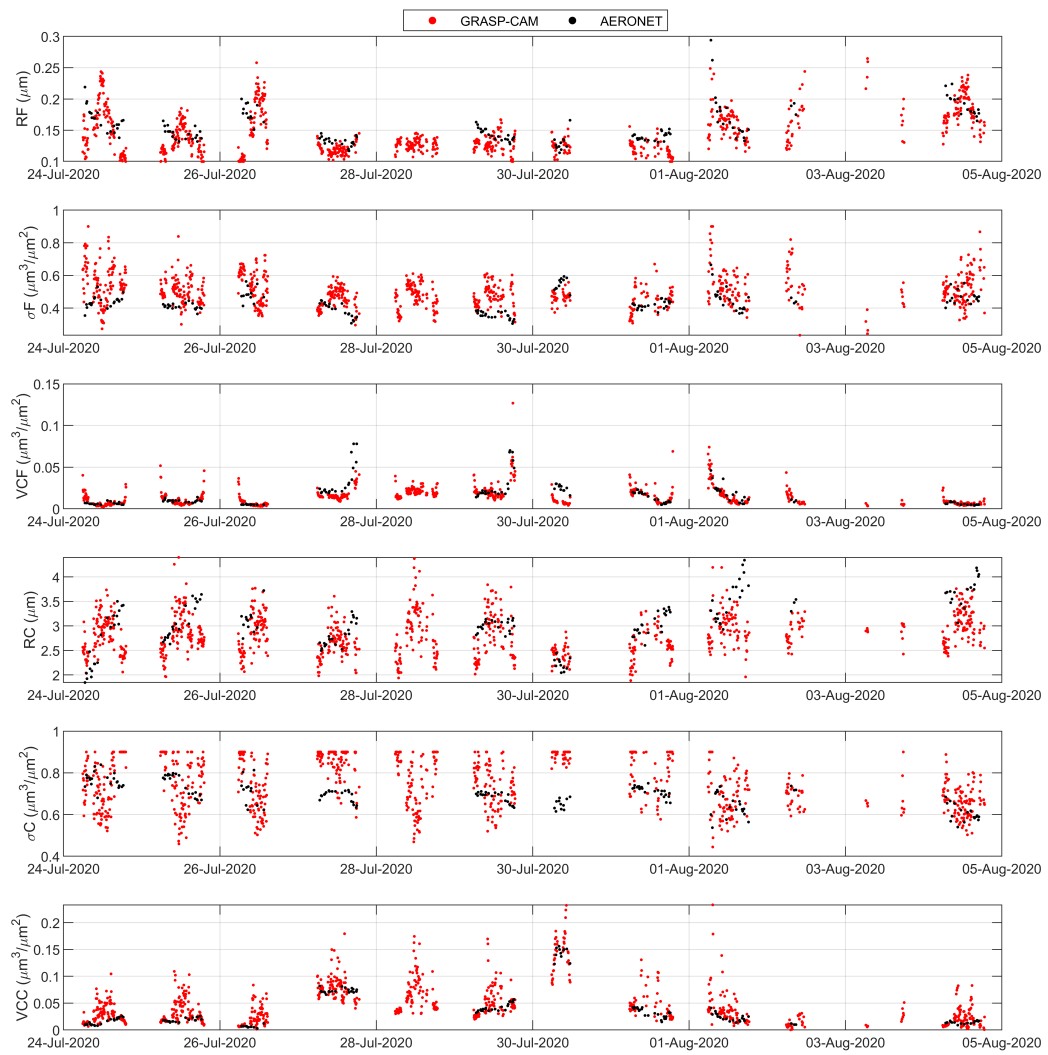

**Figure 12.** Volume median radius, *R*, standard deviation of log-normal distribution, $\sigma$ and aerosol volume concentration, *VC* for fine (F; upper panels) and coarse (C; bottom panels) modes of the aerosol size distribution retrieved by GRASP under single-pixel approach (GRASP-CAM) and by AERONET at Valladolid from $24^{th}$ July 2020 to $4^{th}$ August 2020.



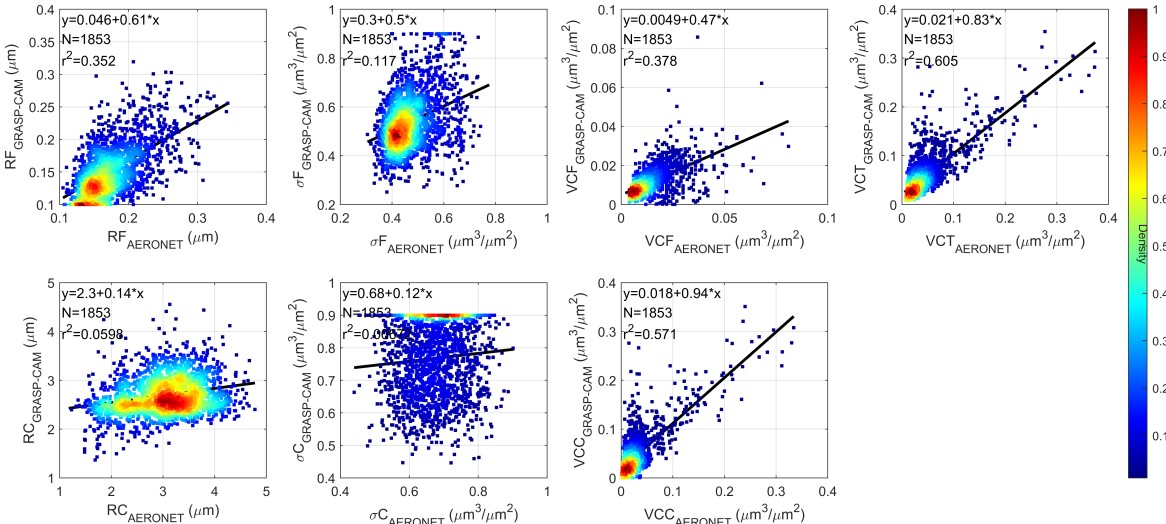

**Figure 13.** Density scatter-plots of the aerosol size distribution properties retrieved by GRASP in single-pixel approach (GRASP-CAM) versus the ones retrieved by AERONET; linear fit (black line), its equation and the determination coefficient ($r^2$) are also shown. These size distribution properties are: volume median radius of fine (RF) and coarse (RC) modes; standard deviation of log-normal distribution for fine ($\sigma$F) and coarse ($\sigma$C) modes; and aerosol volume concentration for fine (VCF) and coarse (VCC) modes and the total value (VCT).

respectively. It means that GRASP-CAM overestimates/underestimates the coarse/fine volume concentration for high aerosol

concentrations, in agreement with Figure 13. The precision on these parameters is about 0.007 $\mu m^3/\mu m^2$ and 0.03 $\mu m^3/\mu m^2$ for fine and coarse mode, respectively; the results for the total concentration are similar to those obtained for VCC. In general, the accuracy and precision obtained in the size distribution parameters from GRASP-CAM compared with AERONET products are worse than ones the obtained by comparison with synthetic data (Section 3.2, see also Figure A17 and A18), except for the aerosol volume concentration. This can be caused, at least in part, by the uncertainty of the AERONET retrievals used as

reference.

The rest of retrieved parameters, RRI and SPH from GRASP-CAM and AERONET for the analyzed period of Figure 8, are provided as supplementary material in Figure A32. These parameters retrieved by GRASP-CAM look noisy compared with AERONET ones. It points out that GRASP-CAM could be not too sensitive to these parameters and likely they should not be used for studies requiring more accuracy. Anyway, these last results must be taken with care since the quality of some of these

AERONET products is only assured for high AOD values.

### 4.2  Multi-pixel approach

The GRASP-CAM method is based on stand-alone retrievals and occasionally the input data do not contain enough information on the aerosol, especially in a range of Sun positions due to technical problems of the all-sky camera (dome reflections). One





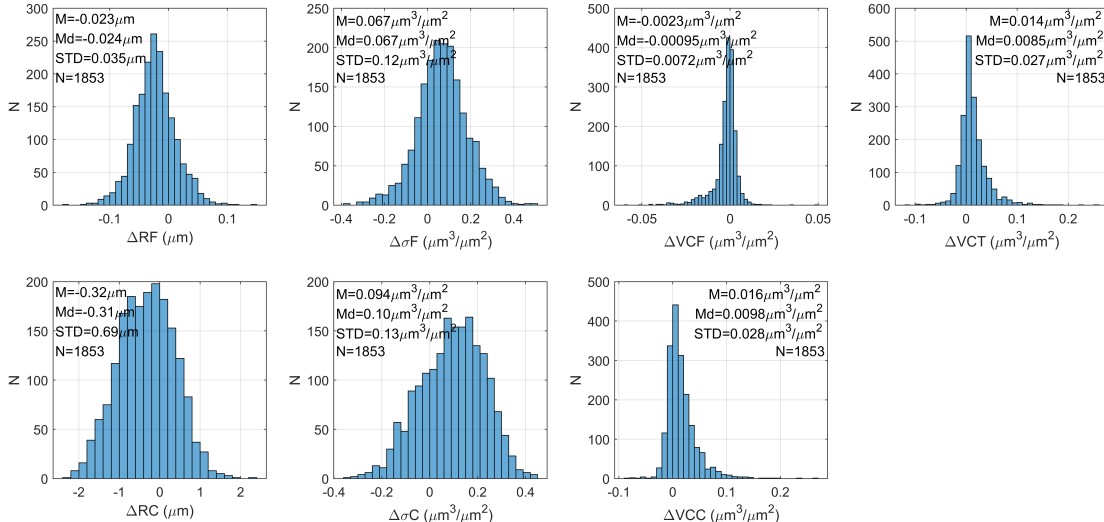

**Figure 14.** Frequency histograms of the Δ differences in the aerosol size distribution properties retrieved by GRASP in single-pixel approach (GRASP-CAM) and the ones retrieved by AERONET; The mean (M), median (Md) and standard deviation (STD) of these differences are also shown. These size distribution properties are: volume median radius of fine (RF) and coarse (RC) modes; standard deviation of log-normal distribution for fine ($\sigma$F) and coarse ($\sigma$C) modes; and aerosol volume concentration for fine (VCF) and coarse (VCC) modes and the total value (VCT).

way to add more information at each single retrieval, is to use information from the temporal adjacent measurements or
retrievals, which is the idea behind the temporal multi-pixel approach (GRASPmp-CAM; see Section 2.2). Information from the measurements taken at one time is transferred to other times under this approach; in addition, it provides more stability to all the retrievals of one day since they are linked so that the retrieved aerosol properties cannot vary abruptly between consecutive retrievals. This section is focused, same as Section 4.1, on the analysis of the performance of the aerosol properties derived by NSR camera measurements but retrieved by GRASPmp-CAM instead of GRASP-CAM.

The quality-assurance criteria for input measurements for GRASPmp-CAM are slightly different than those for GRASP-CAM. A GRASPmp-CAM retrieval is done with all NSR sequences of one day, each one with at least 6 cloud-free NSR measurements per wavelength. Therefore, for GRASPmp-CAM, it is only demandeded that one of the available NSR sequences of a full day must contain one NSR measurement under scattering angle ≤14º and another one above 80º. These criteria are demanded for all single NSR sequences with GRASP-CAM since every NSR sequence is inverted alone; but in GRASPmp-
CAM the information of the measurements with scattering angle ≤14º or >80º in one NSR sequence is transferred to the other sequences within the same day. Moreover, it allows the retrieval under SZA values between 47.2º and 64.2º. The NSR sequences taken under SZA>82.5º have not been added in the daily GRASPmp-CAM retrieval, since the Sun is close to horizon and these angles are not contemplated in the GRASP-CAM.

Once the input measurements are ready for each day, GRASP has been run under GRASPmp-CAM configuration, which
provides as result the aerosol properties (AOD, size distribution, etc.) for each NSR sequence used in the input, i.e., for each
measurement time. A total of 35615 time inversions are available from $11^{st}$ July 2018 to $15^{th}$ September 2020 at Valladolid.
Each retrieval also provides the individual residual at each time (or used NSR sequence) between the input measurements and
the observations reproduced by the retrieved aerosol properties. These residual values are also used to remove individual non-
convergent retrievals, applying the same criteria than in the GRASP-CAM method (see Section 4.1). Up to 32621 inversions
pass these convergence criteria. Finally, the same cloud-screening criteria of Section 4.1, based on AOD, have been applied to
remove cloud contaminated data. The final amount of individual cloud-free aerosol retrievals obtained by GRASPmp-CAM is
32062. Cloud-screening criteria remove less data for GRASPmp-CAM than for GRASP-CAM, because adjacent retrievals, and
hence aerosol properties, of GRASPmp-CAM are constrained to a smooth temporal variation; therefore some cloud-screening
criteria, like the one based on time variation of AOD, are not as frequently triggered.

### 4.2.1 Aerosol Optical Depth

The AOD time series retrieved by GRASPmp-CAM at Valladolid is shown in Figure 15. The time period is the same than
in Figure 8 (the full AOD time series is provided in Figure A33). The behaviour of AOD is similar than the obtained by
GRASP-CAM but less noisy, therefore the GRASPmp-CAM AOD values are closer to the AERONET ones.

The AOD values, and the rest of aerosol properties, obtained from GRASPmp-CAM under SZA between 47.2º and 64.2º are
not a priori discarded like in GRASP-CAM. In this sense, Figure 16 shows the AOD differences between GRASPmp-CAM
and AERONET as a function of SZA, same as in Figure 11 but including the mentioned SZA interval. The results are similar
than for GRASP-CAM for the SZA values out of 47.2º-64.2º range, but with some additional outliers for low SZA values, that
overestimate the AOD from AERONET. AOD from GRASP-CAM overestimates AERONET values in the 47.2º-64.2º SZA
range (see Figure A31) due to the lack of information from low scattering angles and, hence these data were rejected; however,
most of the AOD differences between GRASPmp-CAM and AERONET are close to zero, as observed in Figure 16 even when
several data outliers overestimate the AERONET values. It is in agreement with the multi-pixel idea of transferring aerosol
information from pixels (observations) with more information (with low scattering angle measurement) to other pixels with
less information (without low scattering angles, as it is the case in the 47.2º-64.2º SZA range).

Figure 17 presents, for the three camera wavelengths, the density scatter-plots for GRASPmp-CAM vs. AERONET and
the frequency distribution of their differences. The amount of available data (25008) is much higher than for GRASP-CAM
(see Figure 9), which is partially caused by the addition of data on the 47.2º-64.2º SZA range. The AOD underestimation of
GRASPmp-CAM to the highest AERONET AOD values is clearly shown. The determination coefficient is a bit lower than for
GRASP-CAM, especially for longer wavelengths. Regarding the Md and STD of the AOD differences with AERONET, the
values for GRASPmp-CAM are similar to the ones obtained under single-pixel approach. The mentioned results of GRASP-
CAM under single-pixel, were obtained without data in the 47.2º-64.2º SZA range, therefore, for a more proper comparison,
the same values of Figure 17 have been represented in Figure A34, i.e. excluding the mentioned SZA range. As a result,
the amount of available data is still higher than for GRASP-CAM; the correlation, with $r^2$ values about 0.88, is a bit higher

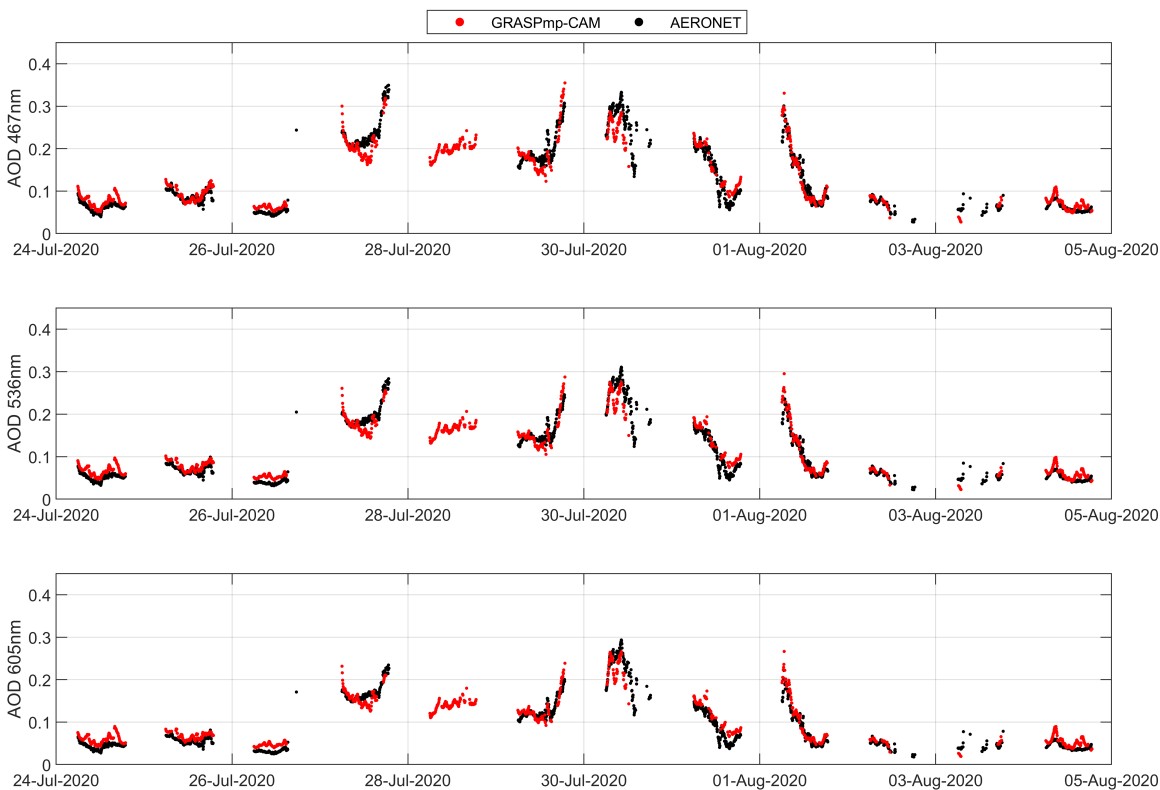

**Figure 15.** Aerosol optical depth (AOD) at 467 nm (upper panel), 536 nm (middle panel) and 605 nm (bottom panel) retrieved by GRASP under multi-pixel approach (GRASPmp-CAM) and by AERONET at Valladolid from $24^{th}$ July 2020 to $4^{th}$ August 2020.

than for GRASP-CAM; and the accuracy and precision are similar, with Md values between 0.005-0.010 and STD between 0.024-0.030.

In a similar way than in Section 4.1.1, the GRASPmp-CAM and AERONET AOD differences have been calculated for various AOD bins in order to study the dependence of accuracy and precision on AOD. Figure 18 shows the amount of data per AOD bin and the Md and STD of the AOD differences for each wavelength. The amount of data is, as expected, higher for GRASPmp-CAM than for GRASP-CAM (see Figure 10). The dependence on AOD of the Md and STD from GRASPmp-CAM and GRASP-CAM are similar, but GRASPmp-CAM shows slightly lower absolute values than GRASP-CAM in both

statistical estimators. Md is between -0.02 and 0.01 and STD below 0.04 for AOD below 0.25; the Md goes from -0.04 to -0.11±0.01 as AOD increases from 0.25 to 0.5; the STD in this interval is about 0.04 and 0.06. The accuracy and precision on GRASPmp-CAM AOD is approximately equal if data under 47.2º-64.2º SZA range is not considered in the analysis (Figure A35); but it presents better accuracy and precision for high AOD values.



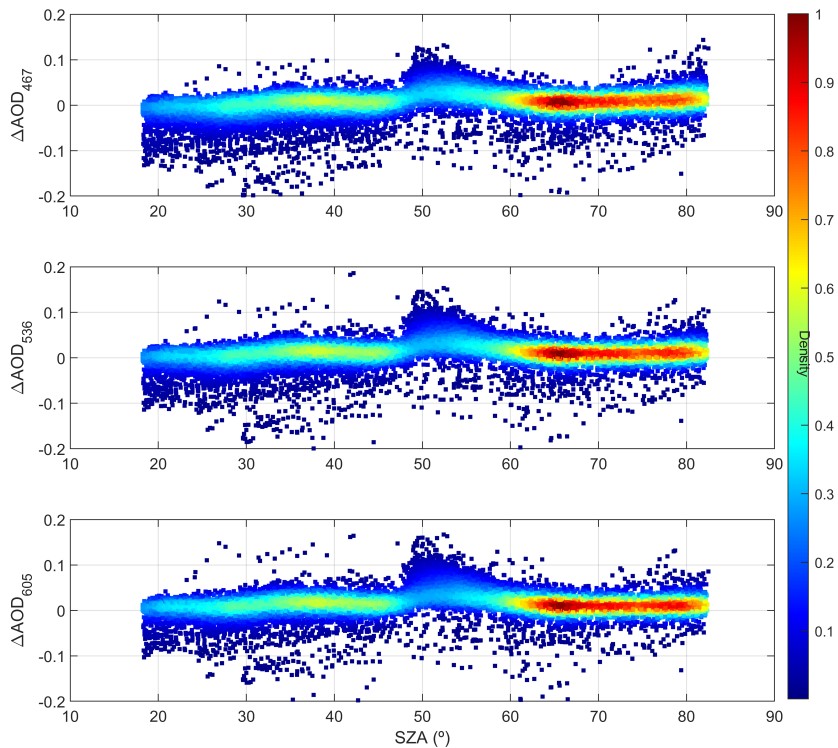

**Figure 16.** $\Delta$ differences on the aerosol optical depth (AOD) retrieved by GRASP under multi-pixel approach (GRASPmp-CAM) and the obtained from AERONET at 467 nm (upper panel), 536 nm (middle panel) and 605 nm (bottom panel) as a function of the solar zenith angle (SZA). Colour legend represents the density of the plotted data points.

### 4.2.2 Other aerosol properties

The time series of size distribution parameters retrieved by GRASPmp-CAM are shown in Figure 19 from $24^{th}$ July 2020 to $4^{th}$ August 2020 at Valladolid. Same as in AOD, the time series of size distribution parameters is smoother and looks less noisy than the values retrieved by the single-pixel approach of GRASP-CAM (see Figure 12).

The GRASPmp-CAM size distribution parameters are represented as a function of the AERONET ones in Figure 20. The amount of available data (3418) is higher than in the GRASP-CAM case (see Figure 13). GRASPmp-CAM data is more

correlated with AERONET than GRASP-CAM for the RF ($r^2$=0.39) and VCF ($r^2$=0.46), but less correlated for VCC ($r^2$=0.50) and VCT ($r^2$=0.54). This correlation between GRASPmp-CAM and AERONET increases when data under 47.2°-64.2° SZA range is not considered (see Figure A36). In this case, some GRASPmp-CAM data overestimating the low VCC, and hence VCT, disappear. As a result, he correlation of GRASPmp-CAM values of VCC ($r^2$=0.69) and VCT ($r^2$=0.72) with AERONET is higher than for GRASP-CAM.





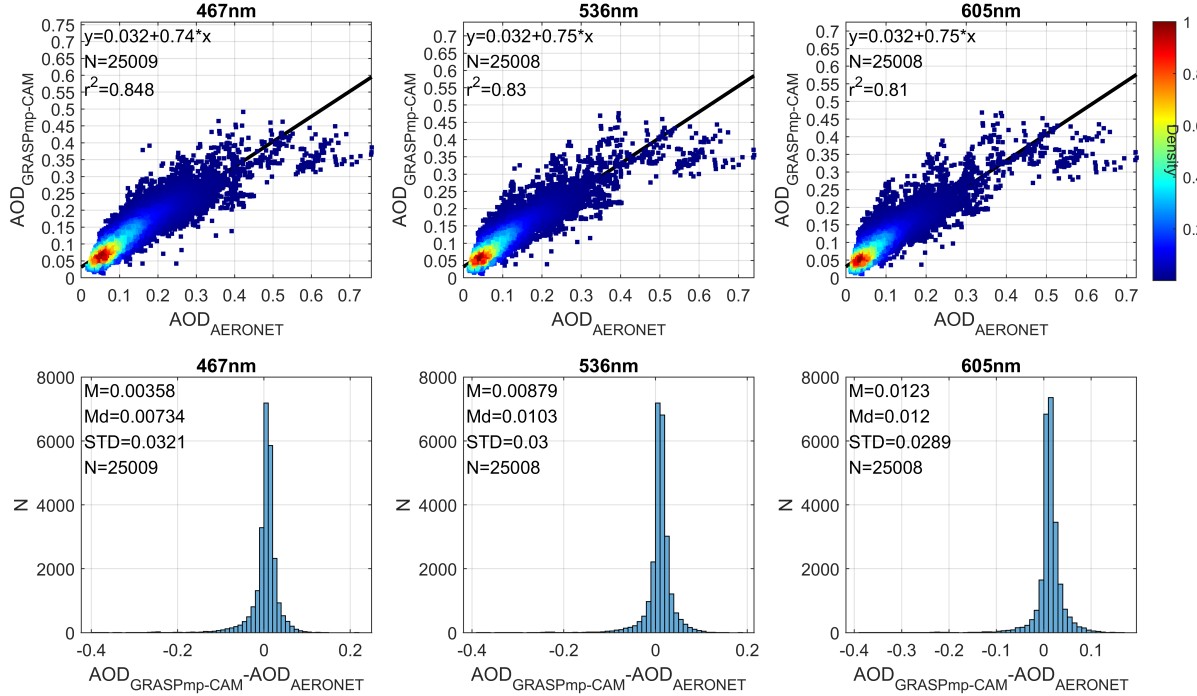

**Figure 17.** Upper panels) density scatter-plots of the aerosol optical depth (AOD) retrieved by GRASP in multi-pixel approach (GRASPmp-CAM) versus the AOD from AERONET at 467 nm (left panel), 536 nm (middle panel) and 605 nm (right panel); linear fit (black line) and its equation and the determination coefficient ($r^2$) are also shown. Bottom panels) Frequency histograms of the differences on AOD from GRASPmp-CAM and AERONET at 467 nm (left panel), 536 nm (middle panel) and 605 nm (right panel); The mean (M), median (Md) and standard deviation (STD) of these differences are also shown.

The differences between GRASPmp-CAM and AERONET on size distribution parameters are shown Figure 21. In general, the STD of the differences is slightly lower for GRASPmp-CAM than for GRASP-CAM except for VCC and VCT. On the other hand, GRASPmp-CAM presents Md values slightly farther from zero than GRASP-CAM for all size parameters except RF, σF and σC. These results vary if the data under 47.2º-64.2º SZA range are excluded in the frequency distributions (see Figure A37). In this case, the STD is reduced for VCC and VCT, while it is similar for the other parameters. The Md is also 500    closer to zero for VCF, VCC and VCT, but not for the other parameters, if the mentioned SZA values are discarded.

Finally, the time series for RRI and SPH retrieved by GRASPmp-CAM from $24^{th}$ July 2020 to $4^{th}$ August 2020 are shown as supplementary material in Figure A38. In a similar way than for AOD, the aerosol properties retrieved by GRASPmp-CAM present a less noisy behaviour in the time series compared with GRASP-CAM (see Figure A32). These retrieved parameters qualitatively fit better with AERONET than in GRASP-CAM case, but significant differences between GRASPmp-CAM and 505    AERONET can be still observed.





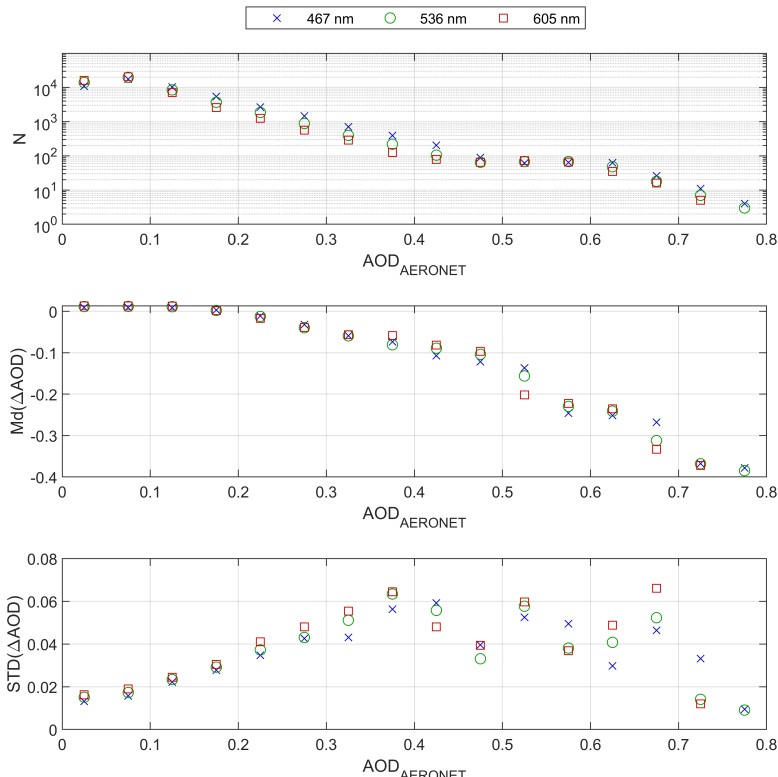

**Figure 18.** Median (Md; middle panel) and standard deviation (STD; bottom panel) of the $\Delta$ differences on the aerosol optical depth (AOD) retrieved by GRASP under multi-pixel approach (GRASPmp-CAM) and the obtained from AERONET at 467, 536 and 605 nm for different AOD bins. The available number of $\Delta$AOD data (N) per AOD bin is also shown in the upper panel for the three wavelengths.

## 5 Conclusions

This paper has analyzed in detail the feasibility of using normalized sky radiance (NSR) measurements at three effective wavelengths from an all-sky camera to retrieve aerosol properties using the GRASP code. This inversion method (camera NSR measurements on GRASP) has been called 'GRASP-CAM'. For this study, NSR measurements in the AERONET hybrid scan geometry are used, but with some limitations caused by technical problems of the camera: NSR measurements with lower scattering angles than 10º and any NSR measurement with solar zenith angle between between 48º and 65º cannot be used.

Thanks to an analysis with synthetic data, we can conclude that NSR measurements are sensitive to changes in AOD, at least until $AOD_{467}$ (AOD at 467 nm) values of 0.4-0.5. In this AOD range, the sky radiance is largely dominated by both Rayleigh and aerosol scattering, and the weight of each process is controlled by the AOD, hence sky radiance shape varies with AOD; for higher AOD values, the aerosol scattering dominates and hence the sensitivity to increasing AOD is reduced. NSR measurements are also sensitive to the aerosol type, even for high aerosol loads. This sensitivity is mainly located at low





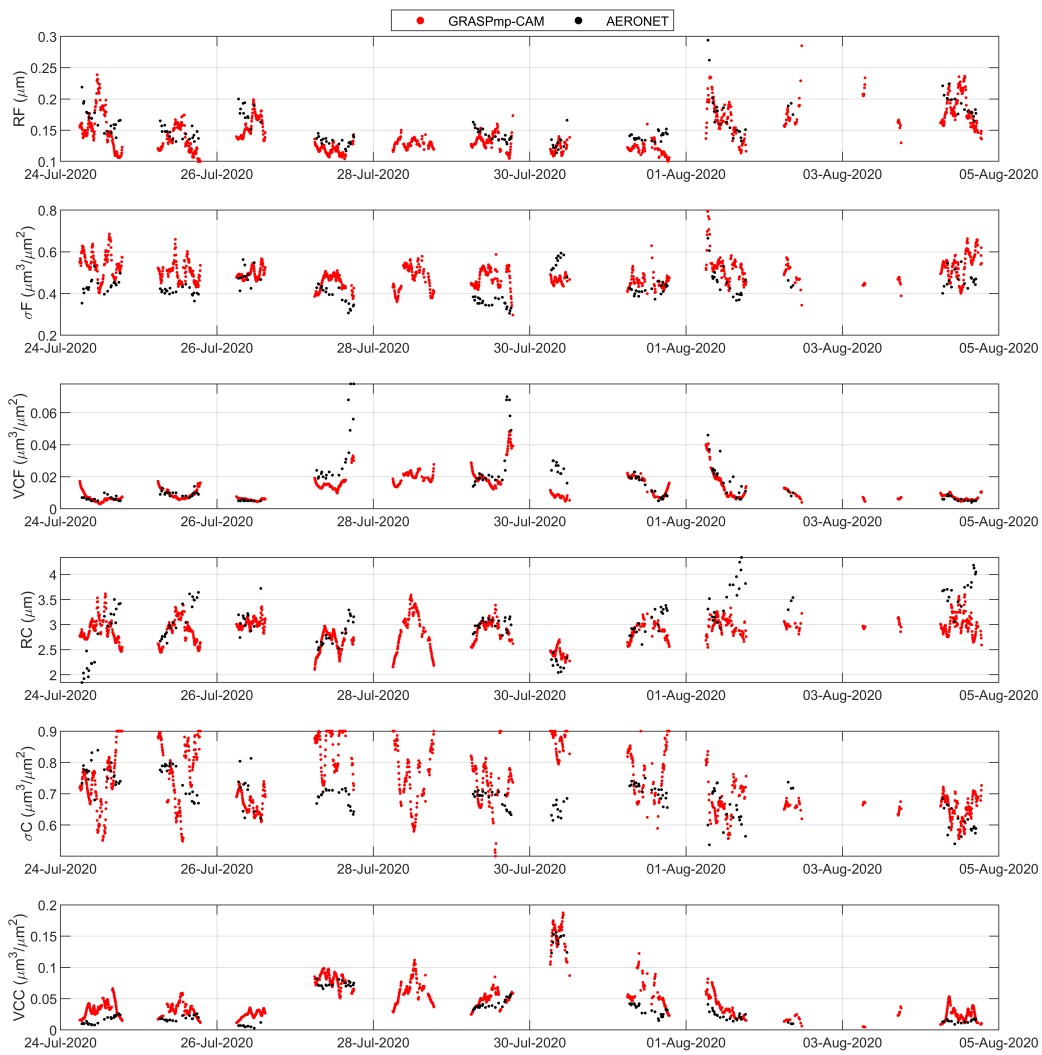

**Figure 19.** Volume median radius, *R*, standard deviation of log-normal distribution, $\sigma$ and aerosol volume concentration, *VC* for fine (F; upper panels) and coarse (C; bottom panels) modes retrieved by GRASP under multi-pixel approach (GRASPmp-CAM) and by AERONET at Valladolid, from 24[th] July 2020 to 4[th] August 2020.



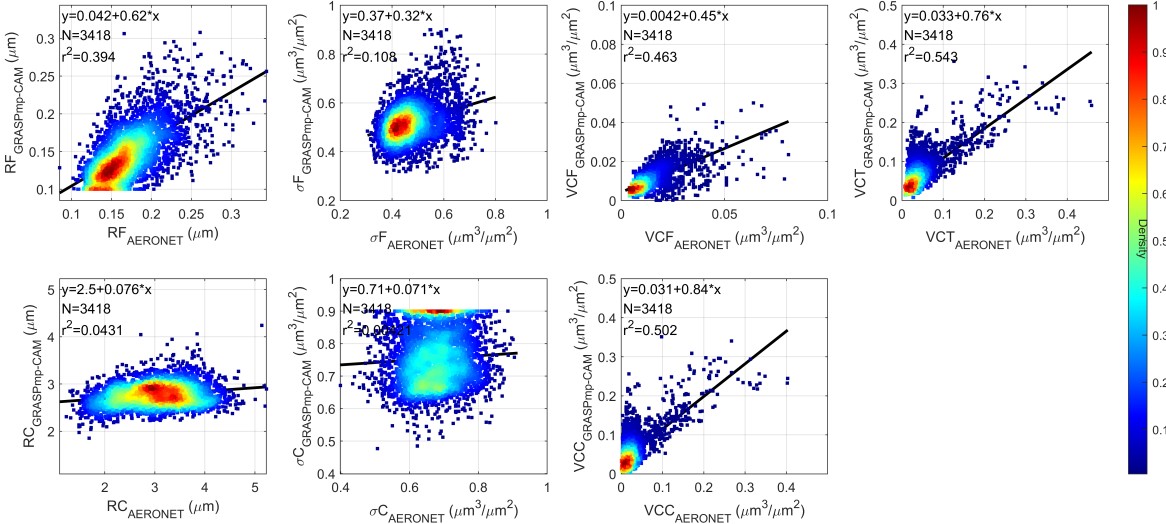

**Figure 20.** Density scatter-plots of the aerosol size distribution properties retrieved by GRASP in multi-pixel approach (GRASPmp-CAM) versus the ones retrieved by AERONET; linear fit (black line), its equation and the determination coefficient ($r^2$) are also shown. The size distribution properties are: volume median radius of fine (RF) and coarse (RC) modes; standard deviation of log-normal distribution for fine ($\sigma$F) and coarse ($\sigma$C) modes; and aerosol volume concentration for fine (VCF) and coarse (VCC) modes and the total value (VCT).

scattering angles, indicating that NSR at these low angles contains valuable information about the aerosol type, especially for coarse mode. There is not sensitivity of the NSR measurements to the aerosol absorption.

The accuracy and precision on the aerosol properties retrieved by GRASP have been tested also with synthetic data. The
theoretical accuracy on the retrieved AOD is generally within ±0.02 for AOD at $AOD_{467}$ values below or equal 0.4; while the theoretical precision goes from 0.01 to 0.05 as $AOD_{467}$ varies from 0.1 to 0.5. The AOD retrieved by GRASP using real NSR measurements correlates ($r^2 \approx 0.87$-$0.88$) with independent measurements taken by an AERONET Sun-sky photometer. The differences between both AOD sources are, as expected, higher for high AOD values, GRASP-CAM underestimating AERONET AOD as AOD increases. In general, the median and standard deviations of all these AOD differences have been
between 0.006-0.010 and between 0.024-0.030, respectively; it points out an overall combined uncertainty of AOD retrieved with GRASP-CAM about 0.026-0.030.

Regarding the aerosol volume size distribution, the theoretical precision and accuracy on the retrieved aerosol coarse mode improves if scattering angles lower than 10º are added to the inversion. Aerosol size distribution parameters retrieved by GRASP-CAM and by AERONET have been compared, and they have shown better correlation for the total ($r^2 \approx 0.61$), coarse
($r^2 \approx 0.57$) and fine ($r^2 \approx 0.38$) volume concentration and fine radius ($r^2 \approx 0.35$). GRASP-CAM has not shown enough sensitivity to the size distribution parameters of the coarse mode (radius and standard deviation), hence these products should be carefully used.





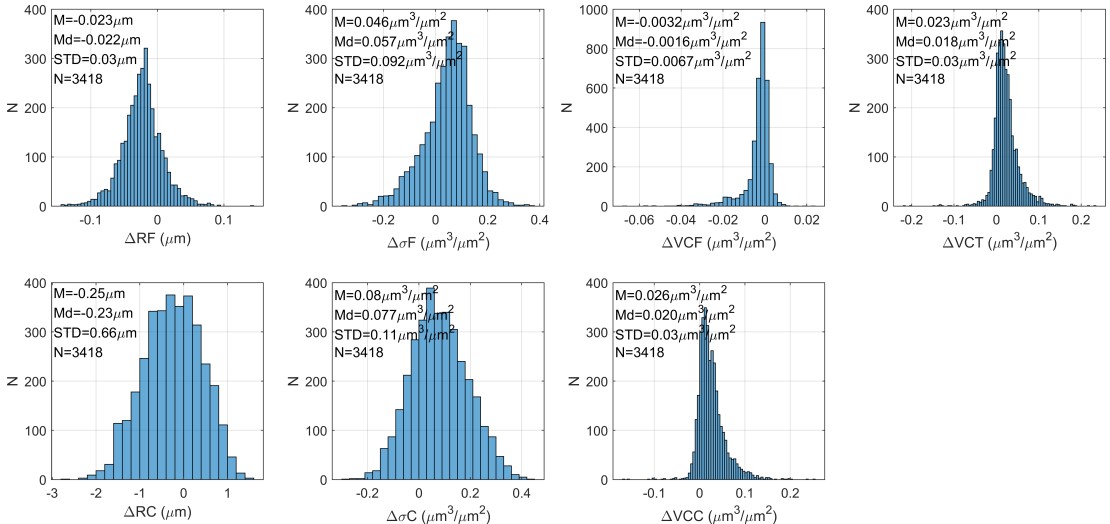

**Figure 21.** Frequency histograms of the $\Delta$ differences on the aerosol size distribution properties retrieved by GRASP in multi-pixel approach (GRASPmp-CAM) and the ones retrieved by AERONET; The mean (M), median (Md) and standard deviation (STD) of the differences are also shown. The size distribution properties are: volume median radius of fine (RF) and coarse (RC) modes; standard deviation of log-normal distribution for fine ($\sigma$F) and coarse ($\sigma$C) modes; and aerosol volume concentration for fine (VCF) and coarse (VCC) modes and the total value (VCT).

The GRASP retrievals have been done under single-pixel approach (each retrieval stand-alone). However, this method discards a significant number of potential retrievals since NSR measurements do not reach low scattering angles, where valuable

aerosol information is contained. Trying to solve that, the multi-pixel approach has been explored, which is a technique linking all the measurements of a full day, and constraining the temporal evolution of aerosol properties along that day (smoothness critaria). As result, more retrievals are achieved, showing more accurate AOD values when low scattering angles are not available than in the single-pixel approach. However, the accuracy on AOD in the retrievals where low scatterings angles can not be reached is still low in multi-pixel approach. A slight improvement in the retrieved AOD has been observed when multi-pixel

approach is used instead of single-pixel, if the most problematic Sun positions (due to dome reflections) are not considered. The temporal evolution of other aerosol properties is less noisy and make more physical sense using multi-pixel than single-pixel approach.

This work takes all-sky cameras one step beyond in their capability to obtain atmospheric aerosol properties. Normalized sky radiance measurements contain information about the aerosol properties and, therefore, they are useful to obtain the AOD, at

least for low and moderate aerosol loads. The retrieved AOD under high aerosol loads should be carefully taken. The sensitivity of NSR measurements to aerosol type even for high AOD values, suggest that these camera measurements could be combined in GRASP with other kind of measurements more sensitive to AOD, or even directly AOD from other instruments, in order



to obtain more accurate aerosol properties, including high aerosol load scenes. The availability of sky radiances under low scattering angles is important to obtain an accurate retrieval of aerosol properties, hence, we recommend manufacturers of all-
sky cameras to work on the elimination of reflections in these instruments, in order to cover a greater range of useful scattering angles. The multi-pixel approach seems to be an interesting technique to retrieve aerosol properties, and its potential should be explored in more detail in future works.

*Data availability.*  AERONET data are publicly available on the AERONET web page (https://aeronet.gsfc.nasa.gov/, last access: 16 June 2021, NASA, 2021). Normalized sky radiances from the all-sky camera and the retrieved aerosol products are available upon request to the
authors.

*Author contributions.*  RR and JCAS designed and developed the main concepts and ideas behind this work and wrote the paper with input from all authors. They also processed the camera measurements and the GRASP retrievals. RR, BT, VEC, and CT discussed and defined the used inversion strategy. CL developed and provided the all-sky camera used. RG was responsible for the camera's operation at the Valladolid station. VEC and CT were responsible for the Valladolid AERONET station. DF and TL contributed with the changes on GRASP code for
the use normalized sky radiances as input. DM, OD and AMdF contributed in the interpretation of the results. All authors were involved in helpful discussions and contributed to the manuscript.

*Competing interests.*  The authors declare that they have no conflict of interest.

*Acknowledgements.*  This research was funded by the Ministerio de Ciencia, Innovación y Universidades (grant no. RTI2018-097864-B-I00) and by Junta de Castilla y León (grant no. VA227P20). Sieltec Canarias SL is acknowledged for lending the all-sky camera. The
authors acknowledge the use of GRASP inversion algorithm software (https://www.grasp-open.com, last access: 16 June 2021). The authors gratefully thank AERONET for the aerosol products used. Finally, the authors thank the GOA-UVa staff members (R. Carracedo, S. Herrero and P. Martín), who helped with the maintenance of the instruments and support the station infrastructure. R. Román wants to acknowledge the interesting personal talks about GRASP code with J.A. Benavent, M. Herrera and M. Herreras.





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

**Appendix A: Supplementary material**





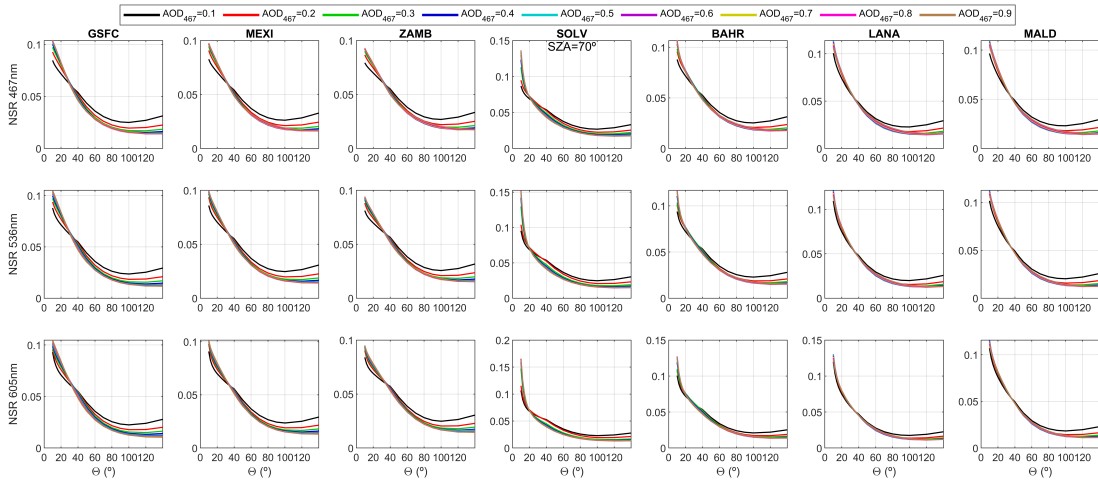

**Figure A1.** Normalized sky radiance (NSR) under a solar zenith angle (SZA) of 70º at 467 nm (top row), 536 nm (middle row) and 605 nm (bottom row) as a function of scattering angle ($\theta$) for different AOD (at 467 nm) values and and for nine aerosol models.

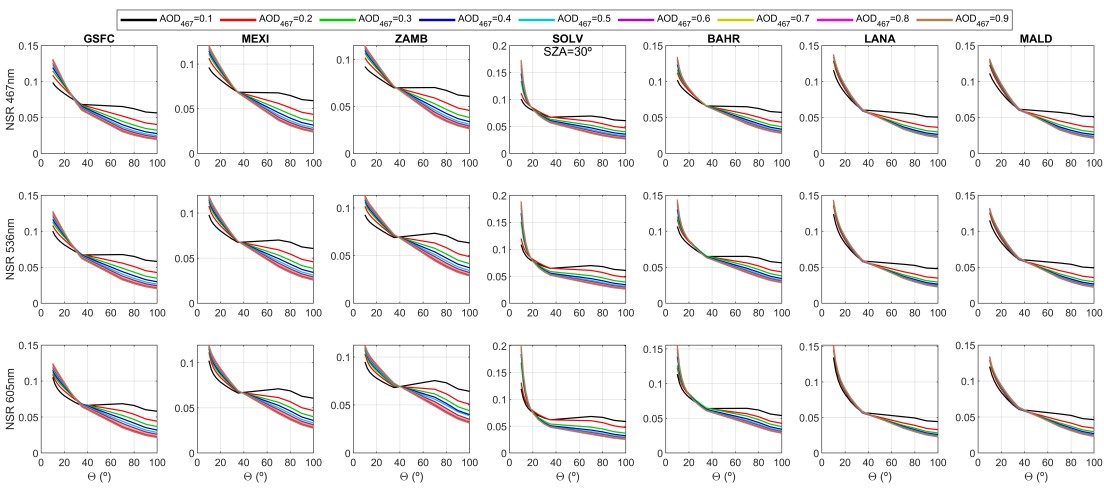

**Figure A2.** Normalized sky radiance (NSR) under a solar zenith angle (SZA) of 30º at 467 nm (top row), 536 nm (middle row) and 605 nm (bottom row) as a function of scattering angle ($\theta$) for different AOD (at 467 nm) values and and for nine aerosol models.





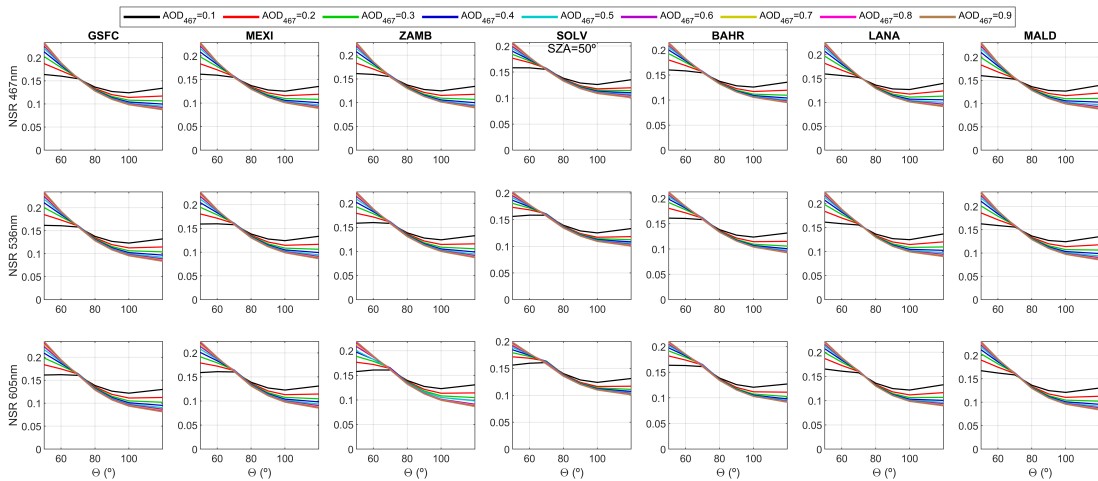

**Figure A3.** Normalized sky radiance (NSR) under a solar zenith angle (SZA) of 50° at 467 nm (top row), 536 nm (middle row) and 605 nm (bottom row) as a function of scattering angle ($\theta$) for different AOD (at 467 nm) values and and for nine aerosol models.

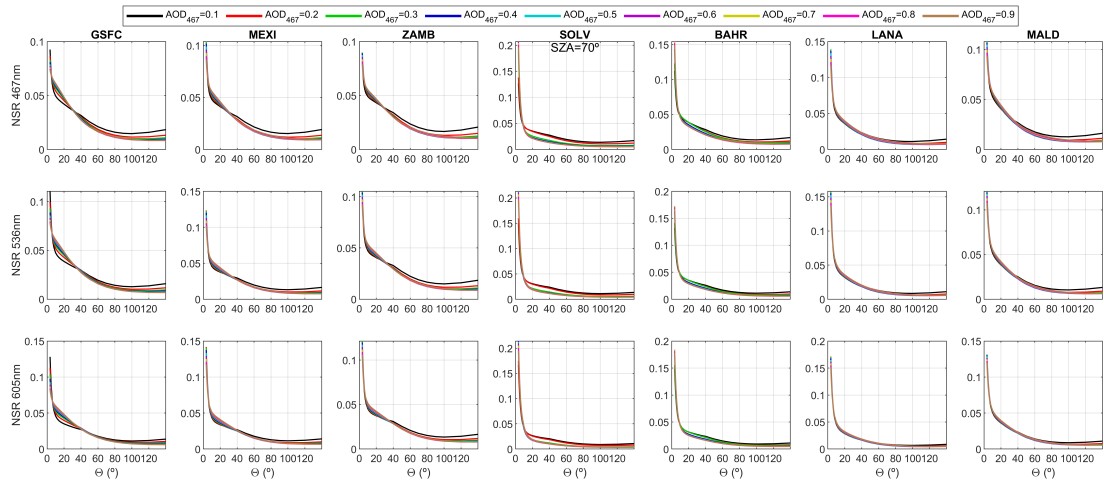

**Figure A4.** Normalized sky radiance (NSR) under a solar zenith angle (SZA) of 70° at 467 nm (top row), 536 nm (middle row) and 605 nm (bottom row) as a function of scattering angle ($\theta$) for different AOD (at 467 nm) values and and for nine aerosol models. Scattering angles up to 3° instead of 10° are included.





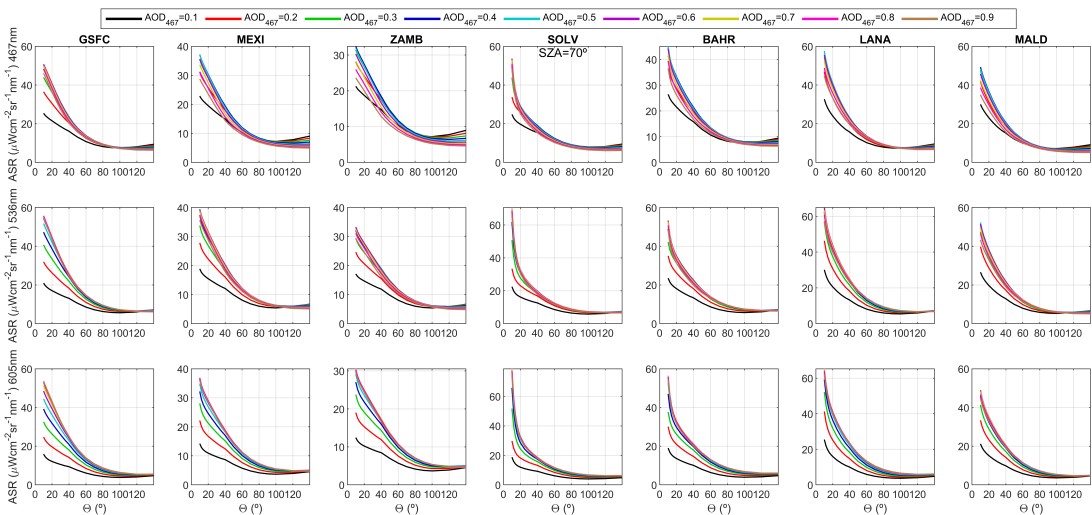

**Figure A5.** Absolute sky radiance (ASR) under a solar zenith angle (SZA) of 70º at 467 nm (top row), 536 nm (middle row) and 605 nm (bottom row) as a function of scattering angle ($\theta$) for different AOD (at 467 nm) values and and for nine aerosol models.

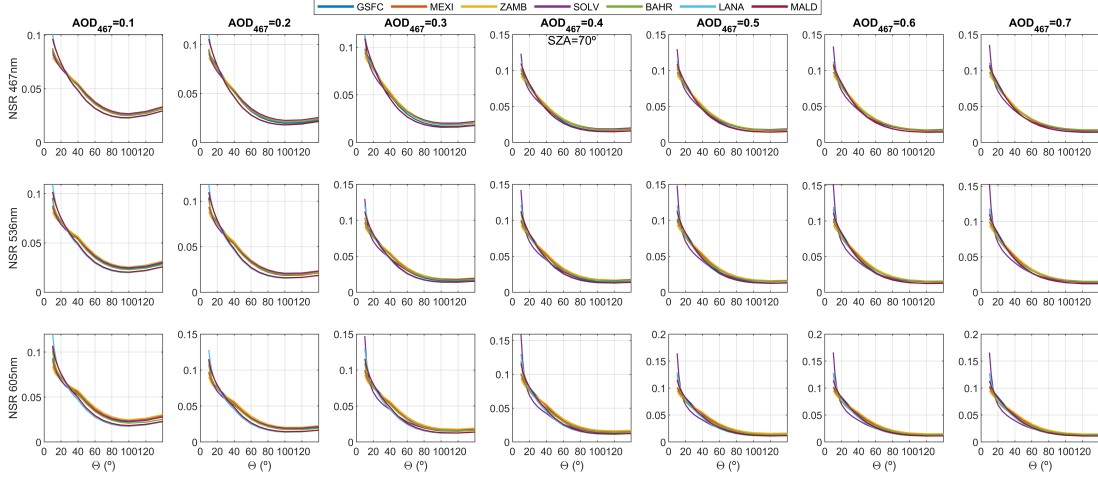

**Figure A6.** Normalized sky radiance (NSR) under solar zenith angle (SZA) of 70º at 467 nm (top row), 536 nm (middle row) and 605 nm (bottom row) as a function of scattering angle ($\theta$) for different aerosol models and AOD at 467 nm ($AOD_{467}$) values.





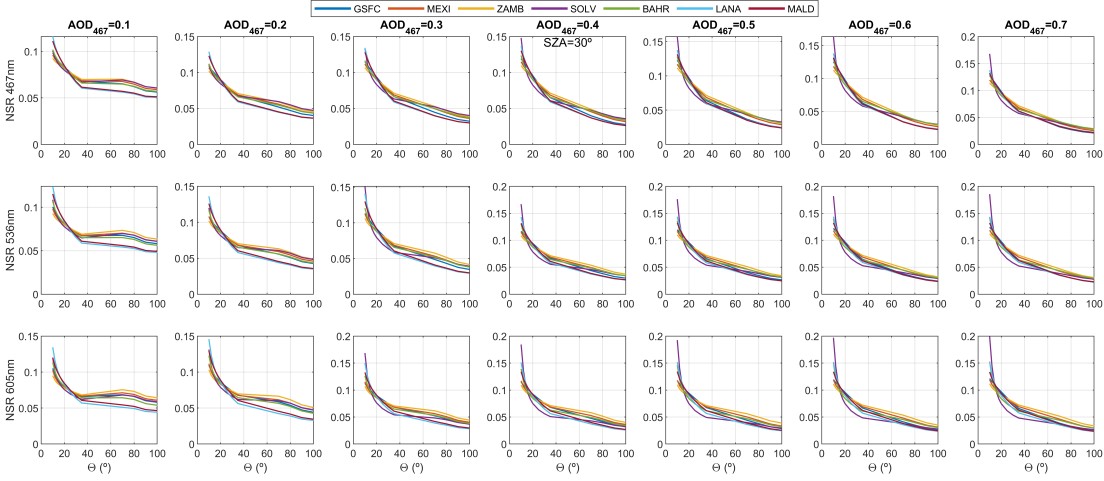

**Figure A7.** Normalized sky radiance (NSR) under solar zenith angle (SZA) of 30º at 467 nm (top row), 536 nm (middle row) and 605 nm (bottom row) as a function of scattering angle ($\theta$) for different aerosol models and AOD at 467 nm ($AOD_{467}$) values.

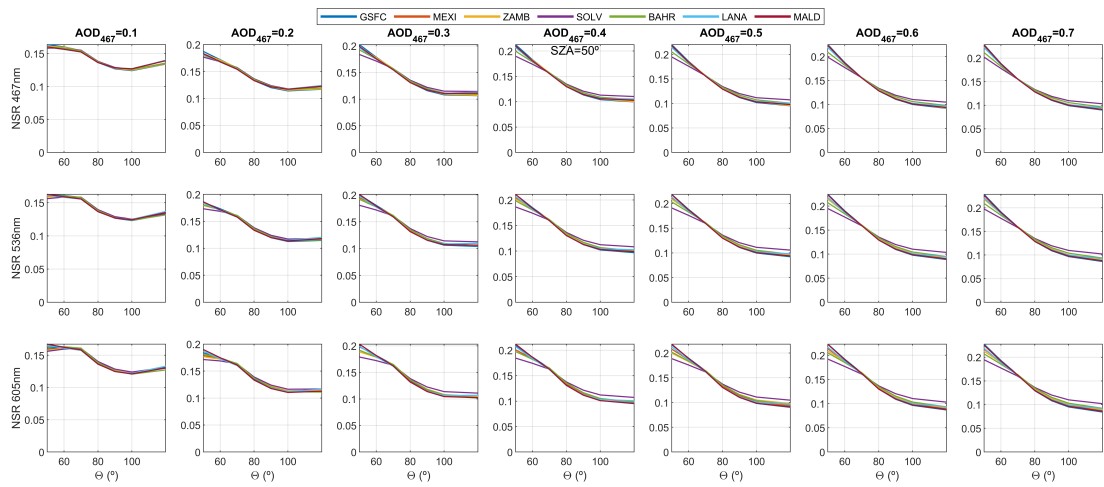

**Figure A8.** Normalized sky radiance (NSR) under solar zenith angle (SZA) of 50º at 467 nm (top row), 536 nm (middle row) and 605 nm (bottom row) as a function of scattering angle ($\theta$) for different aerosol models and AOD at 467 nm ($AOD_{467}$) values.





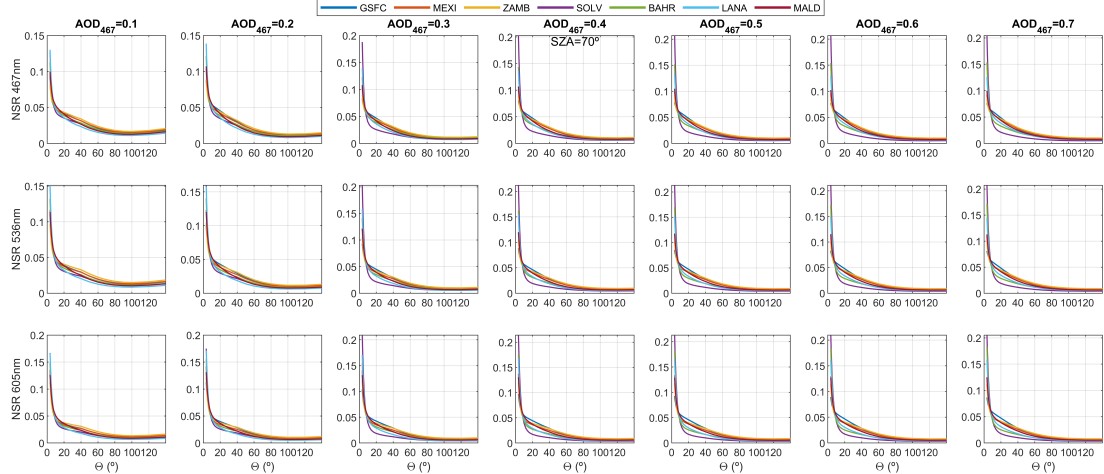

**Figure A9.** Normalized sky radiance (NSR) under solar zenith angle (SZA) of 70º at 467 nm (top row), 536 nm (middle row) and 605 nm (bottom row) as a function of scattering angle ($\theta$) for different aerosol models and AOD at 467 nm ($AOD_{467}$) values. Scattering angles up to 3º instead of 10º are included.

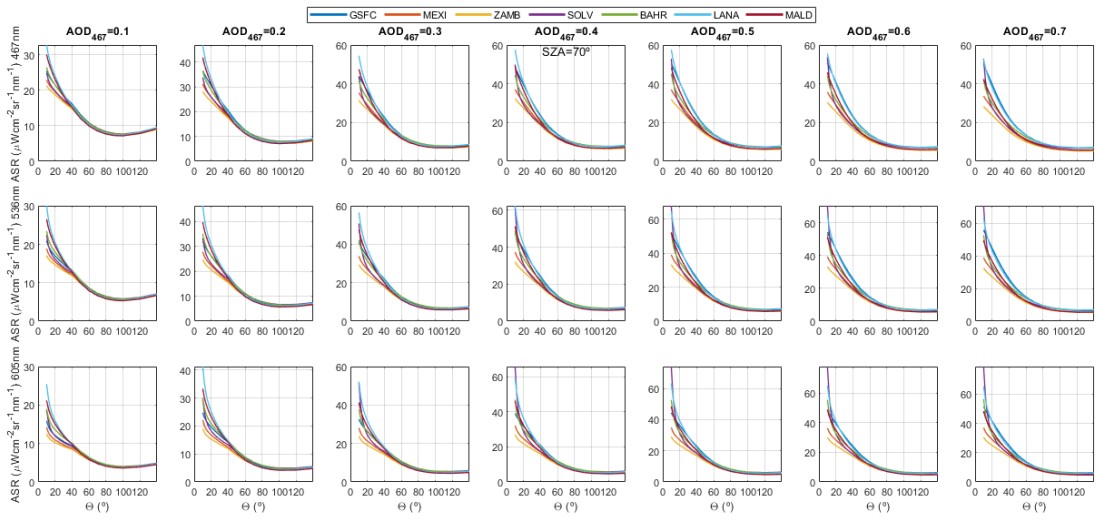

**Figure A10.** Absolute sky radiance (ASR) under solar zenith angle (SZA) of 70º at 467 nm (top row), 536 nm (middle row) and 605 nm (bottom row) as a function of scattering angle ($\theta$) for different aerosol models and AOD at 467 nm ($AOD_{467}$) values.



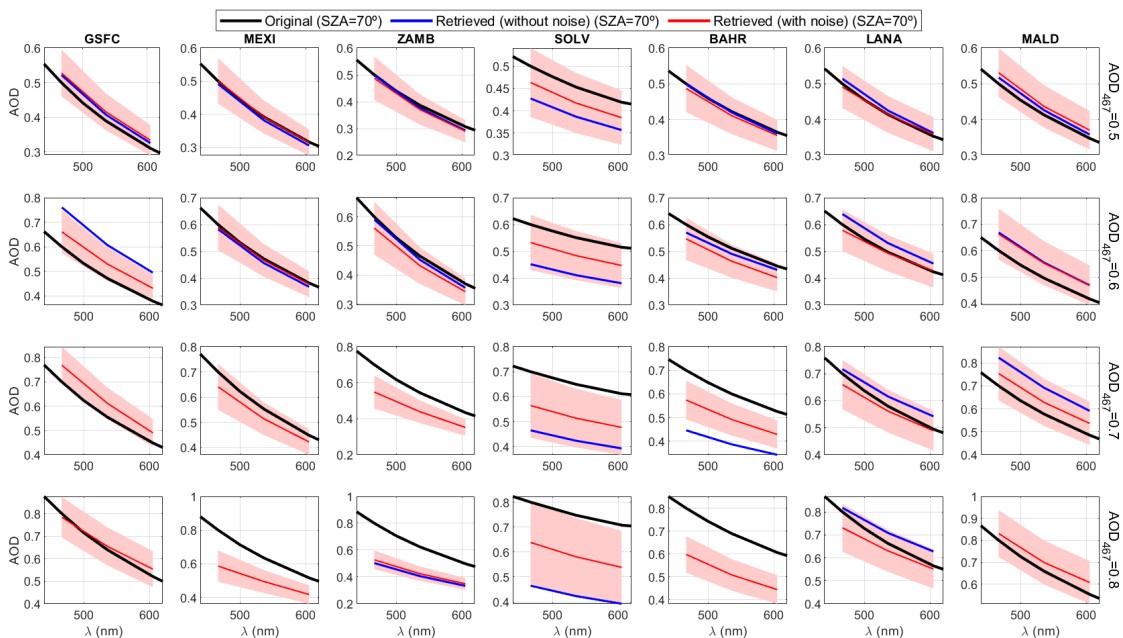

**Figure A11.** Original (black line), retrieved without noise (blue line), and median of all retrieved with noise (red line) aerosol optical depth (AOD) under solar zenith angle (SZA) equal to 70°. These AOD values are represented for different aerosol types (one type per column) and for AOD at 467nm ($AOD_{467}$) values of 0.5 (first row), 0.6 (second row), 0.7 (third row) and 0.8 (last row). Red shadowed-area corresponds to ± the standard deviation of all averaged size distributions retrieved with noise.



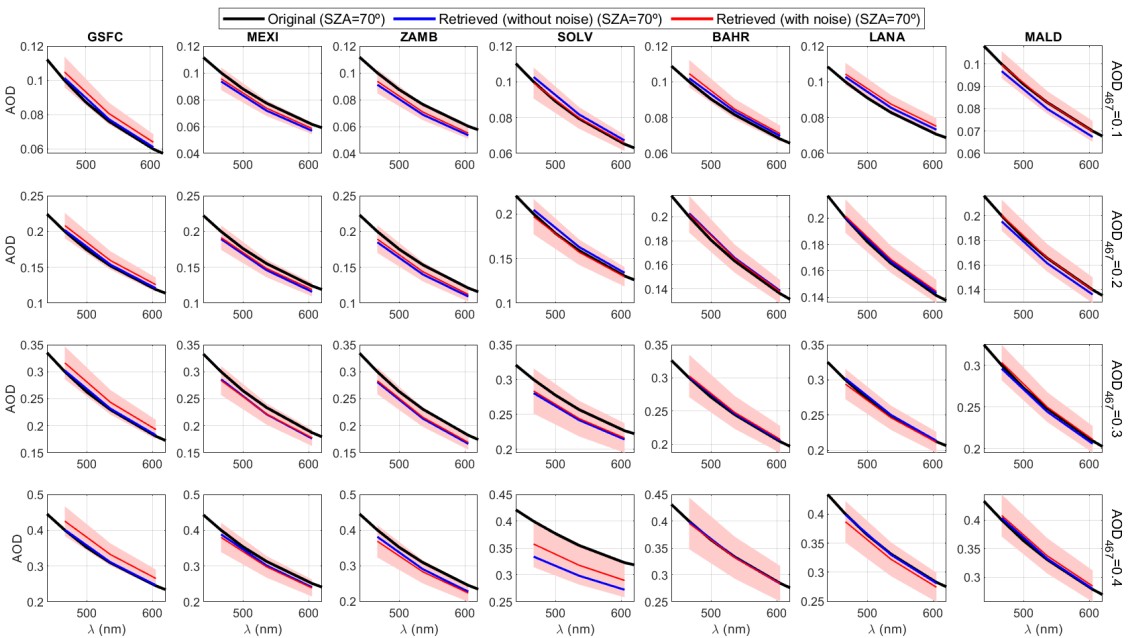

**Figure A12.** Original (black line), retrieved without noise (blue line), and median of all retrieved with noise (red line) aerosol optical depth (AOD) under solar zenith angle (SZA) equal to 70º. The retrievals have been done including scattering angles up to 3º instead of 10º. These AOD values are represented for different aerosol types (one type per column) and for AOD at 467nm ($AOD_{467}$) values of 0.1 (first row), 0.2 (second row), 0.3 (third row) and 0.4 (last row). Red shadowed-area corresponds to $\pm$ the standard deviation of all averaged size distributions retrieved with noise.



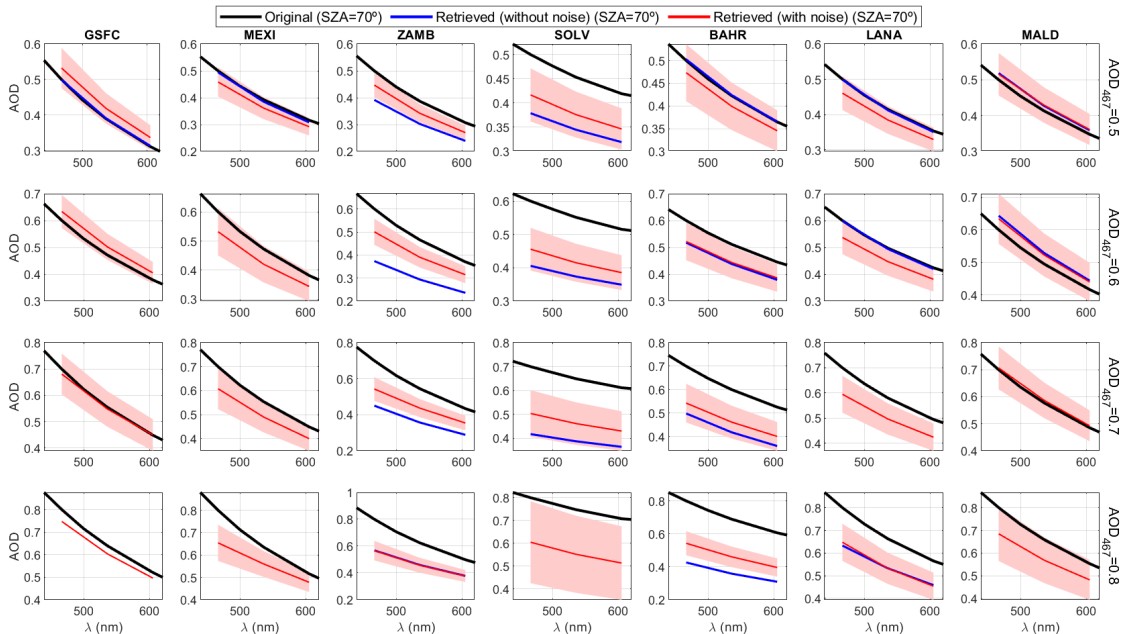

**Figure A13.** Original (black line), retrieved without noise (blue line), and median of all retrieved with noise (red line) aerosol optical depth (AOD) under solar zenith angle (SZA) equal to 70°. The retrievals have been done including scattering angles up to 3° instead of 10°. These AOD values are represented for different aerosol types (one type per column) and for AOD at 467nm ($AOD_{467}$) values of 0.5 (first row), 0.6 (second row), 0.7 (third row) and 0.8 (last row). Red shadowed-area corresponds to ± the standard deviation of all averaged size distributions retrieved with noise.

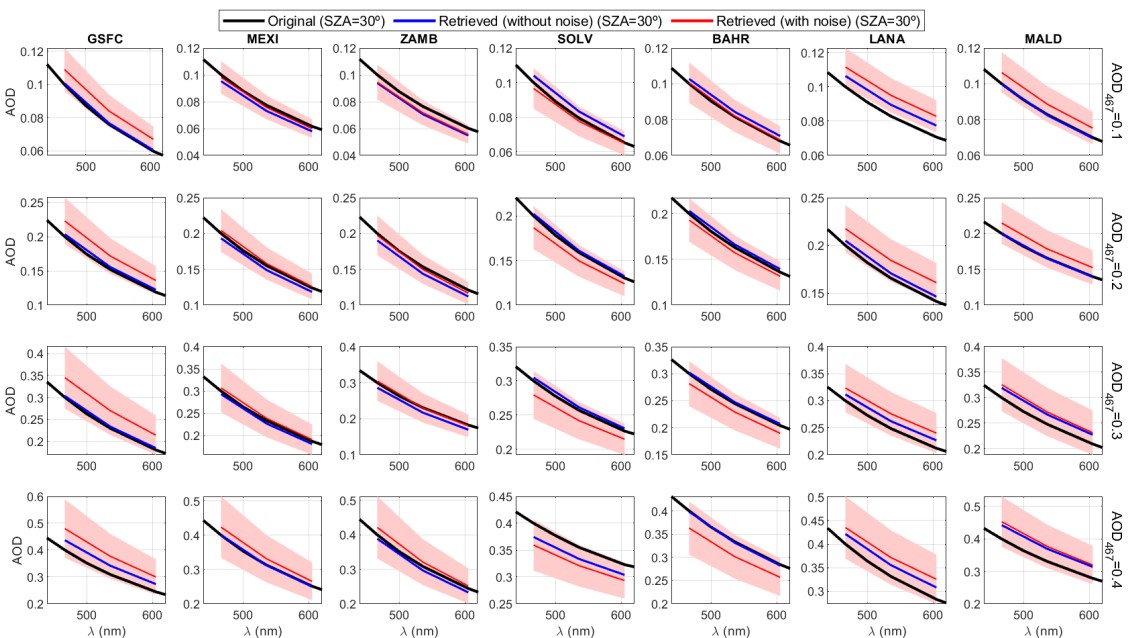

**Figure A14.** Original (black line), retrieved without noise (blue line), and median of all retrieved with noise (red line) aerosol optical depth (AOD) under solar zenith angle (SZA) equal to 30°. These AOD values are represented for different aerosol types (one type per column) and for AOD at 467nm ($AOD_{467}$) values of 0.1 (first row), 0.2 (second row), 0.3 (third row) and 0.4 (last row). Red shadowed-area corresponds to $\pm$ the standard deviation of all averaged size distributions retrieved with noise.



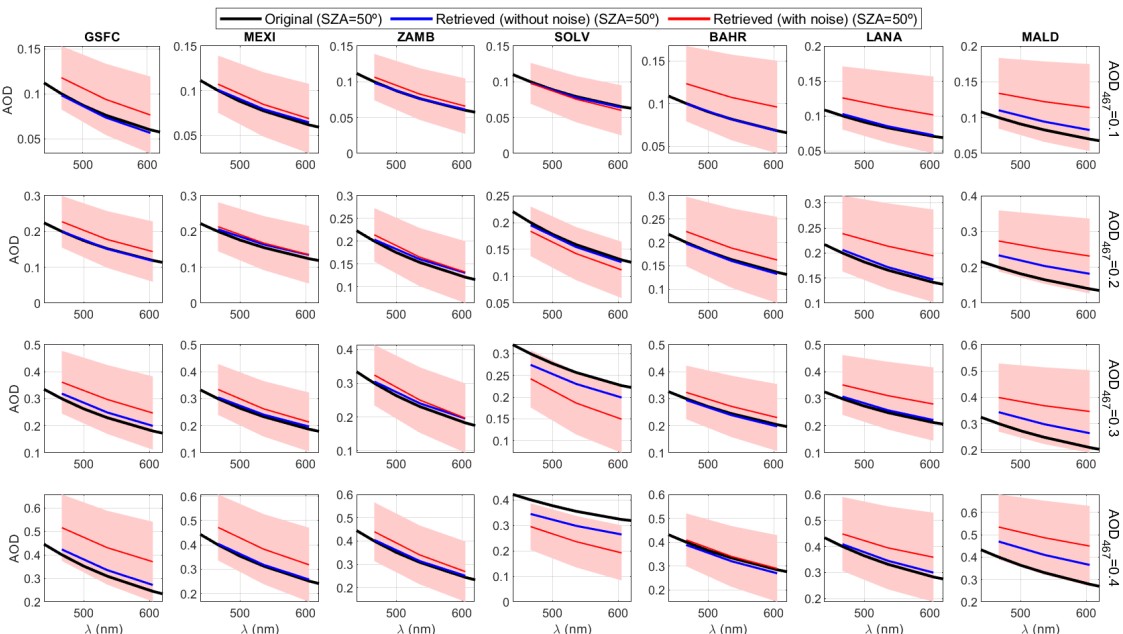

**Figure A15.** Original (black line), retrieved without noise (blue line), and median of all retrieved with noise (red line) aerosol optical depth (AOD) under solar zenith angle (SZA) equal to 50º. These AOD values are represented for different aerosol types (one type per column) and for AOD at 467nm ($AOD_{467}$) values of 0.1 (first row), 0.2 (second row), 0.3 (third row) and 0.4 (last row). Red shadowed-area corresponds to $\pm$ the standard deviation of all averaged size distributions retrieved with noise.

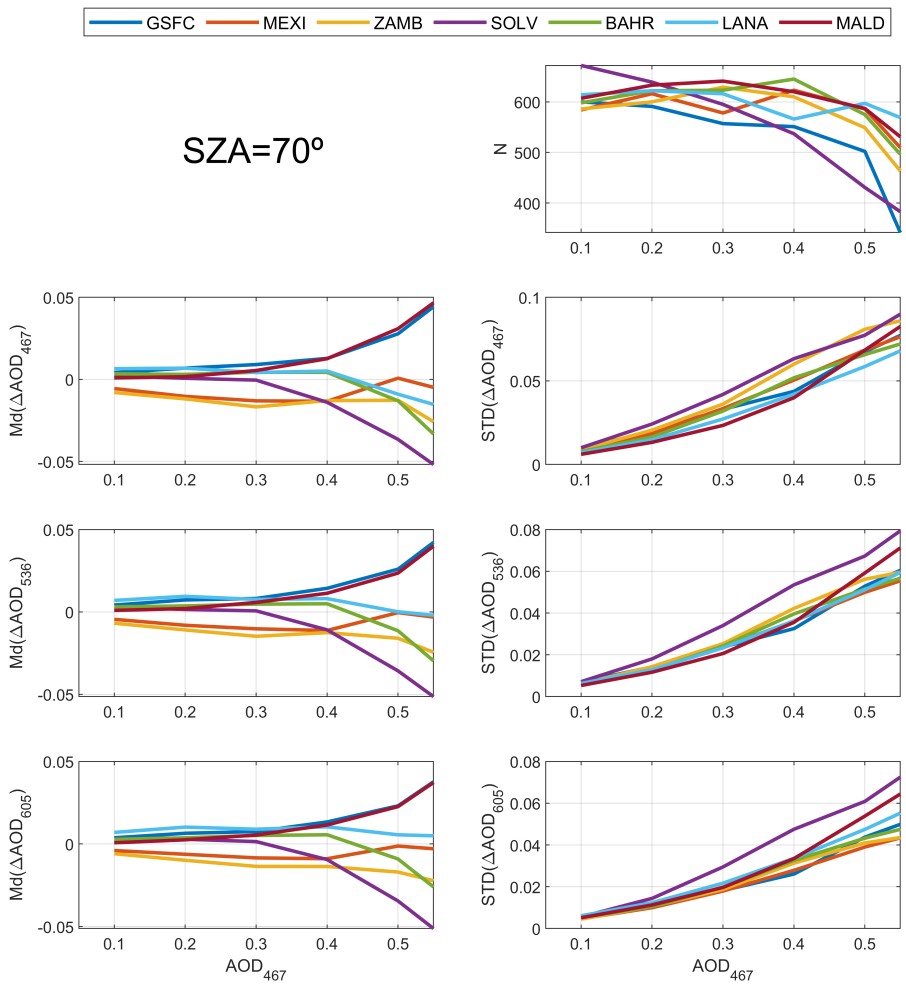

**Figure A16.** Median (Md) and standard deviation (STD) of the Δ differences between the available retrieved aerosol properties with noise and the original ones. The amount of available retrievals (N) is also shown. Only the retrievals under solar zenith angle (SZA) equal to 70º are used. The aerosol properties shown are aerosol optical depth (AOD) at 467 nm ($AOD_{467}$), 536 nm ($AOD_{536}$) and 605 nm ($AOD_{605}$). These Md and STD values are represented as a function of ($AOD_{467}$) for different aerosol types. x-axis goes from 0 to 0.5 to a better observation of this range.

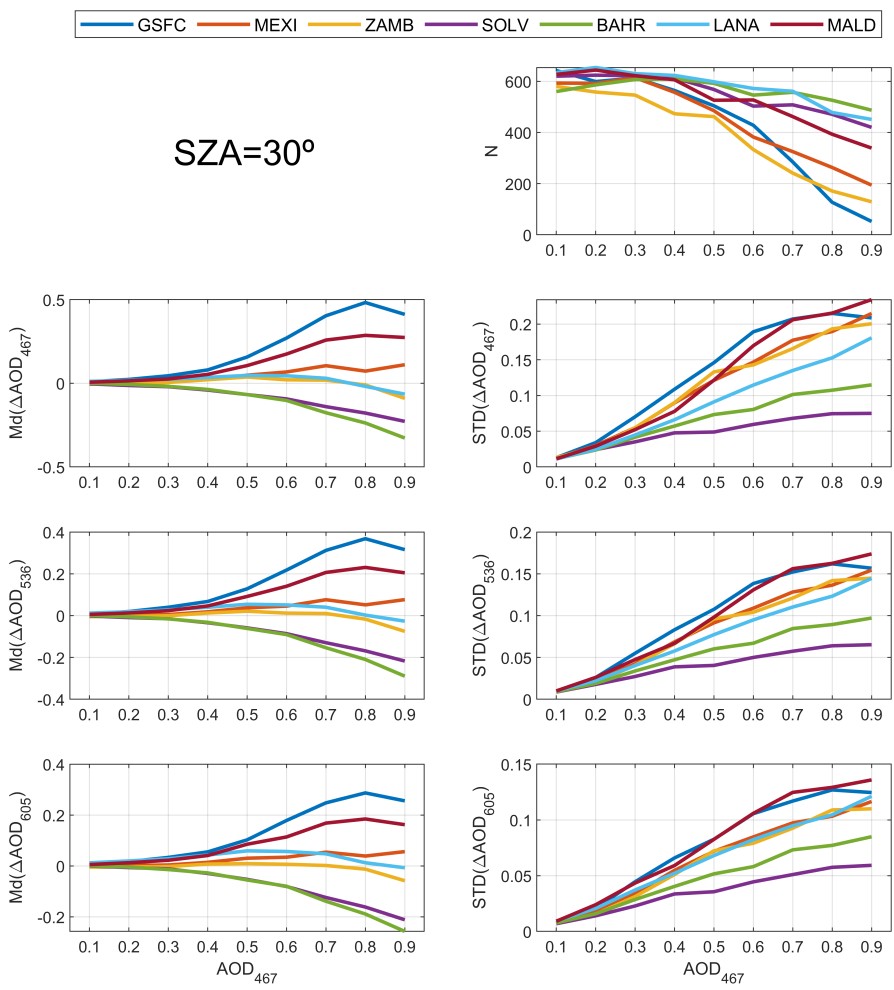

**Figure A17.** Median (Md) and standard deviation (STD) of the $\Delta$ differences between the available retrieved aerosol properties with noise and the original ones. The amount of available retrievals (N) is also shown. Only the retrievals under solar zenith angle (SZA) equal to 30º are used. The aerosol properties shown are aerosol optical depth (AOD) at 467 nm ($AOD_{467}$), 536 nm ($AOD_{536}$) and 605 nm ($AOD_{605}$). These Md and STD values are represented as a function of ($AOD_{467}$) for different aerosol types.



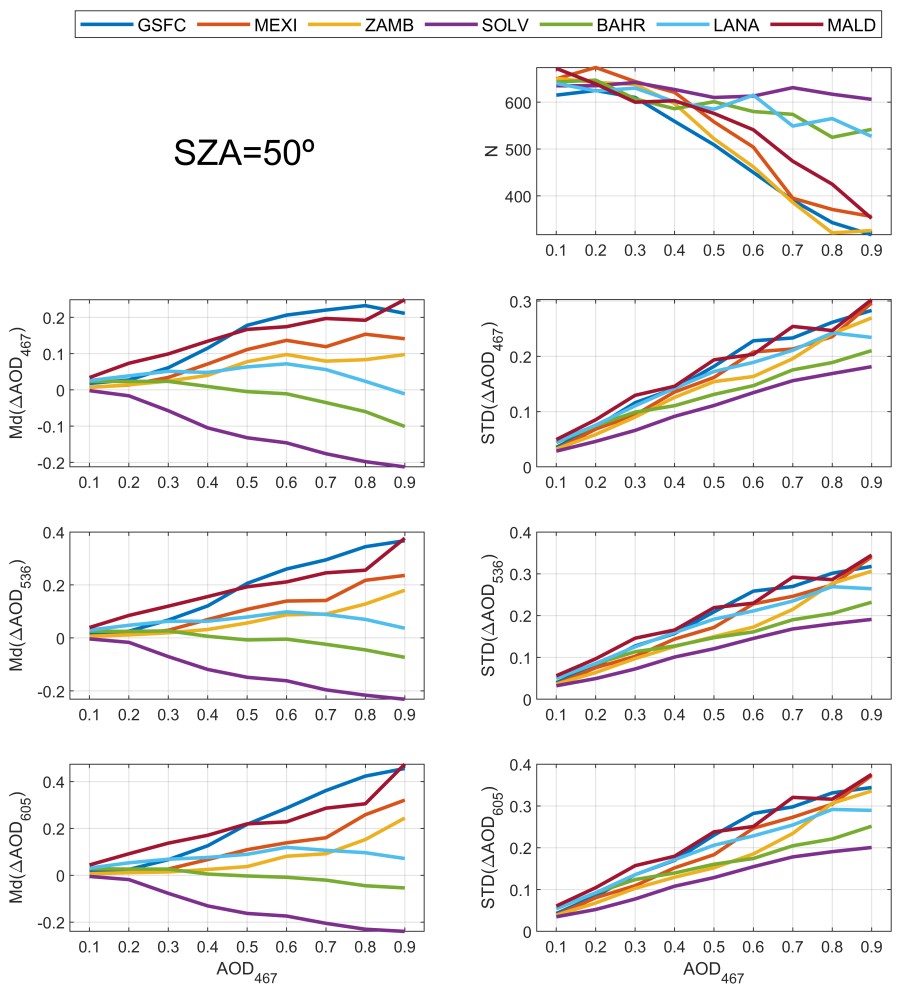

**Figure A18.** Median (Md) and standard deviation (STD) of the $\Delta$ differences between the available retrieved aerosol properties with noise and the original ones. The amount of available retrievals (N) is also shown. Only the retrievals under solar zenith angle (SZA) equal to 50º are used. TThe aerosol properties shown are aerosol optical depth (AOD) at 467 nm ($AOD_{467}$), 536 nm ($AOD_{536}$) and 605 nm ($AOD_{605}$). These Md and STD values are represented as a function of ($AOD_{467}$) for different aerosol types.



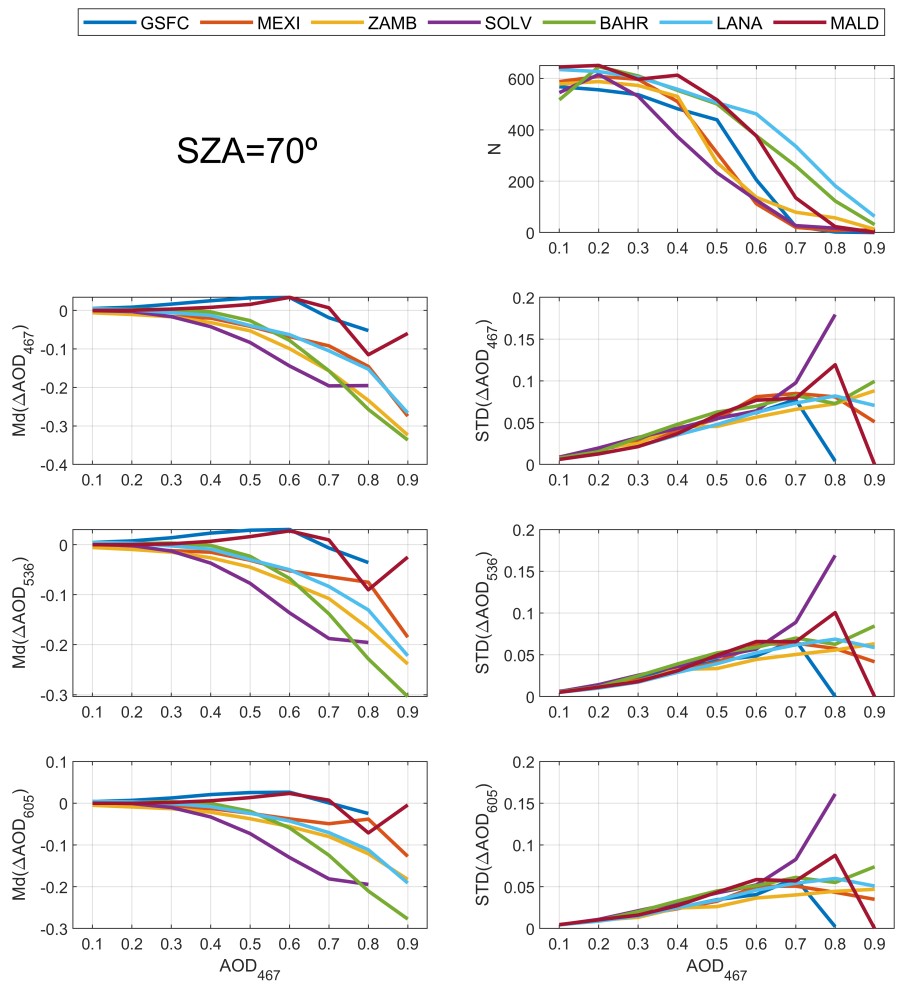

**Figure A19.** Median (Md) and standard deviation (STD) of the Δ differences between the available retrieved aerosol properties with noise and the original ones. The amount of available retrievals (N) is also shown. Only the retrievals under solar zenith angle (SZA) equal to 70º are used. The aerosol properties shown are aerosol optical depth (AOD) at 467 nm ($AOD_{467}$), 536 nm ($AOD_{536}$) and 605 nm ($AOD_{605}$). These Md and STD values are represented as a function of ($AOD_{467}$) for different aerosol types. The retrievals have been done including scattering angles up to 3º instead of 10º.



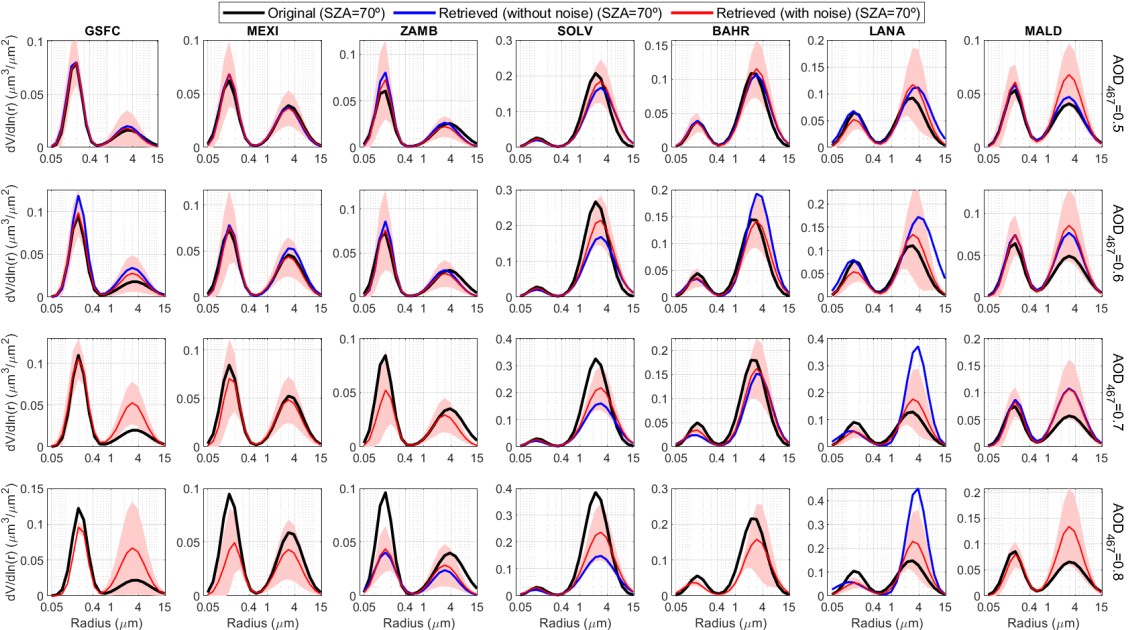

**Figure A20.** Original (black line), retrieved without noise (blue line), and median of all retrieved with noise (red line) aerosol volume size distributions under solar zenith angle (SZA) equal to 70º. These size distributions are represented for different aerosol types (one type per column) and for AOD (aerosol optical depth) at 467nm ($AOD_{467}$) values of 0.5 (first row), 0.6 (second row), 0.7 (third row) and 0.8 (last row). Red shadowed-area corresponds to $\pm$ the standard deviation of all averaged size distributions retrieved with noise.

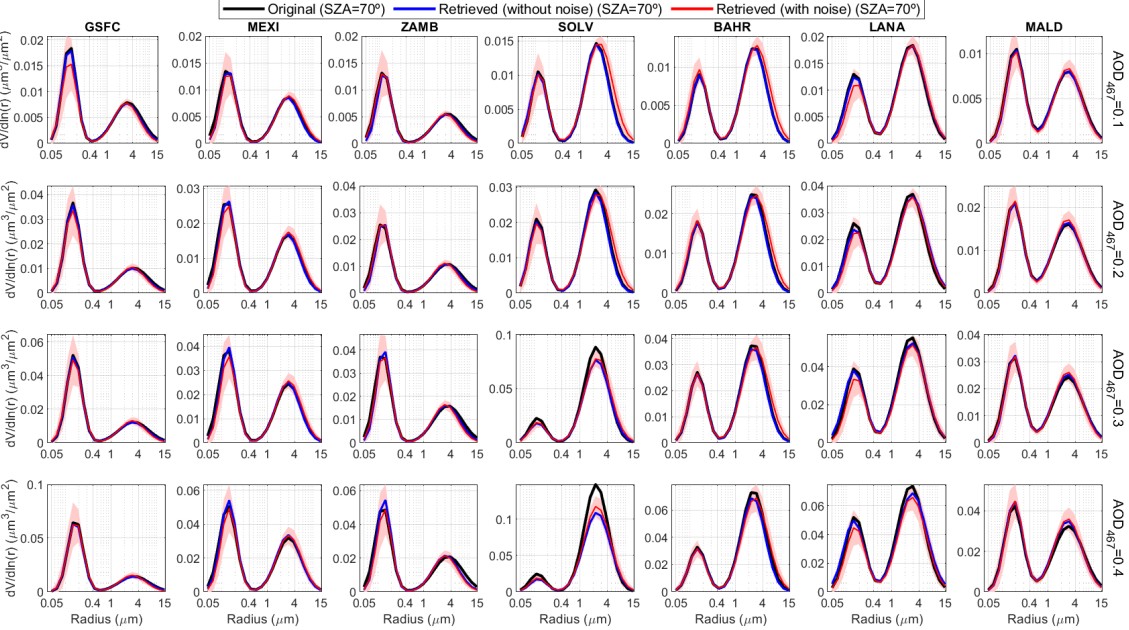

**Figure A21.** Original (black line), retrieved without noise (blue line), and median of all retrieved with noise (red line) aerosol volume size distributions under solar zenith angle (SZA) equal to 70°. The retrievals have been done including scattering angles up to 3° instead of 10°. These size distributions are represented for different aerosol types (one type per column) and for AOD (aerosol optical depth) at 467nm ($AOD_{467}$) values of 0.1 (first row), 0.2 (second row), 0.3 (third row) and 0.4 (last row). Red shadowed-area corresponds to $\pm$ the standard deviation of all averaged size distributions retrieved with noise.



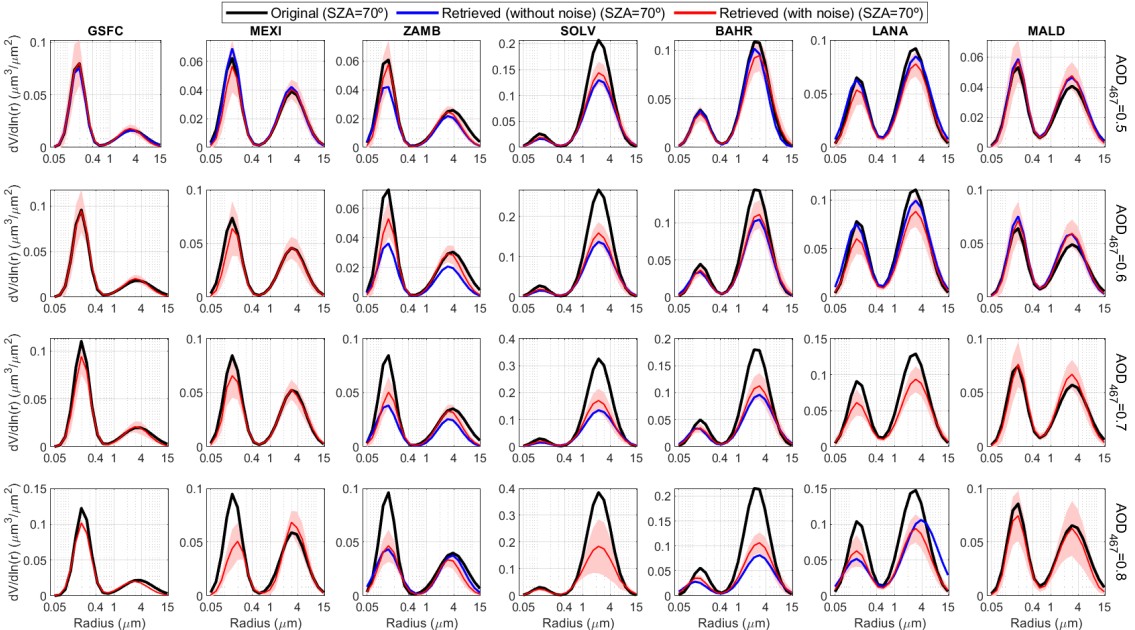

**Figure A22.** Original (black line), retrieved without noise (blue line), and median of all retrieved with noise (red line) aerosol volume size distributions under solar zenith angle (SZA) equal to 70º. The retrievals have been done including scattering angles up to 3º instead of 10º. These size distributions are represented for different aerosol types (one type per column) and for AOD (aerosol optical depth) at 467nm ($AOD_{467}$) values of 0.5 (first row), 0.6 (second row), 0.7 (third row) and 0.8 (last row). Red shadowed-area corresponds to $\pm$ the standard deviation of all averaged size distributions retrieved with noise.



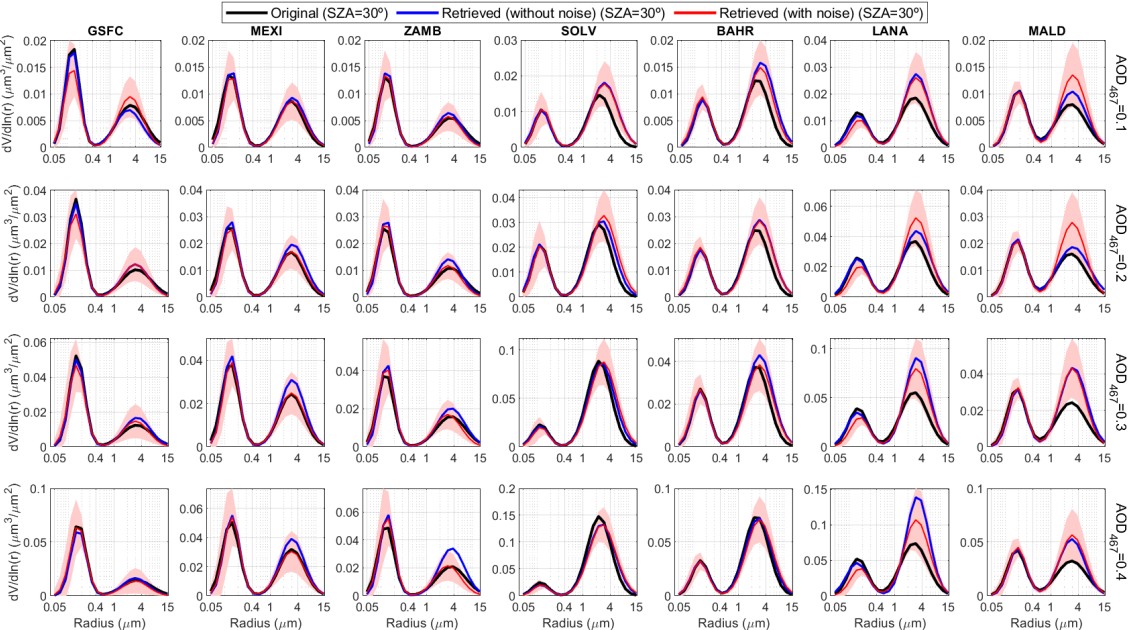

**Figure A23.** Original (black line), retrieved without noise (blue line), and median of all retrieved with noise (red line) aerosol volume size distributions under solar zenith angle (SZA) equal to 30º. These size distributions are represented for different aerosol types (one type per column) and for AOD (aerosol optical depth) at 467nm ($AOD_{467}$) values of 0.1 (first row), 0.2 (second row), 0.3 (third row) and 0.4 (last row). Red shadowed-area corresponds to ± the standard deviation of all averaged size distributions retrieved with noise.

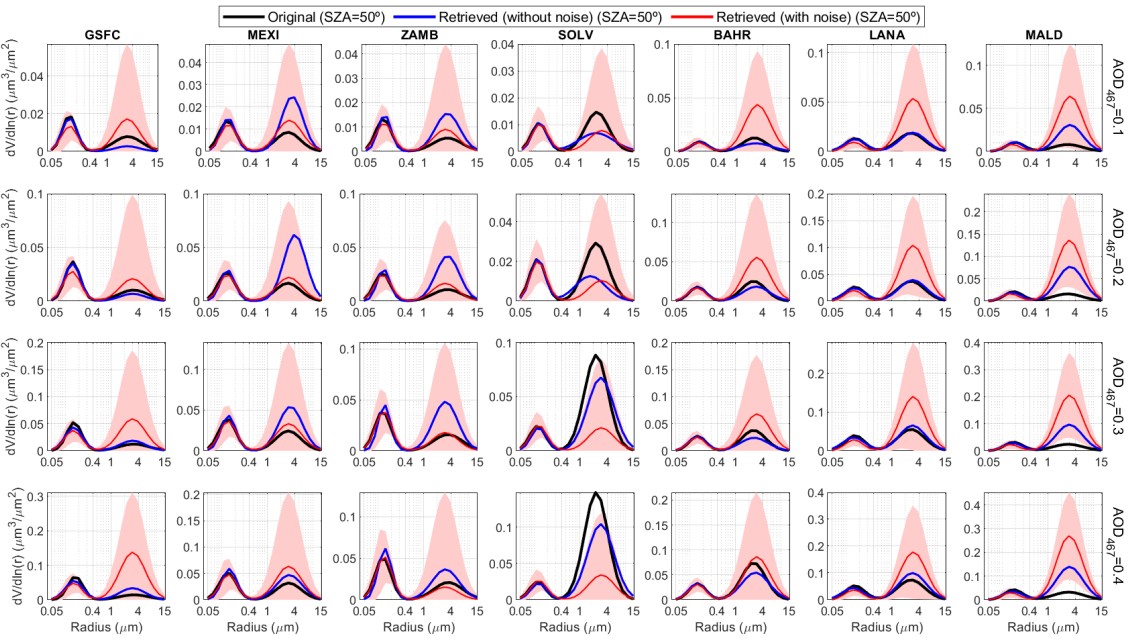

**Figure A24.** Original (black line), retrieved without noise (blue line), and median of all retrieved with noise (red line) aerosol volume size distributions under solar zenith angle (SZA) equal to 50º. These size distributions are represented for different aerosol types (one type per column) and for AOD (aerosol optical depth) at 467nm ($AOD_{467}$) values of 0.1 (first row), 0.2 (second row), 0.3 (third row) and 0.4 (last row). Red shadowed-area corresponds to ± the standard deviation of all averaged size distributions retrieved with noise.



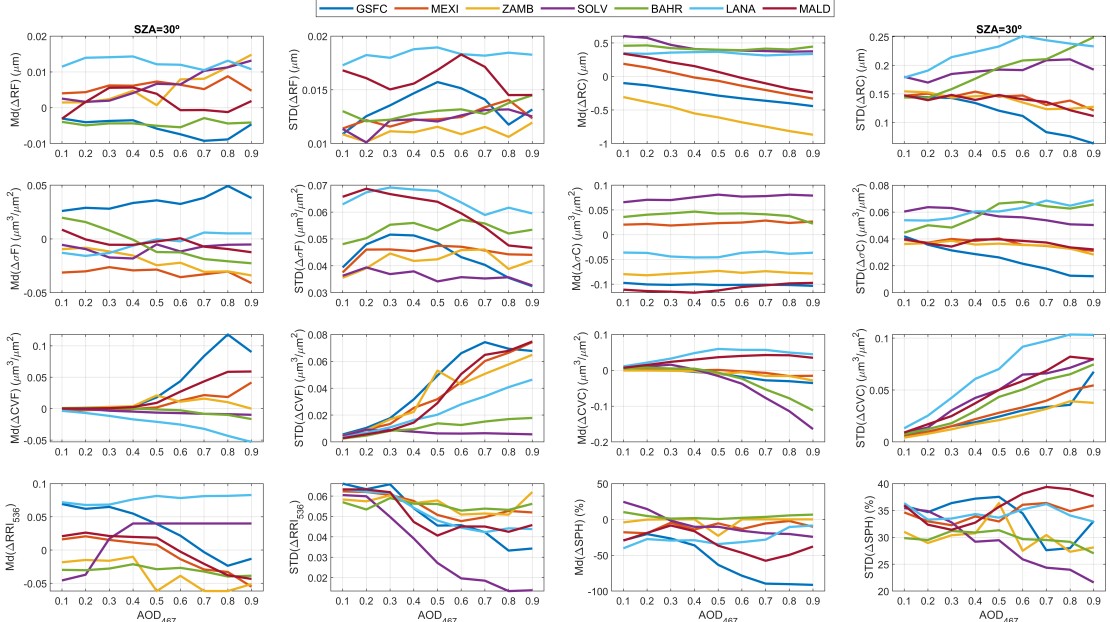

**Figure A25.** Median (Md) and standard deviation (STD) of the $\Delta$ differences between the available retrieved aerosol properties with noise and the original ones under solar zenith angle (SZA) of 30°. The aerosol properties shown are: volume median radius of fine (RF) and coarse (RC) modes; standard deviation of log-normal distribution for fine ($\sigma$F) and coarse ($\sigma$C) modes; aerosol volume concentration for fine (VCF) and coarse (VCC) modes; real part of refractive index at 536 nm ($RRI_{536}$); and the fraction of spherical particles (SPH). These Md and STD values are represented as a function of ($AOD_{467}$) for different aerosol types.



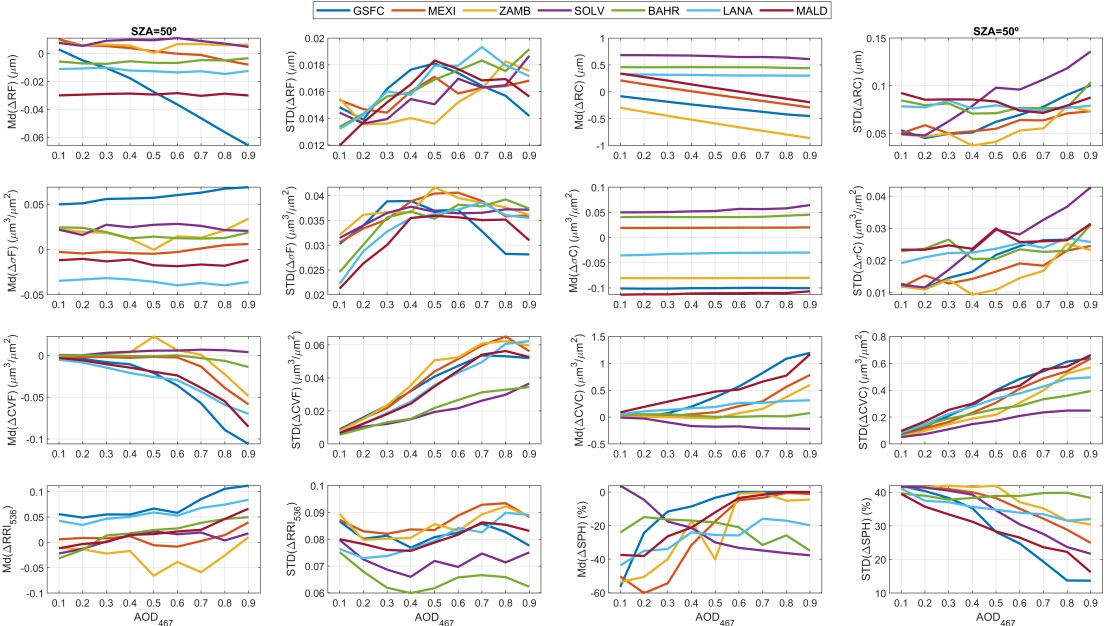

**Figure A26.** Median (Md) and standard deviation (STD) of the $\Delta$ differences between the available retrieved aerosol properties with noise and the original ones under solar zenith angle (SZA) of 50°. The aerosol properties shown are: volume median radius of fine (RF) and coarse (RC) modes; standard deviation of log-normal distribution for fine ($\sigma$F) and coarse ($\sigma$C) modes; aerosol volume concentration for fine (VCF) and coarse (VCC) modes; real part of refractive index at 536 nm ($RRI_{536}$); and the fraction of spherical particles (SPH). These Md and STD values are represented as a function of ($AOD_{467}$) for different aerosol types.



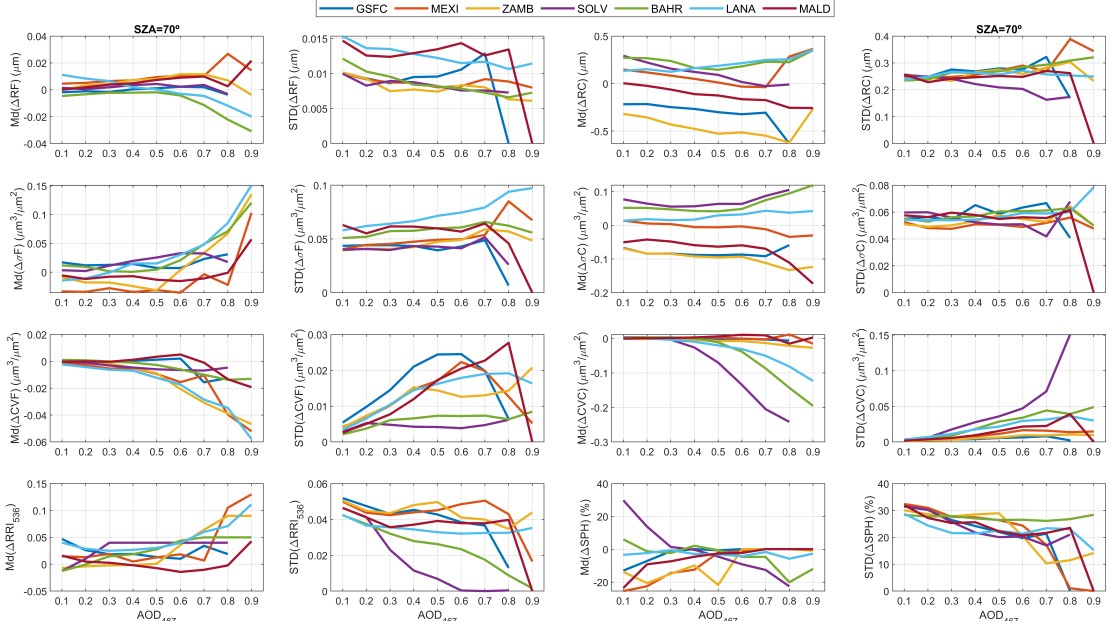

**Figure A27.** Median (Md) and standard deviation (STD) of the $\Delta$ differences between the available retrieved aerosol properties with noise and the original ones under solar zenith angle (SZA) of 70º. The retrievals have been done including scattering angles up to 3º instead of 10º. The aerosol properties shown are: volume median radius of fine (RF) and coarse (RC) modes; standard deviation of log-normal distribution for fine ($\sigma$F) and coarse ($\sigma$C) modes; aerosol volume concentration for fine (VCF) and coarse (VCC) modes; real part of refractive index at 536 nm ($RRI_{536}$); and the fraction of spherical particles (SPH). These Md and STD values are represented as a function of ($AOD_{467}$) for different aerosol types.



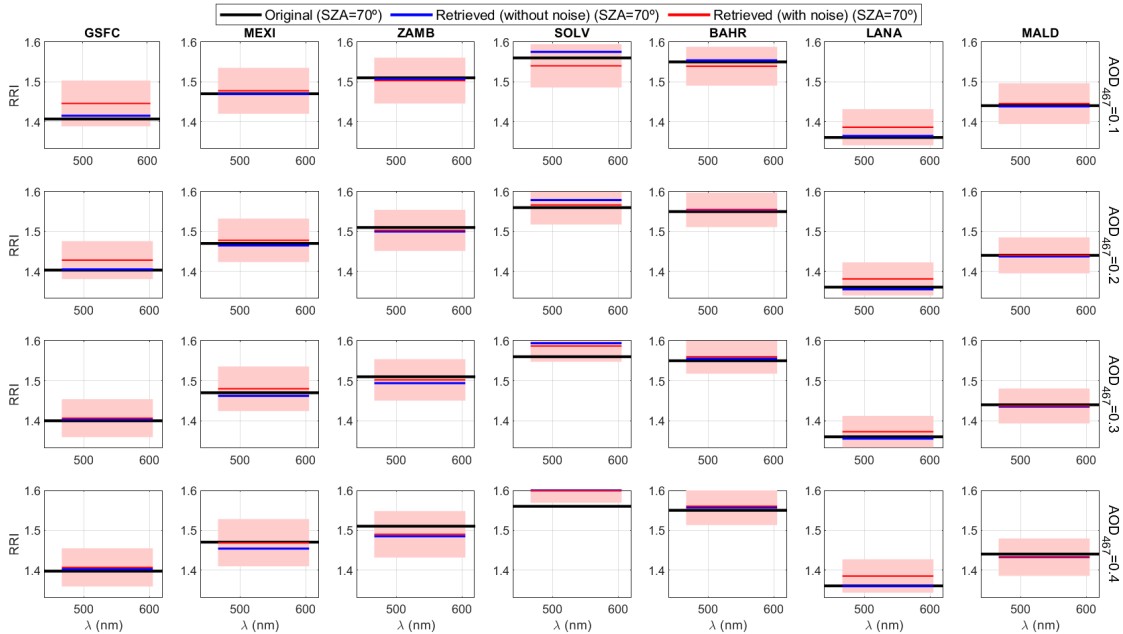

**Figure A28.** Original (black line), retrieved without noise (blue line), and median of all retrieved with noise (red line) real part of refractive indices (RRI) under solar zenith angle (SZA) equal to 70°. These AOD values are represented for different aerosol types (one type per column) and for AOD at 467nm ($AOD_{467}$) values of 0.1 (first row), 0.2 (second row), 0.3 (third row) and 0.4 (last row). Red shadowed-area corresponds to ± the standard deviation of all averaged size distributions retrieved with noise.

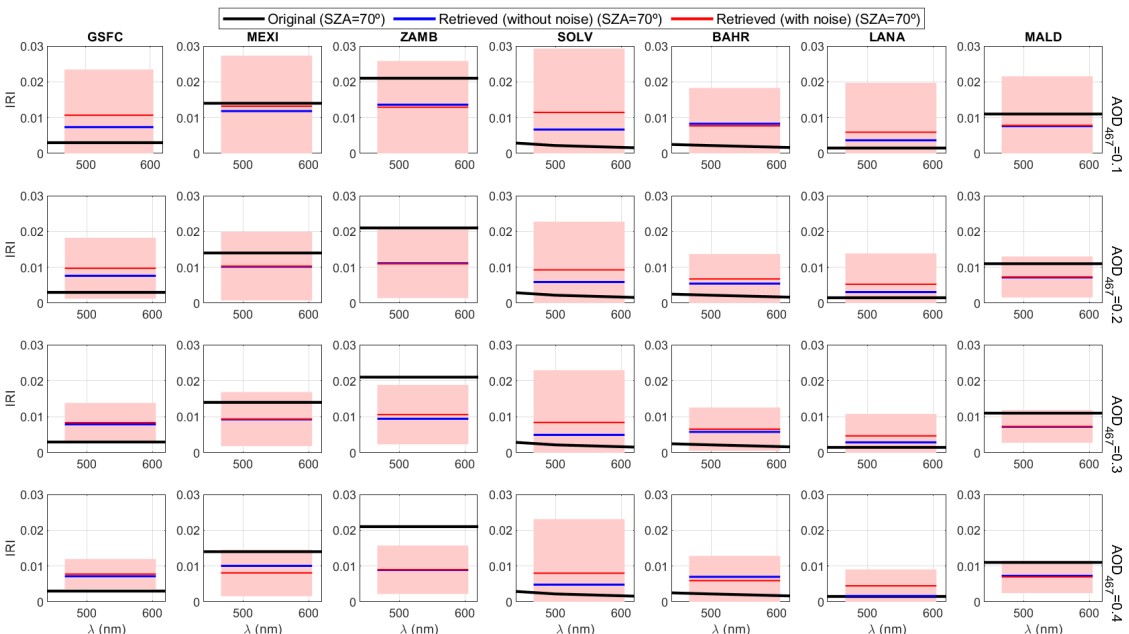

**Figure A29.** Original (black line), retrieved without noise (blue line), and median of all retrieved with noise (red line) imaginary part of refractive indices (IRI) under solar zenith angle (SZA) equal to 70º. These AOD values are represented for different aerosol types (one type per column) and for AOD at 467nm ($AOD_{467}$) values of 0.1 (first row), 0.2 (second row), 0.3 (third row) and 0.4 (last row). Red shadowed-area corresponds to $\pm$ the standard deviation of all averaged size distributions retrieved with noise. These results are obtained assuming IRI as an additional parameter to be retrieved in GRASP-CAM configuration.



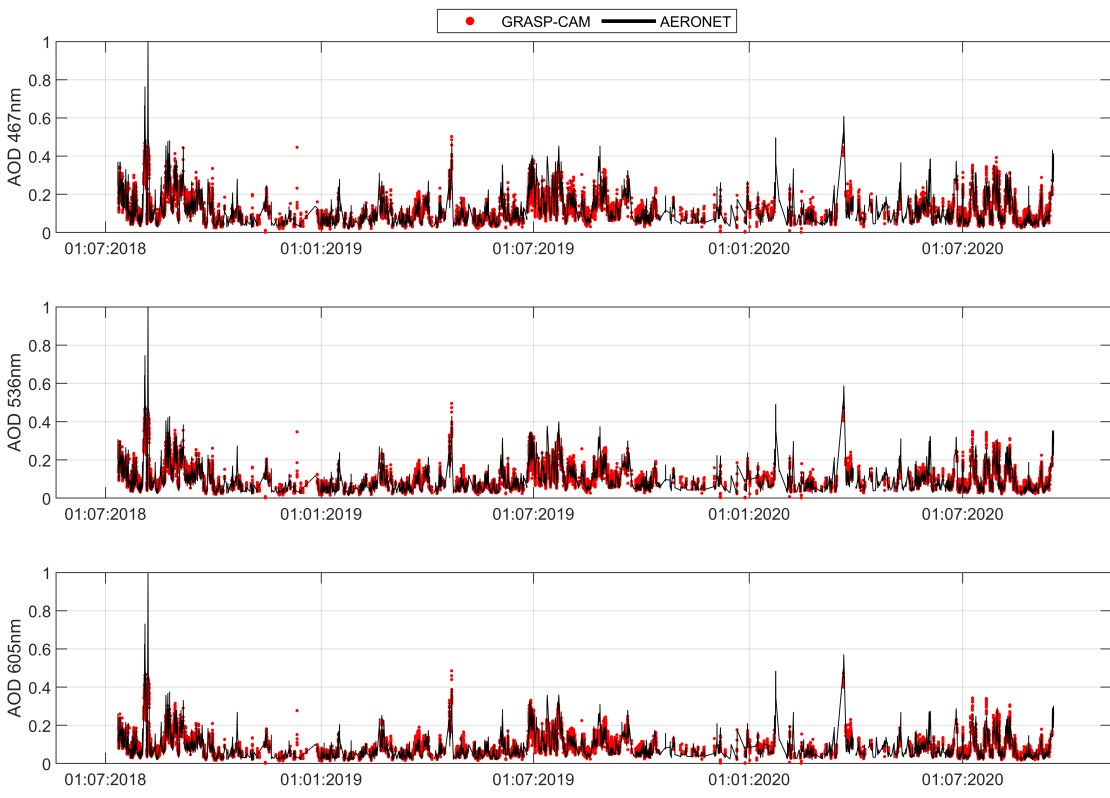

**Figure A30.** Aerosol optical depth (AOD) at 467 nm (upper panel), 536 nm (middle panel) and 605 nm (bottom panel) retrieved by GRASP under single-pixel approach (GRASP-CAM) and by AERONET at Valladolid from $11^{st}$ July 2018 to $15^{th}$ September 2020.





**Figure A31.** Δ differences on the aerosol optical depth (AOD) retrieved by GRASP under single-pixel approach (GRASP-CAM) and the obtained from AERONET at 467 nm (upper panel), 536 nm (middle panel) and 605 nm (bottom panel) as function of the solar zenith angle (SZA). The retrievals not satisfying the criteria of at least one scattering angle $leq14°$ are also included in the graphs. Colour legend represents the density of the plotted data points.



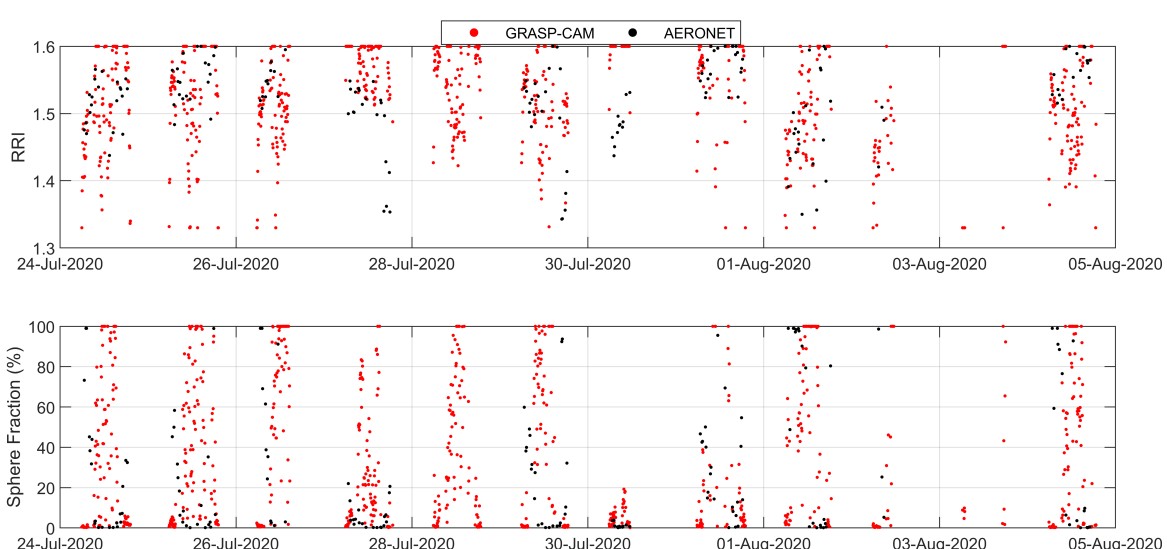

**Figure A32.** Real part of refractive index (RRI; upper panel) and the fraction of spherical particles (SPH; bottom panel) retrieved by GRASP under single-pixel approach (GRASP-CAM) and by AERONET at Valladolid from $24^{th}$ July 2020 to $4^{th}$ August 2020. RRI from AERONET corresponds to the mean of the RRI at 440 and 675 nm.



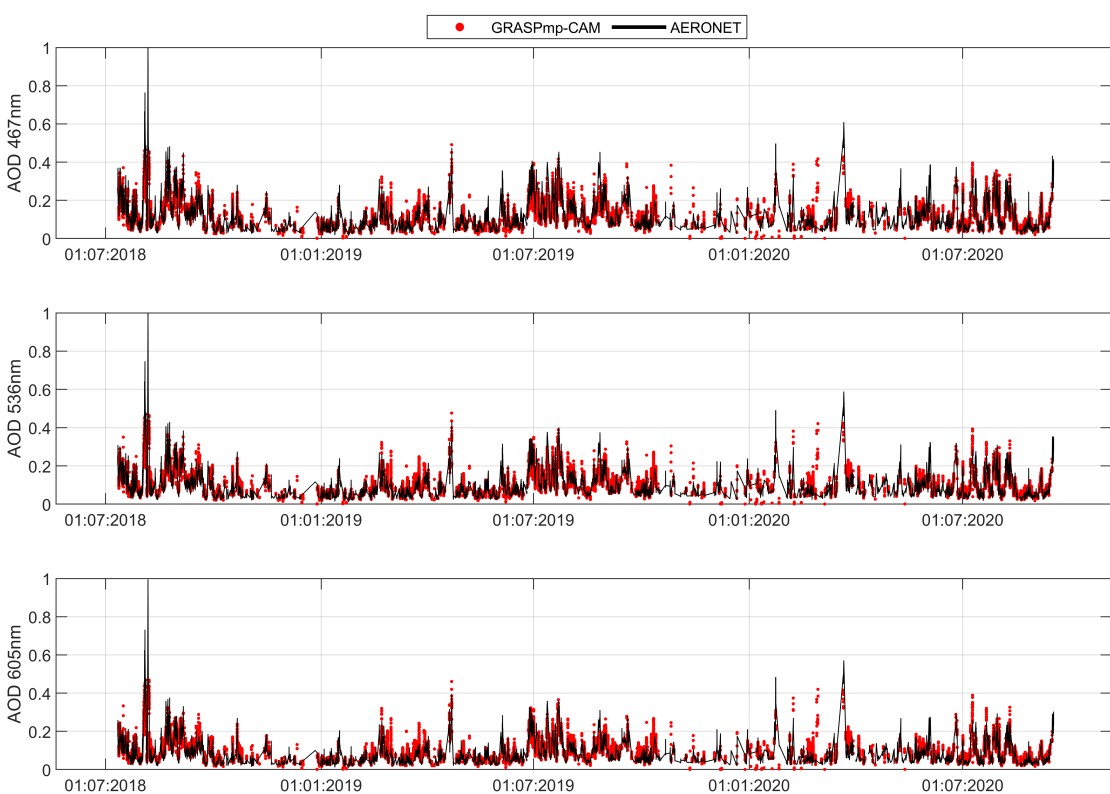

**Figure A33.** Aerosol optical depth (AOD) at 467 nm (upper panel), 536 nm (middle panel) and 605 nm (bottom panel) retrieved by GRASP under multi-pixel approach (GRASPmp-CAM) and by AERONET at Valladolid from $11^{st}$ July 2018 to $15^{th}$ September 2020.

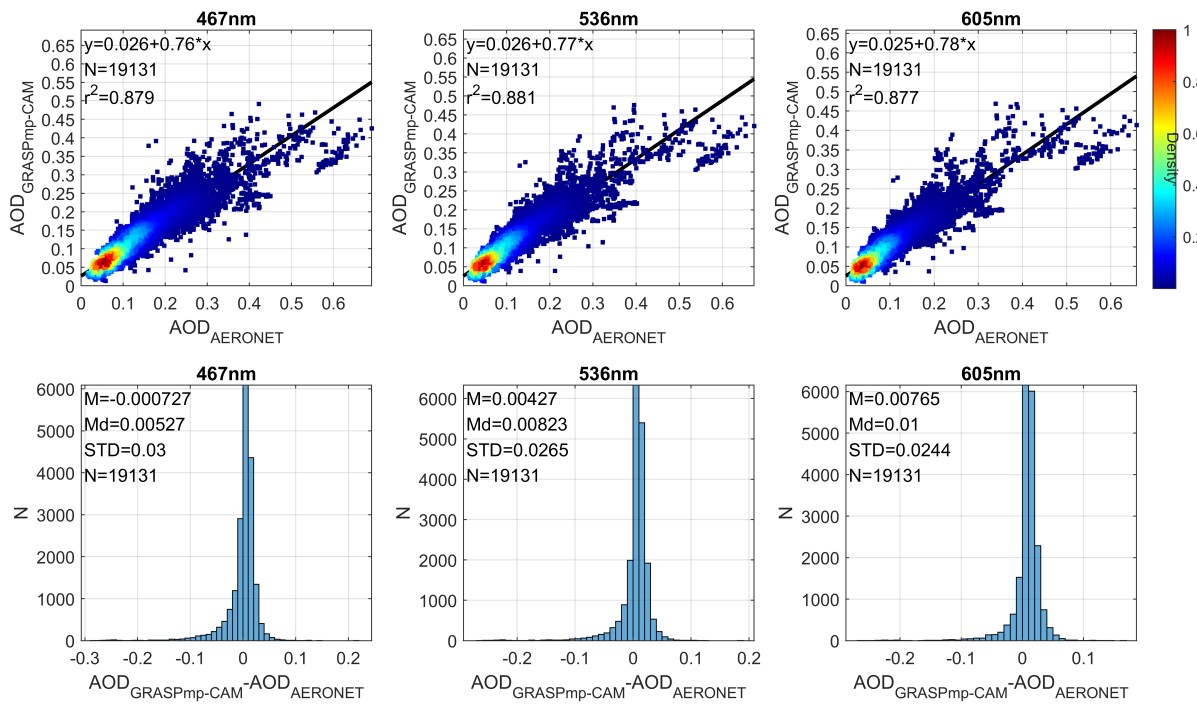

**Figure A34.** Upper panels) density scatter-plots of the aerosol optical depth (AOD) retrieved by GRASP in multi-pixel approach (GRASPmp-CAM) versus the AOD from AERONET at 467 nm (left panel), 536 nm (middle panel) and 605 nm (right panel); linear fit (black line) and its equation and the determination coefficient ($r^2$) are also shown. Bottom panels) Frequency histograms of the differences on AOD from GRASPmp-CAM and AERONET at 467 nm (left panel), 536 nm (middle panel) and 605 nm (right panel); The mean (M), median (Md) and standard deviation (STD) of these differences are also shown. Data obtained under solar zenith angle (SZA) values between 47.2º and 64.2º are not included.



**Figure A35.** Median (Md; middle panel) and standard deviation (STD; bottom panel) of the Δ differences on the aerosol optical depth (AOD) retrieved by GRASP under multi-pixel approach (GRASPmp-CAM) and the obtained from AERONET at 467, 536 and 605 nm for different AOD bins. The available number of ΔAOD data (N) per AOD bin is also shown in upper panel for the three wavelengths. Data obtained under solar zenith angle (SZA) values between 47.2º and 64.2º are not included.



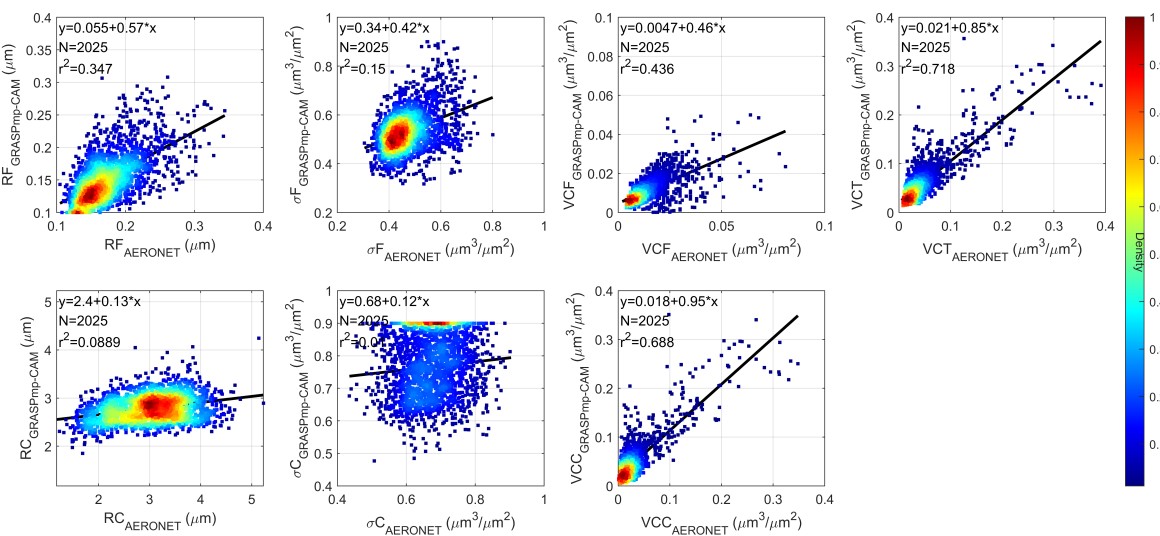

**Figure A36.** Density scatter-plots of the aerosol size distribution properties retrieved by GRASP in multi-pixel approach (GRASPmp-CAM) versus the ones retrieved by AERONET; linear fit (black line) and its equation and the determination coefficient ($r^2$) are also shown. These size distribution properties are: volume median radius of fine (RF) and coarse (RC) modes; standard deviation of log-normal distribution for fine ($\sigma$F) and coarse ($\sigma$C) modes; and aerosol volume concentration for fine (VCF) and coarse (VCC) modes and the total value (VCT). Data obtained under solar zenith angle (SZA) values between 47.2° and 64.2° are not included.





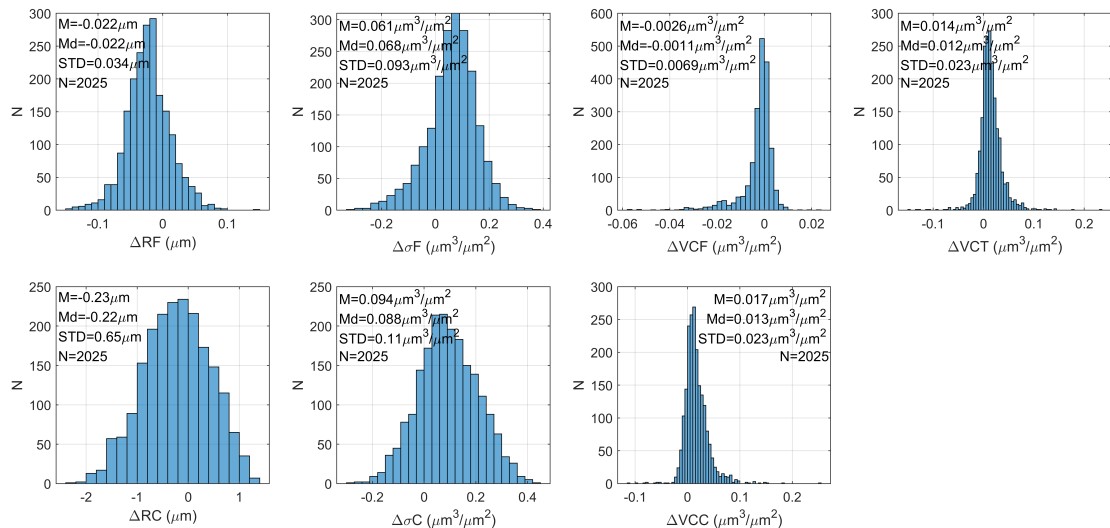

**Figure A37.** Frequency histograms of the Δ differences on the aerosol size distribution properties retrieved by GRASP in multi-pixel approach (GRASPmp-CAM) and the ones retrieved by AERONET; The mean (M), median (Md) and standard deviation (STD) of these differences are also shown. These size distribution properties are: volume median radius of fine (RF) and coarse (RC) modes; standard deviation of log-normal distribution for fine ($\sigma$F) and coarse ($\sigma$C) modes; and aerosol volume concentration for fine (VCF) and coarse (VCC) modes and the total value (VC). Data obtained under solar zenith angle (SZA) values between 47.2º and 64.2º are not included.

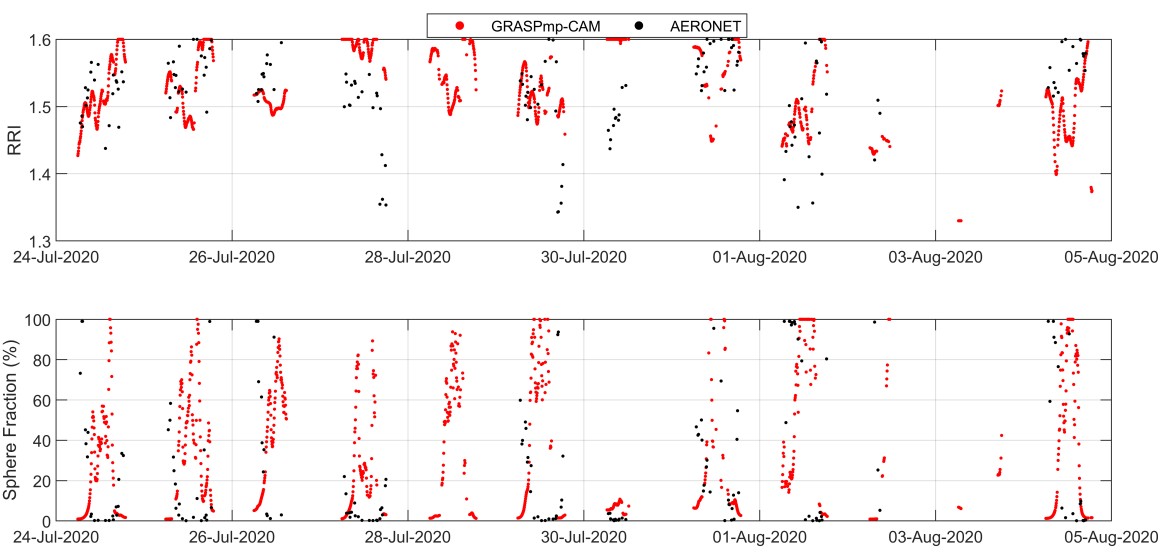

**Figure A38.** Real part of refractive index (RRI; upper panel) and the fraction of spherical particles (SPH; bottom panel) retrieved by GRASP under multi-pixel approach (GRASPmp-CAM) and by AERONET at Valladolid from $24^{th}$ July 2020 to $4^{th}$ August 2020. RRI from AERONET corresponds to the mean of the RRI at 440 and 675 nm.