# Peer review of "Retrieval of aerosol properties using relative radiance measurements from an all-sky camera"

_Atmospheric Measurement Techniques, 2021_

## Author Comment (AC1)

**Response to the Referee Reed Espinosa comments for the manuscript "Retrieval of aerosol properties using relative radiance measurements from an all-sky camera" By Roberto Román et al. in AMTD**

First, we are grateful for the effort of the referee and his detailed review. Reviewer comments are in black font (RC), and author comments (AC) in red font.

**Author's answer to Referee Reed Espinosa**

RC: The manuscript investigates aerosol retrievals performed on normalized sky radiances obtained from an upward looking all-sky camera. Retrievals are first simulated using synthetic data corresponding to a variety of aerosol types to estimate uncertainties under known conditions. The retrieval is then applied to real data obtained from a sky camera located in Valladolid, Spain and the resulting retrieved parameters are compared with their counterparts from AERONET inversions. It is found that the method has good sensitivity to AOD in scenes with low to moderate aerosol loading and it can characterize some aspects of the particle size distribution, especially in the fine mode.

The study demonstrates that upward looking cameras have the potential to retrieve useful aerosol information even without an absolute calibration. This approach has the potential to provide a low cost means of significantly expanding ground-based remote sensing coverage and the topic is certainly relevant for AMT. There are a few points that I feel need further clarification though, particularly some details regarding the preparation of the measurements used as input to the retrieval. Additionally, I would recommend another round of proof reading for English language usage.

———————————————

Detailed Comments

———————————————

RC: 1) LN 60: Since the comparisons are with retrieved values the term "retrievals" is probably more appropriate than "real measurements".
AC: Done.

RC: 2) LN 91: "geometries" should be "scan" since each of the two scans contains observations at multiple geometries.
AC: Done.

RC: 3) LN 104: It would be helpful to specify the field of view of this fisheye lens.
AC: The field of view is 185º and this information has been added in the revised manuscript.

RC: 4) LN 106: Given that stock Bayer filters are typically quite broad, will there still be significant spectral crosstalk (i.e., band overlap) between the three channels? It would be good to provide some description of the magnitude of this effect as well as a definition of exactly how the "effective wavelengths" are calculated. I would also be interested to hear the authors thoughts on how this could potentially impact the retrievals (perhaps at a later point in the manuscript).

AC: The spectral response of the Bayer filters, the triband filter and the convolution of both are on Figure 1 of the paper Antuña-Sánchez et al. (2021). The overlapping between Bayer filters is reduced with the triband filter, but unfortunately not fully neglected (Mainly the Green channel presents sensitivity to blue and red wavelengths).

The effective wavelengths are the averages of the different effective wavelengths obtained by the convolution of different sky radiation spectra (simulated under different aerosol conditions) with the camera spectral response. This is well explained in the Section 3.1 of Antuña-Sánchez et al. (2021); hence we think it is not necessary to explain it in this manuscript and the reference is enough. To be more concise we have rewritten the paragraph as:

"*The sensor has an RGB Bayer filter plus another triband filter reducing the bandwidth of the three RGB channels; these responses are shown in Figure 1 of Antuña-Sánchez et al. (2021). The effective wavelengths of these channels are 467, 536 and 605 nm (see Section 3.1 of Antuña-Sánchez et al. (2021) for the explanation of how these wavelengths were calculated).*"

This mentioned overlap is undesirable, but we need to deal with it as a first approximation. It is interesting to discuss about how this technical problem affects the retrievals, but it is already done indirectly. The overlap affects indirectly to the retrieval, but its effect is direct on the measured sky radiances. The uncertainty on these radiances caused by this overlap and the rest of uncertainty sources together was quantified in Antuña-Sánchez et al. (2021) by a comparison with independent observations. This uncertainty is mentioned in Section 2.1.3 of this paper: "*(-0.4±3:3)%, (-0.5±4:3)%, and (-0.4±5:3)% for 467, 536 and 605 nm, respectively*". In the present work this uncertainty has been used in the Sensitivity Analysis of Section 3.2 for the aerosol retrievals. In fact, this section analyses how the spectral overlap problem and other sources of uncertainty are propagated onto the final aerosol retrieval.

RC: 5) LN 122: I believe the measurements made here cover to most of the sky. Presumably using observations at so many viewing angles would place a high computational burden on the retrieval but what was the motivation for subsampling these observations to the AERONET hybrid scan viewing geometry in particular?

AC: Reviewer is right, it would be possible to use many more angular measurements and not only hybrid or almucantar. However, the number of input parameters has strong influence on the retrieval: computation time would increase, and we should not change the number of input angles in arbitrary way, otherwise the results would be hardly comparable. Moreover, there is considerable redundant information. Therefore, we have chosen the hybrid scan for two main reasons: 1) For cloud-screening, where a symmetric scan with respect to the Sun position, is helpful because the symmetric points must be similar under cloudless conditions (thus scans like principal plane are undesirable; in contrast, almucantar and hybrid scans are good candidates); and 2) for covering a long range of scattering angles, in this case almucantar and hybrid scans satisfy this condition if the solar zenith angle (SZA) is high, but the reached scattering angles by almucantar at

low SZA values are low while hybrid scan still reaches high scattering angles when the SZA is low. In addition, the almucantar scan in this work cannot be used for SZA angles between 48° and 65°, since all the almucantar angles are in the banned area for these conditions; fortunately, hybrid scan presents points out of the banned area even for the 48°-65 SZA range. Finally, AERONET uses this scan (and also almucantar) for its retrievals, and this is a well-established network that can be considered as reference. The text tries to explain that in the next sentence:

"*The chosen geometry to extract relative radiances is the AERONET hybrid geometry – rejecting the angles over the banned areas–, since this geometry allows long scattering angles even for low SZA values and it presents a symmetry with respect to the Sun position which is useful for cloud-screening.*"

RC: 6) LN 126: This is a little ambiguous in my view. Is the spectral dependence accurately captured here as well, or just the angular dependence? It would be good to include an equation and/or make the description of the normalization procedure more precise.
AC: The normalization captures the angular dependence, not the spectral dependence. Trying to clarify that, an equation has been added and the text has been modified as:

"*Afterwards, the remaining relative radiances for each wavelength are normalized, dividing each value by the total sum of all of observations, obtaining a normalized sky radiance (NSR). To clarify, NSR values are calculated by Eq. 1:*

$$NSR_i(\lambda) = \frac{SR_i(\lambda)}{\sum_{i=1}^{N(\lambda)} SR_i(\lambda)}$$

*where $SR_i(\lambda)$ is the i-measurement of the total of N (this value depends on wavelength) camera sky radiances (in arbitrary units) at the effective $\lambda$-wavelength.*"

RC: 7) LN 128: I am struggling to track the remainder of this paragraph. Which uncertainties are used for the 5% screening criteria? If all observations with uncertainties greater than 5% are removed in QA, how is the uncertainty of the remaining quality screened 605nm data still above 5%?
AC: The sky radiances from camera have their own uncertainty due to readout noise and random shot noise and the method itself to obtain these radiances (it is explained in detail in Antuña-Sánchez et al., 2021). The propagation of these noises to the camera sky radiances can be calculated (and it is calculated), which provides the propagated uncertainty on the camera sky radiance. Unfortunately, the uncertainty of the real camera sky radiance is larger than the propagated value because there are other sources of uncertainty like the width of the spectral response of the camera filters, among others. The real uncertainty on camera sky radiances is bigger than the propagated, therefore this propagated uncertainty looks like an indicator of low-quality data because a high propagated uncertainty points out high real uncertainty and then, a not enough quality. Antuña-Sánchez et al. (2021) found that rejecting the normalized sky radiances with a propagated uncertainty above 5%, the real uncertainty, obtained using independent sky radiance simulations, on the normalized sky radiances from the camera are the mentioned in the paper: (-0.4±3.3)%, (-0.5±4.3)%, and (-0.4±5.3)% for 467, 536 and 605 nm, respectively.
Trying to clarify that, the manuscript has been modified as follows:

*"Once the relative radiances are extracted, both left and right symmetrical sky points are averaged for each wavelength. The points with right-left differences above 5% are rejected (cloud contaminated); moreover, the points with a propagated uncertainty (derived from method uncertainties and camera readout and shot noises; see Antuña-Sánchez et al. (2021)) above 5% are also rejected. Both rejections are to warranty cloud-free conditions and high-quality data."*

We think the reviewer will find in the paper of Antuña-Sánchez et al. (2021) most of the answers to his questions related to the measurements. This reference is added several times in our paper, especially for the measurement's description, to help the reader about where the detailed information can be found.

RC: 8) LN145: Were any corrections applied to account for gas absorption or was it assumed negligible?
AC: It has been assumed negligible.

RC: 9) LN 150: How is the aerosol's vertical distribution modeled by GRASP in this study?
AC: The assumed vertical distribution in this study is a Gaussian layer centred at 2 km agl with a standard deviation of 250 m. This information has been added in the new manuscript version.

RC: 10) LN 152: Could other points in the FOV be used to provide more measurements if <6 data points are available within the hybrid scan geometry?
AC: Yes, it could. This is an advantage of the all-sky cameras because they view every sky angle, but also of GRASP which accepts any sky angle for the retrieval. To this end, we mainly need to develop two issues: 1) a good cloud mask to discard any cloud contaminated sky angle; and 2) some criteria to choose other cloudless points to cover a high range of scattering angles. Both issues could be achieved, but we think that is out of the scope of this paper. However, we will try to develop and implement that in future works. It is added in the conclusions:

*"The possibility that all-sky cameras offer to select alternative cloudless sky points when the standard sky points (from hybrid or almucantar scans) are contaminated by clouds should also be explored in the future."*

RC: 11) LN 156: A more precise explanation of "residual higher than the uncertainty" should be provided. The exact expression used to calculate the residual and the corresponding threshold value would be helpful.
AC: We add the next sentences to provide a more precise explanation about the residual and what are the thresholds:

*"This residual is calculated for each wavelength in relative way (%) as the root of the quadratic sum of all relative differences (%) between measured NSR values and modeled (forward module) ones, as Eq. 2 shows:*

$$Residual(\lambda) = 100\% \cdot \sqrt{\sum_{i=1}^{N(\lambda)} \left[ 2 \cdot \frac{NSR_i^{fwd}(\lambda) - NSR_i^{meas}(\lambda)}{NSR_i^{fwd}(\lambda) + NSR_i^{meas}(\lambda)} \right]^2}$$

*where the superscripts "fwd" and "meas" refer to modeled (calculated by the forward module) and measured NSR, respectively; N is the total number of NSR measurements used in the retrieval for each wavelength. This residual information is useful to reject non convergent retrievals; in this configuration we classify a retrieval as non convergent if the residual in NSR is higher than the uncertainty of the measured NSR for any of the three wavelengths. It means that a retrieval is only considered as convergent if the residual is below 3.7% for 467 nnm ((-0.4±3.3)%), but also below 4.8% for 536 nm((-0.5±4.3)%), and below 5.7% for 605 nm ((-0.4±5.3)%).*"

RC: 12) LN 165: Is it possible to provide some metric conveying the strength of these temporal smoothness constraints? (e.g., typical level of autocorrelation)

AC: The strength of these temporal smoothness constraints depends on the configuration of the GRASP settings. Specifically, this strength will depend on the "lagrange multiplier" chosen for each variable to be retrieved (RF, RC, VCF, VCC, σF, σC, RRI). More information about lagrange multipliers can be found in the references included in the papers with Oleg Dubovik as author. In this work, these lagrange multiplier values were chosen after looking for ones that constricted the temporal evolution enough to help the retrieval but that also provide flexibility for the retrieved parameters to change over time.

This multi-pixel approach is novel, and we only explore its use, however, as is written in the conclusions: its potential should be explored in more detail in future works. Then, we think it is not necessary to add more in-depth studies about temporal smoothness constraints in this paper.

Anyway, we have calculated the daily autocorrelation of the different aerosol properties retrieved by GRASPmp-CAM as well as GRASP-CAM. In this case, only data pairs belonging to the same day have been chosen because multi-pixel approach is applied only with data of one day (two consecutive days are not linked in our approach). Figure R1 shows this autocorrelation for single-pixel and multi-pixel approaches and different aerosol properties. The daily time series (5-min resolution) of retrieved aerosol properties are autocorrelated even without temporal smoothness constraints. As expected, the autocorrelation is higher for the GRASPmp-CAM products, but the autocorrelation differences on the derived AOD values are very low, in cases where both autocorrelations are high. These results point out that retrieved AOD time series are not significantly more autocorrelated with multi-pixel approach than with single-pixel one. This autocorrelation analysis is interesting, but we think that is out of the scope of the paper and, hence, it has not been added.

[Figure]

Figure R1: Autocorrelation coefficient as function of time lag for the different retrieved aerosol properties with GRASP-CAM and GRASPmp-CAM methods. Autocorrelation values calculated with less than 100 data pairs are not shown. Only data pairs measured in the same day are used for correlation.

RC: 13) LN 179: I'm not following what is meant by "using the Valladolid coordinates as a reference" and how the analysis went from seven to nine aerosol scenarios. Were two additional scenarios added corresponding to Valladolid aerosols? If so, how were these two scenarios derived?

AC: GRASP needs the geographical coordinates to simulate the sky radiance at one site. GRASP even allows these simulations for more than one site, which is useful for satellite data. Given that the real measurements are inverted at Valladolid site, we considered that the sensitivity analysis had more sense to be done using the Valladolid coordinates. It is mainly crucial for Rayleigh scattering calculation, since GRASP takes into account the altitude of the site. There are no two additional scenarios, there are 7 aerosol types (or models) and 9 aerosol loads per type. Then, 7 types x 9 loads = 63 aerosol scenarios. We change the sentences in the new manuscript as follows:

*"For each aerosol model, nine aerosol scenarios with different aerosol loads (AOD at 467 nm values ranging from 0.1 to 0.9 in 0.1 steps) have been defined; it is a total of 63 scenarios (7 aerosol models x 9 aerosol loads). In this work, all these scenarios are assumed that take place over Valladolid site (where real measurements are also inverted in Section 4) for the GRASP simulations, since GRASP needs the spatial coordinates (especially the site altitude for Rayleigh scattering) to calculate the sky radiances and aerosol properties."*

RC: 14) Figure 4/6: These figures might be a bit easier to follow if all panels on each row had the same y-axis scaling.

AC: Figure 4 has been changed following the reviewer comment. However, we decided to leave Figure 6 as in the previous version instead of as the reviewer suggested (Figure R2). This decision is due to the fact that, being all y-axes equal, some size distributions look very small and the fit between the retrieved values and original ones cannot be well observed (see Figure R2).

[Figure]

Figure R2: Figure 6 but with the same y-axis limits at each row.

RC: 15) LN 240: It is shown that the normalized sky radiances do not change significantly with AOD at high aerosol loads. Said another way, there are many possible aerosol concentrations that produce the observed normalized radiances so, while the retrieval lacks sensitivity to AOD in these cases, it still should be able to fit the measurements relatively well. Thus, I am wondering why the number of converged cases (N) decreases to near zero at the higher AODs?

AC: As the reviewer mentioned, it is expected that the inversion of sky radiances under high AOD values will converge to any of the multiple effective solutions. However, the results show the opposite. Some different reasons could be behind that. To analyse that, we focus on the retrievals without noise, because many of them do not converge either for high AOD values. We are interested in seeing how the aerosol parameters and residuals of the inversion evolve with the iteration number along the retrieval until it stops. Figure R3 shows the evolution with the iterative process of the three residuals and the 8 retrieved aerosol properties (+ AOD at 467 nm) for different aerosol loads and for the GSFC aerosol model. The retrieved parameters are divided by the same value of the original scenario in order to normalize (some aerosol properties change with the AOD in some aerosol types). Same graphs are shown at the end of this document (Annex RI) for the other 6 aerosol models (Figures RA1 to RA6).

Figure R3 shows that the reduction of residuals is not significant in a few iterations for high aerosol loads, which stops the retrieval. In general, the initial residual values are higher for high aerosol loads. It is caused by the initial guess of the aerosol scenario in the retrievals, i.e., all the retrievals start from the same aerosol scenario before the first iteration. AOD values of Figure R3 are a good indicator of that; the AOD at 467 nm of

the initial guess scenario is between 0.1 and 0.2. Then, the larger the AOD, the further the aerosol scenario to be retrieved is with respect to the initial guess. This could be the responsible of the lack of convergence for high aerosol loads, showing convergence only for the aerosol scenarios close to the initial guess. To confirm that, the same study has been done but considering an initial guess with four times more aerosol concentration (AOD multiplied by 4). Figure R4 shows these results for GSFC (see Figures RA7 to RA12 for the other aerosol models at the Annex RI of this document). In this case, the initial residuals are bigger for the low aerosol loads; however, they are strongly reduced in the first iterations achieving convergence. The non-convergence problem still remains for high aerosol loads; hence, the initial guess is likely not the reason for that.

[Figure]

Figure R3: Ratio between the aerosol parameters retrieved without noise and the original ones (subindex "O") as a function of the number of iterations in the retrieval, for different aerosol loads, for GSFC model and SZA=70º. Residuals and AOD at 467 nm are also included.

Back to Figure R3, it shows how the retrieved aerosol properties are in general closer to the original ones when the number of iterations increases. However, it does not happen for high aerosol loads in the cases of RRI, σF or VCC. This behaviour is also observed for the other aerosol models. It seems that the lack of information leads the retrieval to go down the wrong paths, looking for solutions that are increasingly remote from the original. This may be causing the aerosol scenario to reach a dead end after a few iterations and thus not to converge. This reason could be behind the convergence problem at high aerosol loads, however, this possible explanation has not been added to the final manuscript since we think it is beyond the aim of the paper and still not sufficiently investigated.

[Figure]

Figure R4: Figure R3 but changing the initial guess of aerosol concentration multiplied by 4.

RC: 16) Figure 6: The authors might consider including a plot of normalized (instead of absolute) size distributions so that the retrieval errors are not dominated by uncertainties in the total concentration, especially at high AODs.

AC: The plot of normalized size distribution is interesting, in fact it is shown in Figure R5; however, this plot is only useful to observe the behaviour of radius and σ values, both for fine and coarse modes. We consider that aerosol concentration is also an important part of the size distribution, and Figure 6 helps to visualize how is the agreement between retrieved and original values of the three parameters: radius, σ and volume concentration. Figure 7 already shows a detailed study about these parameters separately.

[Figure]

Figure R5: Figure 6 but for the normalized size distribution.

RC: 17) LN 265: How exactly is the lognormal's σ defined here? I generally take σ to be a unitless quantity.

AC: Reviewer is totally right, and the units of lognormal's σ were mistaken. σ is unitless, and this has been changed throughout the revised manuscript.

RC: 18) LN 295: It is a little surprising to me that more real retrievals pass the convergence test (>80%) than simulated retrievals (<65%, even for the lowest AODs). Do the authors have any thoughts as to why this might be?

AC: We are not sure why this happens. We think it could be caused by the perturbation of the simulated observations, which is done adding Gaussian random noise to each one the simulated observations. This abrupt way to perturb each simulated radiance could make that more retrievals does not converge in the simulations than in the real conditions.

RC: 19) LN 302: My understanding is that GRASP radiative transfer assumes plane-parallel geometry. Are errors in GRASP's forward modeling a significant concern at such high SZAs?

AC: It is true, GRASP assumes plane-parallel geometry. We expect that the uncertainty associated to this assumption add extra uncertainties for high SZA values, when plane parallel approach differs more to the real situation. It has been pointed out in the revised manuscript:

"*The rejection of retrievals under the highest SZA values is also motivated by the GRASP assumption of plane-parallel geometry for forward modeling; this assumption could still affect the remaining retrievals under high SZA angles.*"

RC: RC: 20) LN 368: Since the sentence is referring to the bias (not the spread) of the points "Most of the differences" should probably be replaced with a term that means the center of the distribution (i.e., "mode").

AC: It has been replaced by:

*"The mode of these differences is around zero for SZA values below 47.2°; however, for higher SZA values this mode increases from about 0 to about 0.01 (overestimation)."*

RC: 21) LN 451: I found the first few sentences of this paragraph to be very confusing. I recommend rewording.

AC: The sentences have been rewritten as follows:

*"Figure 15 shows the time series of AOD, retrieved by GRASPmp-CAM, at Valladolid for the same time period than in Figure 8 (Figure A33 shows the same AOD data but for the full measurements period). These AOD values from GRASPmp-CAM are similar to the ones obtained by GRASP-CAM (Figure 8), but their time evolution is less noisy; therefore, the AOD values retrieved by GRASPmp-CAM are closer to the AERONET values."*

RC: 22) LN 518: Why is AERONET able to retrieve absorption with hybrid geometry but GRASP-CAM has no sensitivity? Is it due the lack of absolute calibration in the GRASP-CAM data? If the authors have an intuitive explanation of this phenomenon, it would be helpful to include it.

AC: Yes, it is. AERONET can retrieve absorption, with hybrid and almucantar scans, because AERONET photometers measure absolute radiances. We think that the sentence *"There is not sensitivity of the NSR measurements to the aerosol absorption"* in the conclusions section is direct and concise and it is not needed to add any explanation. However, in order to provide an explanation, the next sentence has been added at the end of Section 3.2.2.

*"This lack of sensitivity to aerosol absorption is due the lack of absolute calibration in the GRASP-CAM measurements, because it is well known that absolute radiances contain the necessary information to retrieve aerosol absorptive properties, like in the AERONET retrievals (Dubovik and King,2000; Sinyuk et al., 2020)."*
* * *
Technical Comments
* * *
RC:

LN 25: "makes" should be "means"

LN 32: "using all" should read "all using"

LN 34: It would be clearer to say "...capable of measuring..."

LN 74: While the acronym is common, "m.a.s.l" should be defined within the text.

LN 144: "AWRONET" needs correcting.

LN 389: I think this should reference Figure 13, not 12.

LN 413: Should say "...than the ones obtained..."

AC: All these technical comments are right, and they have been corrected in the new version of the manuscript. Thanks.

**Annex RI. Supplementary figures for the Reviewer #1**

[Figure]

Figure RA1: Figure R3 but for MEXI aerosol model.

[Figure]

Figure RA2: Figure R3 but for ZAMB aerosol model.

[Figure]

Figure RA3: Figure R3 but for SOLV aerosol model.

[Figure]

Figure RA4: Figure R3 but for BAHR aerosol model.

[Figure]

Figure RA5: Figure R3 but for LANA aerosol model.

[Figure]

Figure RA6: Figure R3 but for MALD aerosol model.

[Figure]

Figure RA7: Figure R4 but for MEXI aerosol model.

[Figure]

Figure RA8: Figure R4 but for ZAMB aerosol model.

[Figure]

Figure RA9: Figure R4 but for SOLV aerosol model.

[Figure]

Figure RA10: Figure R4 but for BAHR aerosol model.

[Figure]

Figure RA11: Figure R4 but for LANA aerosol model.

[Figure]

Figure RA12: Figure R4 but for MALD aerosol model.

---

## Author Comment (AC2)

**Response to the Anonymous Referee #2 comments for the manuscript "Retrieval of aerosol properties using relative radiance measurements from an all-sky camera" By Roberto Román et al. in AMTD**

First, we are grateful for the effort of referee #2 and his/her review in detail. Reviewer comments are in black font (RC), and author comments (AC) in red font.

**Author's answer to Anonymous Referee #2**

**General comment**

RC: The manuscript covers aerosol retrievals using normalized radiances from an all-sky camera (simulated and measured) and the GRASP algorithm. The study has been very thoroughly performed and described, and presents interesting and relevant results regarding generalizing all-sky cameras (or normalized radiances) towards aerosol retrievals. The subjects also fits the scope of the journal and I recommend its publication, with some very minor comments.

**Specific comments**

RC: P4, L117. Is there a reference, where the hybrid geometry is described in more detail?
AC: Section 2.1.2 refers to the hybrid scan as first time. Here, the reference of Sinyuk et al. (2020) is included. Section 4 of Sinyuk et al. (2020) describes in detail the geometry of hybrid scan. It has been added in the new manuscript version:

*"The chosen geometry to extract relative radiances is the AERONET hybrid geometry (see Section 4 of Sinyuk et al., 2020) –rejecting the angles over the banned areas–, since this geometry allows long scattering angles even for low SZA values and it presents a symmetry with respect to the Sun position which is useful for cloud-screening."*

RC: P7, L192pp. I think the explanation of NSR sensitivity to AOD with dominance of aerosol scattering (Rayleigh vs. aerosol) is not very convincing (wouldn't this AOD sensitivity be the same in an atmosphere without Rayleigh?). Maybe it is because the normalization factor (sum over all radiances) increases with AOD and reduces the relative differences in NSR.
AC: The sensitivity of NSR to AOD has been studied for an atmosphere without Rayleigh in order to support our hypothesis. The same simulations shown in Figure 2 have been obtained with GRASP forward module but assuming no Rayleigh atmosphere. Figure R1 shows the obtained values for three aerosol types (Figure R2 shows all aerosol types). As result, we can observe that normalized radiances do not present a clear dependence on AOD load as it happens in the real atmosphere (Rayleigh + aerosol) at least for low aerosol loads (Figure 2 of the manuscript). In fact, all NSR values of Figure R1 are like the ones obtained in Figure 2 for high AOD loads (dominance of aerosol scattering). All

these results support our hypothesis about AOD sensitivity in the range between pure Rayleigh and pure aerosol.

[Figure]

Figure R1: Normalized sky radiance (NSR) for solar zenith angle (SZA) of 70º at: 467 nm (top row), 536 nm (middle row) and 605 nm (bottom row), as a function of scattering angle (Θ) for different AOD (at 467 nm) values. Left, middle and right columns correspond to GSFC, ZAMB and SOLV aerosol models, respectively, under an atmosphere without Rayleigh scattering.

[Figure]

Figure R2: Normalized sky radiance (NSR) under a solar zenith angle (SZA) of 70º at 467 nm (top row), 536 nm (middle row) and 605 nm (bottom row) as a function of scattering angle (Θ) for different AOD (at 467 nm) values and for nine aerosol models under an atmosphere without Rayleigh scattering.

Regarding the normalization factor, it has been represented in Figure R3 for all the conditions used in the paper as a function of AOD. As can be observed, especially for SZA=70º, this normalization factor is not increasing always with AOD. Therefore, we have discarded the hypothesis suggested by the reviewer.

[Figure]

Figure R3: Normalization factor, sum of all sky radiances, to normalize sky radiances at 605, 536 and 467 nm, as a function of AOD (at 467 nm) values and for nine aerosol models under a solar zenith angle (SZA) of 30º, 50º and 70º.

Finally, to clarify these results, Figure R2 has been added as supplementary material (Figure A5) and the following text has been included:

*"To confirm the proposed explanation, the same simulations as in Figure 2 (and Figure A1) have been calculated but considering an atmosphere without Rayleigh scattering in GRASP. These NSR simulations are shown in Figure A5 and point out that NSR does not significantly depend on AOD when Rayleigh scattering is negligible, even for low AOD values, showing always similar values than the ones observed in Figure 2 (and Figure A5) for high AOD values; this result supports our hypothesis. Finally, multiple scattering and surface albedo also affect NSR but their impact on NSR is small, at least for the analyzed aerosol loads"*

RC: Fig 2. What is the reason for the "kinks" in the radiances at around 40° scattering angle? If I am not mistaken, these features do not appear in almucantar scans (radiance plotted against azimuth not scattering angle).

AC: Reviewer is right, this strange behaviour (kinks or smoothness break points) is not observed in a standard cloudless almucantar scan, either as function on scattering or

azimuth angle. However, we are dealing with hybrid scans. To have a better idea about how this scan works, we recommend observing Figure 1 of the present paper and Figure 12 of Sinyuk et al. (2020). When the SZA is above 75º, the hybrid scan starts pointing the sky angles from sun aureole and then it goes increasing the azimuth and zenith angle until the zenith is 75º (sun altitude below 15º). After reaching the elevation angle of 15º the scan further proceeds at a fixed zenith angle by varying the azimuth angle similarly to that of an almucantar scan except that the view angle is not equal to the SZA (Sinyuk et al., 2020). The observed kinks appear just in the change of scanning when the elevation of 15º is reached. This change produces the change in the variation of sky radiance with scattering angle. The view scattering angle when hybrid scan reaches elevation angle of 15º (where kinks should appear) depends on SZA (lower SZA, higher scattering angle); this can be observed in Figures A2 and A3.

RC: P9, L218. So the noise in the synthetic data is the same as stated on p5, L128? Also, what is distribution of the random noise? I would guess normal, but it should better be stated.
AC: Yes, the noise is the same as stated on L128, and the distribution is normal. We add in the new manuscript that the noise distribution is Gaussian:

*"To obtain more realistic results, random noise (Gaussian distributed) has been added to each simulated sky radiance in accordance with the NSR uncertainty of the camera product (see Section 2.1.3)."*

RC: P11, L258. It works surprisingly well for SZA=30°, considering AERONET limit of 50° (is that correct?) with almucantar measurements only. So the hybrid scan seems effective at small SZA.
Would this mean that really most of the information is in the small scatting angles, <100°?
AC: Yes, it works well for low SZA values, this is the advantage of hybrid scan compared to almucantar. Hybrid scan can reach higher scattering angles even for low SZA values. Almucantar scans cannot do that. Sinyuk et al. (2020) remarked that: *"for a hybrid scan made at 25° SZA the scattering angle range of measurements is 100º, which is the same scattering angle range that the almucantar scan is capable of at 50º SZA"*. The limit of 50º of AERONET is for almucantar scans, but AERONET is also using now hybrid scans, extending the quality assurance of the derived products until SZA values of 25º. It is explained by Sinyuk et al. (2020): *"hybrid SSA retrievals for dust aerosols exhibit smaller variability with solar zenith angles (SZAs) than those of almucantar, which allows extension of hybrid SSA retrievals to SZAs less than 50º to as small as 25º."*

Each scattering angle range provide different information. In the present work we observed how the low scattering angles contain valuable information about the coarse mode. The AERONET criteria consider that a SSA retrieval with assured quality must include at least scattering angles up to 100º.

RC: Fig. 5. I am wondering if the higher deviations of the retrievals towards higher AOD, could also be due to (or at least affected by) the lower number of successful retrievals, i.e. a statistical effect. Of course, the retrieval success decreases due to decreasing sensitivity to AOD.

AC: This is an interesting issue, however, the deviations on all AOD values also increase from AOD=0.1 to AOD=0.4, and in this range the number of successful retrievals does not show any significant variation. The main problem is not only the deviation, which could be higher for the low number of data, it is also the median values, which represents the accuracy, and they reach high absolute values for high AOD loads.

A similar analysis of Figure 5 has been done but not considering more than 200 successful retrievals. It is shown in Figure R4. In this case we can observe how the deviation is similar for low AOD values even with less successful retrievals. It indicates that the results could be significant even when the number of successful retrievals available is low for high AOD values.

[Figure]

Figure R4: Median (Md) and standard deviation (STD) of the $\Delta$ differences between the available retrieved aerosol properties with noise perturbed radiances, and the original (reference) properties. The amount of available retrievals (N) is also shown. Only the retrievals with solar zenith angle (SZA) equal to 70° are used. The aerosol properties provided are: aerosol optical depth (AOD) at 467 nm (AOD$_{467}$), 536 nm (AOD$_{536}$) and 605 nm (AOD$_{605}$). The Md and STD are represented as a function of (AOD$_{467}$) for different aerosol types. A maximum of 200 successful retrievals have been used.

RC: Since the STD is really interesting in relation with the Md (e.g. is ΔAOD=0 within the errorbar also for high AOD?), I believe it would be a good alternative to combine right and left columns of the plots and plot STD as error bars (or uncertainty bands) for Md. Maybe with a second uncertainty of the mean MD/sqrt(N), N successful retrievals. This will be too much for all scenarios in one plot, so this would need a different grid of plots. Might be worth a try.

AC: We agree with the reviewer, and we would like to plot standard deviation as an errorbar. However, if this is done, only one type of aerosol could be represented in each panel, because part of the data and the error bars would overlap each other, making graphs very confusing. Putting only one aerosol type on each panel, then we would have to make many more graphs (3.5 times more). This makes the panels much smaller, and also does not allow us an easy comparison of the results between aerosol types. An example has been represented in Figure R5, with the standard deviation represented as a shadow. In this case we also have no space to represent the number of data. All these issues have made us decide to keep the Figure 5 and 7 as they were in the previous version.

[Figure]

Figure R5: Median (Md) of the Δ differences between available retrieved aerosol properties with noise perturbed radiances, and the original (reference) properties as a function of (AOD$_{467}$). Only the retrievals with solar zenith angle (SZA) equal to 70º are used. The aerosol properties provided are: aerosol optical depth (AOD) at 467 nm (AOD$_{467}$), 536 nm (AOD$_{536}$) and 605 nm (AOD$_{605}$). ±standard deviation is added as a shadowband.

RC: Fig. 6 As a general question: If red is the median of noise retrievals, I would expect red and blue line to be very similar (as the retrieval input, the median of noised radiances should be the radiance without noise). This is almost always true except for the coarse mode of MALD. Any idea why?

AC: It is true that the mean of all NSR measurements calculated with noise should be equal to the NSR without noise. However, it cannot be extrapolated to the retrieved properties. The relationship between changes on NSR measurements and changes on aerosol properties is not linear, since the inversion process is more complex. Hence, in the case of MALD the differences between retrieved with and without noise can be related to the lack of information about coarse mode in the used NSR measurements (scattering angles above or equal 10º); in the case of MALD it is compensating a lower σC value with a higher VCC. In Figure A21 of the older version, the same size distributions are shown but retrieved with lower scattering angles (more coarse mode information), and

the mentioned differences between retrieved with and without noise disappear. In addition, in the Figure A20 of the older version, the differences on the coarse mode of the size distributions between retrievals with and without noise are also significant in other aerosol models like LANA or SOLV.

RC: Fig.7. Again, as for Fig.5, I think it would better if Md and STD are plotted together.
AC: We have the same problem than in Fig. 5 (discussed above) and, hence, we have decided not to modify this figure.

RC: Fig. A33. There is actually quite little one can read from this plot of the complete time series. On this time scale, it would be easier to visualize aggregate values, e.g. monthly means.
AC: The main objective of this image is to give a broad vision of the amount of data that has been obtained with GRASP-CAM and to observe how they correlate with AERONET values during the more than two years. The monthly mean (and their standard deviation) values of AOD from GRASP-CAM and AERONET are shown in Figure R6, where we can observe how both data series correlate in time. However, our paper is focused on the performance of the instantaneous AOD obtained by GRASP-CAM, and hence, we think it is more in agreement with the figure of the previous version than with Figure R6.

[Figure]

Figure R6: Monthly means (±standard deviation) of aerosol optical depth (AOD) at 467 nm (upper panel), 536 nm (middle panel) and 605 nm (bottom panel) retrieved by GRASP under single-pixel approach (GRASP-CAM) and by AERONET at Valladolid from July 2018 to September 2020. AERONET data have been interpolated to the all-sky camera wavelengths.

RC: Finally, the manuscript can in general be improved with respect to English grammar / language. For example, missing articles: P1, L1. the GRASP code. P1,L8. As a result.

AC: We have tried to improve English in the last version of the manuscript, including both reviewer suggestions about missing articles.

---

## Author Response (AR2)

**Response to the Referee Reed Espinosa technical corrections for the manuscript "Retrieval of aerosol properties using relative radiance measurements from an all-sky camera" By Roberto Román et al. in AMTD**

First, thank you for this review. Reviewer comments are in black font (RC), and author comments (AC) in red font.

**Author's answer to Referee Reed Espinosa**

The essential details that I felt were missing in the initial submission have all been included in the revised manuscript and all my other concerns have been satisfactorily addressed. A few further comments related to the authors responses are listed below in case the authors want to consider them in their next revised submission, but that is at the authors discretion.

RC (5) LN 122: If the angles are to be subsetted, the justification for using AERONET hybrid geometry in particular is well described in the sentence the authors reference. But would it also help to also add a sentence or two describing the reasons mentioned in the authors review response for taking a subset of the the full range of angles in the first place? (i.e., redundant information, computational expense, etc.)
AC: We understand the comment, but we consider that the most important issues (long angle range and symmetry for cloud-screening) are now clear in the text and the addition of this extra information will not be more helpful and it could make more confuse the text.

RC (7) LN 128: The added clarification on the distinction between propagated and normalized uncertainties is very helpful here. I might also suggest the authors change the word "warranty" to something like "guarantee".
AC: We change it by "assure":
*"Both rejections are to assure cloud-free conditions and high-quality data."*

RC (8) LN 145: I think the assumption that gaseous absorption is negligible here is probably safe, but it I would recommend explicitly stating that this assumption was made in the text so that the reader is not left wondering how that aspect of the system was treated.
AC: Yes, it is better, therefore we have added the next:
*"The impact of gaseous absorption on normalized sky radiance is assumed negligible at the camera effective wavelengths"*

RC (13) LN 179: The word "scenarios" is still taking on multiple meanings here which makes the text confusing. I would recommend changing the first sentence of the referenced passage to say something like: "For each aerosol model, nine aerosol loads with different AOD values (AOD at 467 nm ranging from 0.1 to 0.9 in 0.1 steps) have been defined; this produces a total of 63 scenarios (7 aerosol models x 9 aerosol loads)." Also, I think GRASP only uses lat/lon for bookkeeping purposes. Thus, it would be more concise, and probably clearer, to just say that the altitude of the Valladolid site is used in the tests and avoid the more general term "spatial coordinates" all together.

AC: We agree with referee and the sentence about 63 scenarios has been changed by the suggested one.

Regarding lat/lon issue, we have changed the full sentence and "spatial coordinates" by "geographical coordinates". Rayleigh scattering also depends on latitude, since the acceleration of gravity changes on latitude and it must be considered (see Eq. (10) of Bodhaine et al., 1999). Longitude maybe is only for bookkeeping, but the other two geographic coordinates are important for the reproducibility of our results, and hence we think it is better to indicate that. The text has been changed by:

*"GRASP requires geographical coordinates as input, especially the site elevation and latitude for Rayleigh scattering default calculation (Bodhaine et al. 1999); in this work, for the GRASP simulations, the 63 mentioned scenarios are assumed that take place over Valladolid site coordinates. These coordinates are chosen to be the same coordinates than the used in the inversion of real measurements, recorded at Valladolid, shown in Section 4."*